# Lipofuscin, Its Origin, Properties, and Contribution to Retinal Fluorescence as a Potential Biomarker of Oxidative Damage to the Retina

**DOI:** 10.3390/antiox12122111

**Published:** 2023-12-13

**Authors:** Małgorzata B. Różanowska

**Affiliations:** 1School of Optometry and Vision Sciences, College of Biomedical and Life Sciences, Cardiff University, Maindy Road, Cardiff CF24 4HQ, Wales, UK; rozanowskamb@cardiff.ac.uk; 2Cardiff Institute for Tissue Engineering and Repair (CITER), Redwood Building, King Edward VII Avenue, Cardiff CF10 3NB, Wales, UK

**Keywords:** lipofuscin, retina, fundus autofluorescence, fluorescence imaging, oxidative stress, photooxidation, oxidation, biomarker, age-related macular degeneration

## Abstract

Lipofuscin accumulates with age as intracellular fluorescent granules originating from incomplete lysosomal digestion of phagocytosed and autophagocytosed material. The purpose of this review is to provide an update on the current understanding of the role of oxidative stress and/or lysosomal dysfunction in lipofuscin accumulation and its consequences, particularly for retinal pigment epithelium (RPE). Next, the fluorescence of lipofuscin, spectral changes induced by oxidation, and its contribution to retinal fluorescence are discussed. This is followed by reviewing recent developments in fluorescence imaging of the retina and the current evidence on the prognostic value of retinal fluorescence for the progression of age-related macular degeneration (AMD), the major blinding disease affecting elderly people in developed countries. The evidence of lipofuscin oxidation in vivo and the evidence of increased oxidative damage in AMD retina ex vivo lead to the conclusion that imaging of spectral characteristics of lipofuscin fluorescence may serve as a useful biomarker of oxidative damage, which can be helpful in assessing the efficacy of potential antioxidant therapies in retinal degenerations associated with accumulation of lipofuscin and increased oxidative stress. Finally, amendments to currently used fluorescence imaging instruments are suggested to be more sensitive and specific for imaging spectral characteristics of lipofuscin fluorescence.

## 1. Introduction

Lipofuscin accumulates in postmitotic cells as autofluorescent deposits due to incomplete lysosomal digestion of autophagocytosed and phagocytosed material. The purpose of this review is to provide an update on the current understanding of the role of oxidative stress and lysosomal dysfunction in lipofuscin formation and its effects on cells as well as their fluorescence properties. Next, the retinal pigment epithelium (RPE) is considered as the main site of lipofuscin accumulation in the retina, and the roles of oxidative stress, lysosomal dysfunction, and vitamin A derivatives in the formation of RPE lipofuscin are discussed, as well as its structure, composition, and effects on function and viability of RPE cells and neighbouring structures. Due to the substantial photosensitizing properties of lipofuscin, current evidence of its potential involvement in light-induced retinal injury and contribution to the development of AMD is discussed. The fluorescence of RPE lipofuscin and its changes upon oxidation are discussed in detail.

Next, sources of fluorescence in the retina, other than lipofuscin, are discussed, including recent developments in fluorescence imaging of the retina, as well as current evidence on the prognostic value of retinal fluorescence for the progression of age-related macular degeneration (AMD), the major blinding disease affecting elderly people in developed countries.

In the retina, RPE lipofuscin provides the major contribution to fluorescence emission excited by UV, blue, and green light. RPE lipofuscin has the potential to increase oxidative stress and is susceptible to oxidative degradation, which results in spectral changes in its fluorescence characteristics: a decrease in its yellow-red fluorescence and a concomitant increase in its blue-green fluorescence. The evidence of lipofuscin oxidation in vivo and the evidence of increased oxidative stress and damage in AMD retina are reviewed, leading to the conclusion that imaging of spectral characteristics of lipofuscin fluorescence may serve as a useful biomarker of oxidative damage in vivo, which can be used in assessing the efficacy of potential antioxidant therapies in AMD and other retinal degenerations associated with accumulation of lipofuscin and increased oxidative stress. Finally, amendments to currently used fluorescence imaging instruments based on two-photon excitation are suggested, which could make them more sensitive and specific for imaging spectral characteristics of lipofuscin fluorescence emission while minimizing the excitation of other retinal fluorophores and photosensitizers. 

## 2. Lipofuscin: Formation, Composition, and Potentially Harmful Effects

Lipofuscin is an intracellular deposit composed of lipids, proteins, and carbohydrates, which are cross-linked and highly modified by end products of glycation and/or lipid peroxidation [1,2,3,4,5,6]. It accumulates with age in the form of small yellow-orange granules surrounded by a lipid membrane as a result of incomplete lysosomal digestion of autophagocytosed and phagocytosed material. It is present in most organs of the body and in most types of cells, especially if they are highly metabolically active and/or are postmitotic [7,8,9,10,11].

The lipofuscin content is split during cell division into daughter cells, thereby diluting it. This option is not available in the postmitotic cells, so the lipofuscin level per cell can considerably increase with age [12]. It has been reported that the lipofuscin increase in cardiomyocytes is 0.6% of cell volume per decade, whereas in motor neurons of centenarians, up to 75% of cell volume can be occupied by lipofuscin [1]. Because of its age-related accumulation and colour, lipofuscin is also known as the age pigment. 

### 2.1. Lipofuscin Formation

The theory of the lysosomal origin of lipofuscin is supported by an increased accumulation of lipofuscin in diseases caused by dysfunction of autophagy and/or lysosomal degradation [13,14,15,16,17,18,19,20,21,22,23,24,25,26]. Among them, there is a group of neurodegenerative genetic disorders called neuronal ceroid lipofuscinosis (NCL), which is characterized by the formation of lipofuscin-like intracellular deposits in neurons and other cell types. The deposits are often referred to as ceroid to distinguish their abnormally early accumulation from the age-related accumulation of lipofuscin. However, Katz and Robinson proposed to use the term “lipofuscin-like pigments” instead of ceroids [27]. They also emphasized that different material from which lipofuscin originates is likely to lead to lipofuscin of different composition. Usually, NCL affects the brain and retina, often leading to vision loss, motor and cognitive regression, seizures, and premature death. It is a rare disease with an incidence of 1–3 per 100,000 and a prevalence of about 2–4 per million. 

Inhibition of lysosomal enzymes in experimental animals and in cells in vitro also results in the accumulation of lipofuscin-like deposits [28,29]. Accumulation of such deposits can be also induced in an earlier step than lysosomal degradation by exposure of cultured cells to inhibitors of autophagy [30,31,32]. If both—autophagy and lysosomal activities are inhibited, lipofuscin-like material can rapidly accumulate in the cytosol [33]. 

Consistently, a decrease in lipofuscin accumulation can be achieved by upregulation of autophagy by inhibition of the mammalian target of rapamycin complex 1 (MTORC1), by rapamycin, or by pharmacological activation of the transcriptional control of transcription factor EB (TFEB), which stimulates the expression of autophagy proteins, lysosomal membrane proteins, and lysosomal hydrolases [34,35,36,37,38,39,40,41,42]. 

An increased lipofuscin accumulation is also a characteristic feature of many, but not all, diseases associated with increased oxidative stress, and it has been demonstrated in numerous experiments in vitro and in vivo that accumulation of lipofuscin can be accelerated by increased oxidative stress [9,26,43,44,45]. For example, it has been shown that lipofuscin accumulates more rapidly in cultured cells or in animals exposed to increased oxygen tension, redox-active iron ions, or depleted of antioxidants [32,46,47,48]. 

Based on experimental findings on cultured rat cardiomyocytes, Brunk and colleagues proposed that autophagy and lysosomal degradation of organelles producing reactive oxygen species and containing redox-active iron ions, such as mitochondria, can increase oxidative stress by facilitating lipid peroxidation [49]. Subsequently, the formation of oxidized and crosslinked biomolecules makes them no longer susceptible to lysosomal degradation. Iron chelators and antioxidants, such as vitamin E, can effectively inhibit lipofuscin accumulation by preventing lipid peroxidation.

### 2.2. Deleterious Effects of Lipofuscin and Potential Mechanisms Involved 

While there is some evidence that cells can survive and apparently still perform their function despite having a high lipofuscin content [50], there is also a growing body of evidence showing that lipofuscin can compromise cell function or even lead to cell death, suggesting that accumulation of lipofuscin cannot be viewed as a benign feature of ageing of postmitotic cells, particularly if they exhibit high metabolism and/or are under oxidative stress conditions [17,51]. For example, it has been shown on confluent fibroblasts and astrocytes that inhibition of autophagy by 3-methyladenine or inhibition of lysosomal enzymes by leupeptin leads to a rapid intracellular accumulation of lipofuscin-like material and eventually results in apoptotic cell death [52]. In cultured human fibroblasts, accumulation of lipofuscin is progressing rapidly but once it reaches a certain level, it stops and cells start to die [53]. Reaching that threshold can be accelerated by supplementing the culture medium with mitochondria oxidized by exposure to ultraviolet light. Oxidized mitochondria are phagocytosed and form lipofuscin even more rapidly, and the lipofuscin exerts a toxic effect upon reaching a similar level as that which causes toxicity in cells not exposed to exogenously oxidized mitochondria.

One of the potentially deleterious effects of lipofuscin is a decreased efficiency of lysosomal degradation of newly (auto)phagocytosed material. This effect can be due to the competition between lipofuscin and newly formed (auto)phagosomes for the primary lysosomes delivering the hydrolytic enzymes [46,51]. Autophagy is essential for the maintenance of cellular homeostasis [4,41,54]. This is particularly important in the case of mitochondria, for which autophagy dysfunction means that damaged mitochondria, with impaired ATP production and excessive production of superoxide, can accumulate. It has been proposed that, in the case of cardiac myocytes, such overloading with damaged mitochondria may increase oxidative stress and lead to cell death, eventually contributing to heart failure [54].

Another potential deleterious effect of lipofuscin is increased oxidation of biomolecules within lipofuscin and formation of toxic products, which can cause lysosomal membrane permeabilization, act as inhibitors of lysosomal proton pumps or hydrolases, or diffuse outside the lipofuscin granule, and damage cellular organelles and cytoplasmic molecules [55,56,57].

Such a scenario may occur after the autophagy of a mitochondrion (mitophagy) and its fusion with the lysosome. The low lysosomal pH can facilitate lipid oxidation mediated by the superoxide radical anion produced by mitochondria because the superoxide radical anion becomes protonated, and, in the protonated form, it can initiate lipid peroxidation and dismutate much more rapidly to the hydrogen peroxide than in the unprotonated form. Lysosomal degradation of autophagocytosed mitochondria can release iron ions from the Fe-S centres and metalloproteins, which are abundant in mitochondria [58,59,60]. Indeed, it has been reported for lipofuscin from many different tissues that it contains redox-active iron [61]. The interaction of ferrous iron with hydrogen peroxide leads to the decomposition of hydrogen peroxide releasing the hydroxyl radical. The hydroxyl radical is the most reactive free radical formed in the human body; it can react with different types of biomolecules and damage them with bimolecular rate constants close to diffusion-controlled limits. Importantly, the hydroxyl radical can initiate lipid peroxidation, which is a chain reaction; it can be initiated by a single free radical, such as the hydroxyl radical or the protonated form of superoxide radical (hydroperoxyl radical), leading to the oxidation of numerous lipid molecules via propagation steps. The initial products of lipid oxidation, lipid hydroperoxides, can undergo further reactions forming a wide range of products, including end products of lipid oxidation with reactive carbonyl groups, which can form adducts with the lysosomal proteins and with lipids with amine groups, such as phosphatidylethanolamine, thereby contributing to the formation of material not susceptible to the degradation by lysosomal enzymes. 

Extensive lipid peroxidation can lead to the permeability of the lysosomal membrane and leakage of proteolytic enzymes, such as cathepsin D, into the cytosol, where they can damage cellular components [57]. As a result of oxidative modifications of biomolecules, there is a formation of cross-links between proteins, nucleic acids, carbohydrates, and lipids, rendering them no longer susceptible to lysosomal degradation. Some of the end products of lipid oxidation are hydrophilic enough to diffuse outside the lipofuscin granules damaging biomolecules in the cytoplasm and cellular organelles. 

It is plausible that the oxidation and lysosomal dysfunction can potentiate effects of each other: inhibition of lysosomal degradative enzymes provides time and environment for the oxidation of (auto)phagocytosed material to proceed, resulting in the formation of material no longer susceptible to lysosomal degradation; certain oxidative modifications can directly inhibit the lysosomal enzymes and/or lysosomal proton pumps. The latter can lead to an increase in lysosomal pH, thereby decreasing the activity of several lysosomal enzymes. It is suggested that once the accumulation of lipofuscin starts affecting the lysosomal degradation of newly (auto)phagocytosed material, the process of further lipofuscin accumulation, and accumulation of damaged organelles, such as mitochondria, can spiral out of control, leading to massive cell dysfunction, loss of cell viability, and, eventually, numerous pathologies.

## 3. Lipofuscin Fluorescence

The characteristic feature of all lipofuscins is their golden-yellow fluorescence when excited with ultraviolet A or visible light [62,63,64,65]. There are numerous reports of fluorescence emission spectra of lipofuscin from different tissues. Not all of them include spectra corrected for the changes in sensitivity of detection for different emission wavelengths or specify whether or not such a correction was performed, which can lead to confusion [66,67]. While the measurement of absolute fluorescence emission requires specialized equipment, the fluorescence emission spectra acquired from a spectrofluorometer can be corrected by measurement of the fluorescence spectrum of a compound used as a standard for which the emission spectrum is available and covers the spectral range of interest. The correction factor then can be calculated for each wavelength as the ratio of the fluorescence intensity of the standard spectrum to its measured spectrum. 

In cases where care was taken to consider if the spectral correction was needed, it has been shown that the spectral range of fluorescence emission varies depending on the tissue from which the lipofuscin is derived. For example, it has been reported that the emission from lipofuscin in rat brain neurons exhibits maxima at 570 nm or 600 nm, from the kidneys—at 620 nm, whereas from the adrenal glands—at 660 nm [43,62,65,68]. Fluorescence emissions with maxima at about 600 nm were observed in the resected neocortex of 18–28-year-old patients with drug-resistant epilepsy and in the neocortex from the autopsy of two human cadavers [43].

In attempts to identify the components of lipofuscin responsible for its fluorescence, several model systems have been employed that attempt to create fluorophores with characteristics similar to those of lipofuscin. For example, different products with fluorescent properties have been synthesized by incubation of end products of lipid peroxidation, such as 4-hydroxynonenal (4HNE), malondialdehyde (MDA), or epoxy-heptenal with proteins or amino acids [69,70,71]. These small lipid-derived aldehydes modify the protein structure and form chromophores with the ability to fluoresce. Consistently, oxidation of polyunsaturated lipids, such as arachidonic acid, triggered by iron ions in the presence of glycine, leads to the formation of products with fluorescent properties [72]. Also, proteins modified by nonenzymatic glycation exhibit fluorescence. It has been demonstrated that there are some similarities in fluorescence features between those synthetic products and the fluorescence of oxidized subcellular components [73]. However, in most cases, the obtained fluorophores exhibit the emission maxima corresponding to blue light from the range of 400–490 nm, which shift towards longer wavelengths only upon a considerable increase in their concentrations [74]. In contrast, autoxidation of docosahexaenoic acid (DHA) results in the formation of products with an emission maximum at about 610 nm when excited with 488 nm light and broad-band fluorescence with a maximum in blue when excited with UV light [75].

## 4. Lipofuscin in the Retina

### 4.1. Retinal Pigment Epithelium (RPE) Is the Major Site of Lipofuscin Accumulation in the Retina

The retina is a thin highly organized tissue sandwiched between the vitreous and the choroid (Figure 1). Its neural part is separated from the highly vascularised choroid by the monolayer of RPE cells [76]. RPE cells provide the blood–retina barrier and are responsible for the uptake of vitamin A from the blood and photoreceptive neurons, its storage, and its enzymatic transformation to the chromophore of visual pigments, 11-*cis*-retinaldehyde. RPE cells are also responsible for the transepithelial transport of nutrients and waste products between the photoreceptive part of the retina and the fenestrated choriocapillaris of the choroidal blood supply. Moreover, RPE cells are responsible for the molecular renewal of photoreceptors, including phagocytosis and lysosomal degradation of distal tips of photoreceptor outer segments (POS). Photoreceptors—rods and cones—are considered the most metabolically active cells in the body. Despite that, the major site of lipofuscin accumulation in the retina is the RPE, where, based on quantification of areas of cellular cross-sections, lipofuscin can occupy up to 19% of the cytoplasmic area in RPE from human cadavers of 81–90 years of age [77,78]. 

### 4.2. The Major Source of RPE of Lipofuscin Is Phagocytosis of Photoreceptor Outer Segments

The massive accumulation of lipofuscin in the RPE is mainly due to the high daily phagocytic load, with some of the phagocytosed material accumulating over a lifetime as a result of incomplete lysosomal digestion [77,83,84,85,86,87]. Every day, about 7–10% of POS is shed and phagocytosed by the RPE. POS in rod photoreceptors contain stacks of flattened discs made from a lipid bilayer densely populated by visual pigments. For rods in the human retina, the dense stack of 600–1000 discs is surrounded by the plasma membrane, forming a cylindrical rod-like shape, 2 µm in diameter and 24–35 µm long [88,89]. The outer segments of cone photoreceptors are tapered towards the distal end and are also made of stacks of discs, but the discs are continuous with the plasma membrane and are formed by its deep infoldings, allowing for the diffusion of membrane components from the basal side to the distal tip and vice versa [90,91]. These structural differences in POS of rods and cones may have important consequences for their phagocytosis and lysosomal degradation; in the case of rods, RPE phagocytoses the oldest discs, whereas, in the case of cones, the phagocytosed material is a mixture of old and recently added lipids and proteins. Upon phagocytosis, an individual phagosome content is approximately 10-fold greater in diameter than the diameter of a typical lysosome [92]. A single RPE cell, depending on its location in the human retina, phagocytoses daily tips of 12 to 43 photoreceptors, suggesting a massive phagocytic load of up to 324 µm^3^ POS volume per cell per day [93]. In the centre of the fovea, each of the columnar RPE cells overlies 23 cone outer segments, suggesting that an RPE cell from that area, 14 µm tall and 10 µm in diameter, phagocytoses daily about 173 µm^3^ of POS, corresponding to a remarkable 16% of the volume of the cell body.

### 4.3. Oxidative Stress, Lysosomal Dysfunction, and Vitamin A Derivatives as Contributors to the Accumulation of RPE Lipofuscin

Similarly to lipofuscin in other cell types, it has been demonstrated that oxidative stress induced by excessive iron or depletion of antioxidants and/or autophagy/lysosomal dysfunction can increase the accumulation of RPE lipofuscin (reviewed in [94]). For example, a rapid lipofuscin accumulation in the RPE of rats and dogs can be induced by intravitreal injection of an inhibitor of lysosomal protease, leupeptin [28,95,96,97], or by intravitreal injection of ferrous sulphate or ferric ammonium citrate [98,99,100]. Rats given a diet depleted from vitamin E and selenium for 32 weeks leading to a massive loss of glutathione peroxidase activity and 10- and 4.4-fold decreased levels of vitamin E in the blood plasma and retina, respectively, developed lipofuscin-like fluorescence in the RPE with some fluorescent pigment present in the endothelium of choroidal capillaries and at the tips of the rod outer segment [101]. Selenium is a trace element needed for the synthesis of selenoproteins, including glutathione peroxidases (GPXs), which can detoxify hydrogen peroxide and lipid hydroperoxides. Vitamin E is an efficient scavenger of lipid-derived peroxyl radicals and, therefore, can inhibit the propagation of lipid peroxidation [94]. 

Another study has shown that dietary depletion of albino rats with vitamin E for 26 weeks led to a 3.9-fold increase in RPE lipofuscin when quantified by morphometric analysis of TEM sections, and a 2.3-fold increase when quantified by fluorescence [102]. At the end of the depletion period, the levels of vitamin E were not detectable and the density of photoreceptors decreased by 19%. Remarkably, lipofuscin formed as a result of vitamin E depletion appears to have lower yields of fluorescence than lipofuscin formed in the presence of vitamin E. When animals were depleted with vitamin E, the concomitant depletion of vitamin. A 1.3- and 1.8-fold decrease in lipofuscin in comparison with animals with normal vitamin A when quantified by morphometry and fluorescence, respectively. By morphometry and fluorescence, lipofuscin content in rats depleted from both vitamins was 5.9- and three-fold greater, respectively, than in rats depleted only with vitamin A. In comparison with rats on a normal diet, the dietary depletion of vitamin A resulted in 1.8- and 2.3-fold decreases in lipofuscin content and fluorescence, respectively. The kinetics of depletion of vitamins A and E from the retinas were not investigated in this study. It was reported, however, that after 26 weeks of diet with absent vitamin A, its retinal levels were reduced by 90% and 75% in the presence and absence of dietary vitamin E, respectively. A vitamin A-depleted diet had no effect on the density of photoreceptors. These results demonstrate that increased oxidative stress imposed by the depletion of vitamin E leads to massive accumulation of lipofuscin, whereas the partial depletion of vitamin A leads to a decrease in lipofuscin accumulation. Interestingly, the fluorophores of RPE lipofuscin from rats depleted of both vitamin E and vitamin A were resistant to extraction by organic solvents [27]. 

Accumulation of lipofuscin was also documented by TEM in 14-month-old capuchin monkeys on a vitamin E-depleted diet (*Cebus albifrons* and *apella*) and cynomolgus (*Macaca fascicularis*) monkeys [103]. Interestingly, vitamin E deficiency resulted in the degeneration of photoreceptors and accumulation of lipofuscin in the macula, whereas the peripheral retina appeared unaffected. These findings point to the important roles of increased oxidative stress, antioxidants, lysosomal dysfunction, and vitamin A as contributors to the accumulation of RPE lipofuscin.

#### 4.3.1. Role of Retinaldehydes and Lipids in Lipofuscin Formation

In addition to the effects observed in other cell types, the lysosomal degradation in the RPE can be hindered by oxidative damage to the phagocytosed material due to the photosensitizing properties of retinaldehydes (reviewed in [94,104,105,106,107,108]). Following the photoexcitation of visual pigment, which is the first step leading to visual perception, the Schiff base linkage between the protein and the isomerized chromophore is hydrolysed, resulting in a release of all-*trans*-retinaldehyde from the protein, opsin [109] (Figure 2). The lysosomal degradation of the visual pigments following the RPE phagocytosis of the outer tip of photoreceptors can lead to the release of their intact chromophore, 11-*cis*-retinaldehyde. Overall, both retinaldehydes exhibit similar photosensitizing properties but the absorption spectrum of all-*trans*-retinaldehyde extends more into the visible range than that of 11-*cis*-retinaldehyde. Upon absorption of ultraviolet or blue light, retinaldehyde can form an excited triplet state. In the presence of oxygen, the photoexcited triplet state can interact with oxygen leading to the generation of reactive oxygen species, such as singlet oxygen and superoxide. Both of these reactive oxygen species can lead to lipid peroxidation. The ability of superoxide radicals to initiate lipid peroxidation in the lysosomal environment was discussed in Section 2.2. Singlet oxygen is an electronically excited form of oxygen molecule O_2_, which can oxidize biomolecules, including unsaturated fatty acids, leading to the formation of lipid hydroperoxides. Interaction of the lipid hydroperoxide with iron ion leads to its decomposition and formation of lipid radicals, which can initiate a chain of lipid oxidation. POS contain high concentrations of polyunsaturated fatty acids, such as DHA and arachidonic acid, with six and four unsaturated double bonds, respectively [110]. Therefore, they are very susceptible to lipid peroxidation and formation of numerous oxidation products, some of which are susceptible to fragmentation and/or cyclization, producing a variety of reactive carbonyls [83,111,112,113,114,115,116,117,118,119,120,121]. These reactive carbonyls can form adducts with proteins, which, in the case of bis-carbonyls, lead to the formation of crosslinks. 

It has also been shown that photoexcited all-*trans*-retinaldehyde can inactivate the ATP-binding cassette transporter rim protein (ABCR, also known as ABCA4), which is present in the rims of POS discs [122]. This protein facilitates the enzymatic reduction of all-*trans*-retinaldehyde by transporting it (usually as a Schiff base adduct with phosphatidylethanolamine (PE), N-retinylidene-phosphatidylethanolamine (NRPE)) from the intradiscal leaflet of the lipid bilayer to the cytoplasmic site, where it can serve as a substrate for retinol dehydrogenase 8 (RDH8) and other oxidoreductases [123]. ABCA4 can also transport in the same direction the 11-*cis*-retinaldehyde adduct with PE, thereby preventing its accumulation in the inner leaflet of the disc membrane and possibly facilitating reaching the entrance site of opsin for the regeneration of visual pigment. Moreover, ABCA4 is present in photoreceptor inner segments where it is likely to facilitate the detoxification of both retinaldehydes by making them available to RDHs. In addition to its expression in photoreceptors, ABCA4 is also expressed in the RPE, where it colocalizes with endolysosomal proteins and is thought to aid in the recycling of both retinaldehydes from phagocytosed tips of POS once they are hydrolysed from opsin during proteolysis [124]. Therefore, inactivation of ABCA4 may lead to a decreased clearance of both retinaldehydes. Another oxidoreductase involved in the reduction of all-*trans*-retinaldehyde to all-*trans*-retinol is retinol dehydrogenase 12 (RDH12), which is expressed mainly in photoreceptor inner segments [107,109].

PE is an abundant phospholipid in POS membranes, accounting for about 35% of all lipids [110]. PE reacts with all-*trans*-retinaldehyde forming NRPE with a greater rate than the rate of all-*trans*-retinaldehyde hydrolysis from opsin; therefore, it can reduce the risk of this reactive aldehyde reacting with proteins [83,94,107,110,125]. NRPE can interact with another molecule of retinaldehyde forming a pyridinium bisretinoid, abbreviated as A2PE, as well as other bisretinoids [126,127,128]. These bisretinoids are not degradable by the lysosomal enzymes except for the hydrolysis of the phosphate or fatty acid chains from PE. In the case of A2PE, this leads to the formation of A2E or lysoA2PE (Figure 3). The presence of bisretinoids makes RPE lipofuscin rather unique among lipofuscins from other cells. Several different bisretinoids have been detected in isolated lipofuscin and in the human retina [6,129,130,131,132,133,134].

Considering the release of photoreactive retinaldehydes upon exposure of the retina to light, it is not surprising to see an increased accumulation of RPE lipofuscin in response to rearing in light/dark cycle as opposed to rearing in dark or in response to short-term exposures to light causing damage to photoreceptors (reviewed in [94]). For example, 18-h exposure to 3000–3200 lx cool-white fluorescent light can increase lipofuscin accumulation in the RPE of Japanese quail (*Coturnix Japonica*) [135]. Consistently, a delayed clearance of all-*trans*-retinaldehyde in POS and/or 11-*cis*-retinaldehyde in phagolysosomes in the RPE results in an accelerated accumulation of RPE lipofuscin, which has been observed in the genetically modified mice with deleted genes responsible for synthesis of enzymes involved in clearance of retinaldehydes, such as single gene knockouts: *abca4-/-*, *rdh8-/-*, or *rdh12-/-*; double knockouts: *abca4(-/-)rdh8(-/-)*, *abca4(-/-)rdh12(-/-)*, and *rdh8(-/-) rdh12(-/-)*; and triple knockouts *abca4(-/-)rdh8(-/-)rdh12(-/-)* [124,136,137,138,139,140,141,142,143,144].

#### 4.3.2. Role of Vitamin A Depletion and Inhibition of Synthesis of 11-*cis*-Retinaldehyde in Lipofuscin Formation

Considering the effects of impaired clearance of retinaldehydes, it is not surprising that its precursor, vitamin A (all-*trans*-retinol), and its derivatives play a critical role in the accumulation of lipofuscin in the RPE [145]. Rats fed with a vitamin A-depleted diet accumulate considerably less RPE lipofuscin than rats with normal vitamin A intake, even when they are subjected to increased oxidative stress by depletion of vitamin E, or injection of iron, or their lysosomal degradation is inhibited by leupeptin [95,98,99,102,146]. It has been shown that deficiency in vitamin A leads not only to a decrease of lipofuscin-like fluorescence but also to a decrease in the phagosomal volume in the RPE, which can be ascribed to fewer phagosomes as well as their smaller size [146,147]. Even in the case of *abca4-/-* knockout mice or *abca4(-/-)rdh8(-/-)* double knockout mice, the systemic depletion of vitamin A or pharmacological inhibition of synthesis of 11-*cis*-retinaldehyde results in a decreased accumulation of lipofuscin [148,149,150,151,152,153,154], whereas supplementation with vitamin A increases it [155]. 

#### 4.3.3. Role of Inhibition of Lysosomal Degradation by A2E, Products of Lipid Peroxidation, and Complement Activation in Lipofuscin Formation

As discussed in Section 2, oxidative stress and impaired lysosomal degradation can enhance each other’s effects and accelerate the formation of lipofuscin. In addition to the end products of lipid oxidation, such as MDA or 4HNE, which can affect the susceptibility of POS to lysosomal degradation and inhibit lysosomal enzymes directly, RPE lipofuscin contains A2E, which can inhibit lysosomal enzymes via inhibition of lysosomal ATP-dependent proton pumps [156,157]. Moreover, in case the RPE function is compromised, ameliorating the causes of RPE dysfunction can reduce lipofuscin accumulation. For example, the increased accumulation of lipofuscin in albino *abca4-/-* knockout mice can be partly inhibited by overexpression of the complement regulatory protein, complement receptor 1-like protein y (CRRY) in the RPE [158]. These knockout mice accumulate complement component C3 and/or its cleavage product C3b in the RPE. C3b can trigger various effects, including the formation of a lytic membrane attack complex (MAC) on the plasma membrane, which involves C5 and other complement components. Overexpression of CRRY results in a two-fold decrease in C3/C3b. CRRY prevents the cleavage of complement components C3 and C5, thereby preventing the formation of MAC and its deleterious effects on RPE cell function and viability. To prevent cell lysis, RPE cells can remove MAC by endocytosis and lysosomal digestion. It can be suggested that by preventing MAC formation, phagosomes with tips of outer segments do not need to compete with endocytosed MAC for lysosomes; therefore, the outer segment tips can be digested before substantial amounts of nondigestible crosslinked lipids and proteins form.

#### 4.3.4. The Increased Length of Rod Outer Segments in the Para- and Perifovea May Cause Their Increased Susceptibility to Oxidation and Decreased Susceptibility to Lysosomal Degradation

The length of the rod outer segment varies across the retina, and these variations are similar in the human and monkey retina [159]. The length of rod outer segments and their renewal were studied in the rhesus monkey *Macacca mulatta* and it has been shown that in the parafovea (eccentricity of 0.75–1.25 mm, 2.6–4.3°) and perifovea (eccentricity of 1.75–2.75 mm, 4.3–9.4°), the length of rod outer segments is 35.2 and 31.2 µm, respectively, and gradually decrease towards the periphery, where it is only 23.9 µm. These data of topographical distribution of rods with long outer segments may require some correction when extrapolating to the human eye because *Macaca mulatta* eye is smaller than the adult human eye: the axial length of the 4-year-old monkey is about 18.7 mm whereas the axial length of a normal human eye is about 23–25 mm [160]. There is a possibility that in the human retina, the rods with increased lengths of outer segments are present also in the paramacula where the density of rods is the greatest [161]. The synthesis of new discs occurs at the base of the outer segment, and then rod discs become separated from the plasma membrane so the disc membrane components do not diffuse out of that disc [159]. At the distal tip, the discs are shed, 10–20 at a time, and are phagocytosed by the RPE. It takes about 12.8 and 11.6 days for the disc from the base of the rod outer segment in the parafovea and perifovea, respectively, to reach the distal tip and be shed. In the periphery, the renewal of the rod outer segment takes 8.8 days indicating that shed discs of the outer segment tips in the parafovea and perifovea are 45 and 32% older, respectively, than discs shed in the periphery, and therefore they are likely to contain more oxidized material and increase the oxidative stress at the photoreceptor–RPE interface in comparison with the periphery. 

It can be expected that the oxidative stress induced by blue light can be substantially reduced in the fovea and parafovea where the macular pigment (lutein and zeaxanthin) in the inner retina can absorb blue light [162]. Due to the large inter-individual variations in the optical densities of macular pigment, the absorption of blue light of 460 nm wavelength can vary from 20% to 85% thereby offering different levels of protection to the foveal and parafoveal outer segments from blue light-induced photooxidation [163,164].

### 4.4. Structure and Composition of RPE Lipofuscin

As a result of incomplete lysosomal digestion of POS and intracellular organelles, lipofuscin accumulates in the RPE in the form of compact granules exhibiting Gaussian distribution of sizes, with the mean diameters reported as 0.69 ± 0.63 µm and 0.74 ± 0.24 [6,67]. RPE lipofuscin is an amorphous material of heterogeneous composition with the main components being lipids and extensively modified proteins [6,67,110,165,166,167]. Extraction of lipofuscin with a mixture of chloroform and methanol leads to the extraction of lipophilic material into the chloroform-enriched phase and chloroform-insoluble material at the interphase [94] (Figure 4). The amounts of material in these parts vary with the donors’ age: lipofuscin isolated from 1–40-year-old donors contain 0.09 pg/granule of chloroform-soluble material and 0.08 pg/granule of the interphase material; lipofuscin isolated from 81–98-year-old donors contain 0.10 pg/granule of chloroform-soluble material and 0.14 pg/granule of the interphase material. A2E contributes only about 0.046 fg/granule, which corresponds to 0.02% dry weight, and is a part of the chloroform-soluble lipophilic part of lipofuscin. Oxidative modifications of proteins include adducts with products of lipid oxidation, such as carboxyethylpyrrole (CEP) and iso[4]-levuglandin E_2_, MDA and 4HNE. Moreover, oxidative modifications of proteins in lipofuscin include oxidized methionine to sulphoxide and sulphone, adducts with advanced glycation end products (AGEs), and modifications by peroxynitrite forming nitrotyrosine.

It has been determined that there is 0.77 mmol of lipids per mg of protein in lipofuscin granules isolated from human cadavers below 40 years of age [110]. This ratio decreases with age so lipofuscin from cadavers above the age of 45 years contains only 0.41 mmol lipids per mg protein. It needs to be stressed, however, that the protein was determined by Lowry’s method where the key reagent interacts with tyrosine, tryptophan, and cysteine. In the lipofuscin granule, these amino acids are susceptible to oxidative/nitrative modifications, and therefore it can be expected that as the lipofuscin ages, more of them are modified, which may affect the accuracy of protein determination in lipofuscin from different age groups. The lipid fraction of lipofuscin is composed mainly of free fatty acids accounting for about 40%, followed by phosphatidylcholines accounting for ~30%, phosphatidylethanolamines (~13%), phosphatidylinositol (~7%), phosphatidylserine (~4%) and diacylglycerols (~3%) (Figure 5). Free fatty acids and fatty acids esterified in lipids contain a high proportion of polyunsaturated fatty acids, with arachidonic acid accounting for 16–22% of free fatty acids and 10–13% of fatty acids in total phospholipids. DHA accounts for about 11–15% of free fatty acids, and 7–9% of fatty acids in phospholipids.

Altogether, DHA accounts for 10.2% of total fatty acids in human RPE lipofuscin, which means a 3.2-fold decrease in comparison with the relative content in POS where it accounts for 32.2% [110]. In contrast. the relative content of linoleic acid increases by 4.3- fold. Several potential contributing factors can be responsible for the preferential loss of DHA and the increase in linoleic acid in lipofuscin. One of them may be an enrichment of lipofuscin lipids with lipids from RPE membranes, such as phagosomal and/or lysosomal membranes, which have a relatively lower content of DHA and higher content of linoleic acid than POS [110]. Another reason could be the faster removal of DHA from phagolysosomes than other fatty acids and using that DHA for recycling back to the outer segment and/or for the enzymatic β-oxidation in the RPE to produce ATP [83,168,169,170]. Finally, DHA may be preferentially depleted due to nonenzymatic oxidation.

Oxidation contributes to lipofuscin formation and, once formed, lipofuscin may stay in the RPE for decades. Therefore, it would be highly unlikely for DHA, which is highly susceptible to peroxidation, to remain in lipofuscin in a unoxidized state for long. It can be expected that freshly formed lipofuscin contains less oxidized DHA than lipofuscin aged for months, years, and decades. Lipofuscin isolations include a mixture of lipofuscin of different ages.

DHA has six double bonds separated from each other by a methylene group. A hydrogen on that methylene group is the most susceptible to abstraction by a free radical, thus, initiating a chain of lipid peroxidation where oxygen adds to the newly formed carbon-centred radical of DHA, thereby forming a peroxyl radical. Then, the peroxyl radical can propagate lipid oxidation by abstraction of hydrogen from another methyl group flanked by double bonds [118,171]. Because DHA has five methyl groups flanked by double bonds, it is in proportion more prone to hydrogen abstraction from one of these groups than arachidonic acid, which has only three, or linoleic acid with just one. Moreover, once a chain of lipid peroxidation is initiated in DHA or arachidonic acid, the next propagation step is more likely to occur within the same molecule than in neighbouring fatty acids [118]. Therefore, it can be suggested that DHA may protect linoleic acid from oxidative degradation.

As mentioned before, the lipophilic fraction of lipofuscin contains various bisretinoids and their oxidation products [126,127,128]. A2E is only a minor component of human RPE lipofuscin. It has been determined that A2E accounts for 0.025–0.67% of dry mass of lipofuscin [129,172,173,174]. Interestingly, it has been demonstrated that solubilizing the superficial proteins by treatment with sodium dodecyl sulphate (SDS) or proteinase K removes about 80% of proteins, whereas the content of A2E and isoA2E decreases only by about 14%, and the content of all-*trans*-retinaldehyde dimer-PE decreases by about 7% [6]. This suggests that A2E/isoA2E are strongly anchored within the lipofuscin granule and, therefore, the results observed in experiments in vitro where A2E has been injected from a solution in organic solvent need to be treated with caution.

The bisretinoid content in lipofuscin varies with age across the monkey and human retina, with some bisretinoids, including A2E and its oxidation product monofuran-A2E, being more abundant in the retinal periphery than in the macula in the adult human or monkey eye; however, the opposite is true for the A2E distribution in the young monkey retina [132,133,134,175].

The near-field scanning optical microscopy (NSOM) images of lipofuscin topography show that the lipofuscin surface appears to be composed of small particles 10–30 nm in diameter [176,177]. Due to the high content of polyunsaturated fatty acids, lipofuscin can be easily distinguished from melanosomes by comparing histological sections stained with and without osmium tetroxide and examined with transmission electron microscopy (TEM) [178]. Due to melanin content, melanosome granules appear electron-dense without osmication, whereas lipofuscin granules require osmium tetroxide staining to appear electron-dense. Atomic force microscopy (AFM) investigations of the lipofuscin granules have shown that the surface of the granules is smooth when examined in height images (within 4 nm resolution); however, the phase images show particle-like structures appearing on the surface, about 50 nm in diameter, separated from each other by thin layers [179]. A mixture of chloroform with methanol partly solubilizes the surface of the granules creating 20–40 nm-deep indentations of the same diameter as the observed particles. The mass spectrum of the solubilized material revealed phosphatidylcholines and phosphatidylethanolamines as the main constituents of the extract. A more recent study of lipofuscin granules by AFM confirmed the existence of substructures with sizes of 30–40 nm as well as some larger substructures of 100–120 nm [180].

### 4.5. Distribution of Lipofuscin in the Human RPE

Lipofuscin granules have been detected in the RPE as young as that of 16- and 17-month-old children and their concentration increases rapidly with age in people of White ethnic origin [181,182]. It was suggested that the rapid increase in lipofuscin accumulation in the first two decades of life is due to increased exposure to ultraviolet light transmitted through the lens, which can increase oxidative stress in the retina. By the age of 20 years, there is usually a sufficient build-up of lenticular chromophores to absorb most of the ultraviolet light so it can no longer reach the retina. However, there is great inter-individual variability in the transmission properties of the lens [79]. In some individuals, the transmission window around 320 nm is still present above the age of 60 years.

Initially, lipofuscin granules appear in the basal area of the cells, whereas the melanosomes are present in the apical portion [77,86,181,182]. Later in life, both—melanosomes and lipofuscin become distributed more evenly throughout the cell, and this effect becomes apparent first in the posterior part of the fundus and spreads centrifugally towards the periphery.

The faster accumulation of lipofuscin in people of White ethnic origin than in Blacks has been attributed to the greater optical density of melanin in the iris and choroid of Blacks than of Whites [81,182]. As a result of the greater density of melanin in people with dark pigmentation, less light is transmitted through the iris and the anterior sclera-choroid complex, and less light is transmitted to the posterior sclera. A part of the light that reaches the sclera is reflected back to the choroid where it can be absorbed by melanin preventing it from reaching RPE and photoreceptors.

Weiter et al. have shown that there is a negative correlation between the lipofuscin and melanin contents in the RPE [182]. The lipofuscin and melanin contents have been evaluated in the fovea, perifovea, paramacula, and equator. In the centre of the fovea, where the RPE melanin density is the greatest, the lipofuscin density is about 30% smaller than its maximal density in the perifovea located at half the distance from the fovea to the optic disc. From the area of its maximal density at the perifovea, lipofuscin density decreases towards the equator. In contrast, melanin density decreases from the fovea to the parafovea, remains at a similar level until the paramacula, and then increases towards the equator.

Wing et al. have demonstrated that the greatest accumulation of lipofuscin, quantified by fluorescence, appears in the proximity of the optic disc on the nasal side and at a similar distance from the centre of the fovea on the temporal side [181]. These areas of greatest lipofuscin accumulation are close to the area with the greatest density of rods, which is 3–6 mm from the centre of the fovea [161].

#### 4.5.1. Age-Related Changes in the Topographical Distribution of RPE Lipofuscin in the Human Retina

Feeney-Burns et al. compared numbers of lipofuscin and other pigment granules in RPE cells from donors of different ages [77]. She showed that in the youngest age group of 1–20 year-olds, there are more lipofuscin granules in the macula, 8.8 per section visualized by TEM, than at the equator, 7.9 lipofuscin granules per section, which corresponds to 10% greater numbers of lipofuscin in the macula than at the equator. In contrast, in the age group of 21–60 year-olds, lipofuscin granule numbers are 17.2 and 18.3 granules per profile, respectively, for the macula and equator, which corresponds to 6% greater numbers of lipofuscin at the equator than in the macula. The lipofuscin granule numbers are further increased in the eldest age group, 61–100 year-olds, having 20.8 and 23.3 lipofuscin granules per profile in the macula and equatorial area, respectively, meaning that the equatorial RPE has 12% greater lipofuscin numbers than the macula in this age group. This shows that in the young human retina, macular RPE contains slightly greater numbers of lipofuscin than equatorial RPE but this relationship becomes inversed in the aged retina.

The effect of age on these ratios is much more pronounced when lipofuscin and complex granules called melanolipofuscin are pooled together: in the youngest age group, the number of lipofuscin and melanolipofuscin granules is 1.77-fold greater in the macula than at the equator but this value decreases to 1.14-fold for the middle and the eldest age group [77].

#### 4.5.2. Association of Lipofuscin Accumulation with Retinal Degenerations

RPE lipofuscin accumulates at increased rates in several retinal diseases, including Stargardt’s disease [183], Best’s vitelliform macular dystrophy [184,185], and some types of retinitis pigmentosa [186,187]. It is not clear whether there is an increased accumulation of lipofuscin in age-related macular degeneration (AMD), with some studies providing evidence of decreased fluorescence and concentration of lipofuscin granules in this disease, and an increase in complex granules called melanolipofuscin in comparison with age-matched healthy retinas [188,189].

### 4.6. Effects of RPE Lipofuscin on the Function and Viability of RPE Cells and Photoreceptors

As in other tissues, the presence of lipofuscin in the RPE raises the question of whether lipofuscin can affect the function and viability of the RPE and neighbouring cells, which rely on the RPE for support. It has been demonstrated that there is a correlation between lipofuscin density in the RPE and loss of underlying photoreceptors in people of Caucasian origin but there is no such correlation for people of Black ethnic origin [190]. It has been suggested that there may be a threshold density of lipofuscin, above which the deleterious effects of lipofuscin affect RPE cell function and subsequently, the function and viability of photoreceptors are compromised. Due to the slow accumulation of lipofuscin in Blacks, the threshold may not be achieved, and therefore there is no correlation between the lipofuscin concentration and photoreceptor loss in this population. To date, there are no subsequent studies confirming the findings from this report but there are studies contradicting these findings (reviewed by Curcio [191]). The main argument of Curcio against RPE lipofuscin affecting the viability of photoreceptors is that the age-related loss of rods is not colocalized with the area where the accumulation of lipofuscin (measured by fluorescence) is the greatest. The most pronounced loss of rods occurs in a ring beginning in the outer portion of the fovea and extending from 0.5 to 2 mm from its centre, and is no longer detectable 8 mm from the fovea centre, whereas the greatest accumulation of lipofuscin, measured by its fluorescence, occurs 2–4 mm from the fovea centre.

Like in other cell types, it can be expected that RPE lipofuscin can affect autophagy and contribute to the accumulation of damaged mitochondria. This effect can be exacerbated in the RPE because of the composition of RPE lipofuscin: it has been shown that a component of lipofuscin, A2E exhibits detergent-like properties and disrupts the structure of the lipid membrane causing its permeability [192,193]. Moreover, while A2E does not directly affect the activity of lysosomal hydrolases, A2E can inhibit the vacuolar H^+^-ATPase (v-ATPase) [157,194]. V-ATPase acts as a proton pump, which provides low lysosomal pH needed for activation of lysosomal hydrolases. As a result of A2E effects, the lysosomal degradation of phagocytosed material can be compromised resulting in increased accumulation of phagolysosomes with nondigested contents and/or permeable membranes [17,157]. It can be expected that this can affect the phagosomal and autophagy pathways, and there is a growing body of evidence suggesting that the impairment of autophagy and, resulting from that, accumulation of damaged mitochondria can contribute to the development of AMD [39,195].

#### 4.6.1. Effects of RPE Lipofuscin on Cultured RPE Cells

Testing the potential deleterious effects of RPE lipofuscin on mitochondria, phagocytosis, autophagy, and other cellular processes is facilitated by cell culture models. Several reports demonstrated that lipofuscin accumulation can be induced in cultured RPE cells by supplementing cells with lipofuscin isolated from human RPE cells postmortem or with POS isolated from retinas, usually bovine or rodent [84,196,197,198,199,200,201,202]. Consistent with other studies on lipofuscin accumulation as a result of phagocytosis of oxidized organelles, feeding cultured RPE cells with POS oxidized by exposure to ultraviolet light leads to 50-fold faster lipofuscin accumulation than in cells fed unoxidized POS [198]. This effect is further enhanced by culturing cells in 40% oxygen in comparison to cells cultured under 8% oxygen whereas supplementation of RPE cells with antioxidants: α-tocopherol, lycopene, or lutein and zeaxanthin inhibits lipofuscin accumulation (reviewed in [203]). Also, enrichment of POS with retinaldehyde, 4HNE, MDA, or oxidizable phospholipids, such as PE with polyunsaturated fatty acyl chains or phosphatidylcholine (PC) with monounsaturated fatty acyl chains, leads to an increased accumulation of lipofuscin [204,205,206].

The cell culture model has also enabled numerous studies investigating the effects of various factors, such as lysosomal enzymes, on lipofuscin accumulation. For example, it has been shown that cathepsin D plays a particularly important role in lysosomal degradation, with its monomeric form being more effective in preventing the formation of lipofuscin than its multimeric form [207,208]. The cell culture model of lipofuscin accumulation also allowed for testing various pharmacological approaches to inhibit it. For example, it has been shown that flunarizine and centrophenoxine can decrease lipofuscin formation in cells fed with bovine POS, presumably by decreasing Ca^2+^ overload, which can occur under physiological conditions as a result of exposure to all-*trans*-retinaldehyde [209,210]. Upregulation of autophagy by rapamycin has been shown to prevent the accumulation of lipofuscin in cultured RPE cells fed with POS, whereas inhibitors of autophagy increase lipofuscin formation [36].

Importantly, the RPE cell culture model enables to study of the effects of exposure of lipofuscin-laden cells to light, and therefore, to study them in isolation from the phototoxic effect of retinaldehydes, which can dominate in vivo [94,107,173].

It has been demonstrated in several studies that exposure to visible light of cultured RPE cells with phagocytosed RPE lipofuscin leads to various deleterious effects [6,129,211,212,213]. For example, an exposure for 48 h of such cells in the photosensitizer-free medium to 2.8 mW/cm^2^ blue light (400–500 nm) providing radiant fluence of 484 J/cm^2^ leads to about 50% loss of cell viability [6]. Sublethal exposures of lipofuscin-laden cells to light lead to degradation of lipofuscin, exocytosis of lipofuscin, cytoskeletal changes, decreased phagocytosis of POS, inhibition of antioxidant enzymes (superoxide dismutase and catalase) and lysosomal enzymes (cathepsin D, N-acetyl-β-glucosamidase, acid phosphatase), loss of lysosomal integrity, DNA damage, and enhanced accumulation of lipid peroxidation-derived aldehydes, such as MDA and 4HNE, and damage to mitochondrial and nuclear DNA [75,129,211,212,214,215]. The sublethal exposures used either irradiance of 2.8 mW/cm^2^ and exposure times up to 6 h to avoid cell death [129,214,215] or higher irradiances (9.8–11.4 mW/cm^2^) but with shorter exposure times of 45 min to 3 h [6,75,211,212,213]. Only one of the studies used narrow-band light centered at 490 nm and 3-min exposure to 56 mW/cm^2^ providing fluence of 10.1 J/cm^2^ [213].

The irradiance levels used in these experiments [6,75,129,211,212,213,214,215] are 10- to 100-fold greater than the average irradiance of the human retina in daylight (with no direct projection of the Sun on the retina) [94,216] but much smaller in comparison with the irradiance in the area of the retina, 0.16 mm in diameter, where the image of the midday Sun is focused and provides local irradiance of 1.6 W/cm^2^ for 400–500 nm range [80,217,218]. Therefore, light levels used in experiments on cultured cells described above are of physiological relevance.

The area of the retina with the image of the midday Sun is exposed to irradiance levels, which are 571-fold greater than that causing phototoxicity in lipofuscin-laden cultured cells [6,129]. The duration of the exposure needs to be considered. Due to eye movements and aversion response, the image of the Sun is focused on the same spot of the retina for only about 0.25 s [216]. Within that time the radiant fluence of 400–500 nm light delivered to that part of the retina is 0.4 J/cm^2^. To deliver the dose of 484 J/cm^2^, which was found to kill about 66% and 50% of lipofuscin-laden primary RPE cells and ARPE-19 cells, respectively, after 48-h exposure using irradiance of 2.8 mW/cm^2^ [6,129], the image of the Sun would need to be kept in the same area for 5 min. It has been shown in experiments on monkeys that the reciprocity holds for light-induced injury to the monkey retina by exposures to UV and short-wavelength visible light (blue and green) so a 100 s exposure causes the same injury as 1000 s exposure of 10-fold smaller irradiance [94,219]. It is thought that the reciprocity holds only for relatively short exposure times when the detoxification and repair mechanisms have not substantially counteracted the damage [216]. This may not be the case for 48-h-long exposure. Nevertheless, when reciprocity holds, the Sun will need to be focused on the same spot of the retina 1200 times, a quarter of a second each time, to provide the same fluence as that in the phototoxic exposure of cultured cells with lipofuscin.

#### 4.6.2. Susceptibility to Autooxidation and Photosensitizing Properties of RPE Lipofuscin

It has been shown that isolated lipofuscin is susceptible to light-induced autooxidation: the exposure of a suspension of lipofuscin granules to narrow-band light results in oxygen consumption, the rates of which, when normalized to equal fluxes of incidents photons, monotonically increase with decreasing wavelength within the range of 600 to 280 nm [220]. Some of the consumed oxygen is reduced to superoxide radical, which then dismutates to hydrogen peroxide [220,221]. It has been determined that hydrogen peroxide accounts for only about 1% of oxygen consumed during the exposure of lipofuscin to blue light, indicating that the majority of oxygen is used for the oxidation of intragranular components [220]. As mentioned before, lipofuscin originates from POS with high concentrations of polyunsaturated fatty acids, and therefore they are likely to be the primary target of oxidation. Indeed, it has been shown that exposure of lipofuscin to light leads to the generation of lipid hydroperoxides and end products of lipid peroxidation, such as MDA. It has been also shown that light exposure of lipofuscin in the presence of proteins or lipid vesicles with incorporated unsaturated fatty acids or cholesterol results in increased oxygen consumption, indicating that oxidation is not confined to the lipofuscin granule, but reactive oxygen species can diffuse outside the granule and damage extra-granular molecules [220,221,222]. In the case of cholesterol, different hydroperoxides are formed, including 5α-cholesterol hydroperoxide, which is a specific product of cholesterol interaction with singlet oxygen [220].

Chloroform-methanol extraction of lipofuscin allows for the separation of the chloroform-soluble lipophilic fraction and chloroform-insoluble material [172]. Both fractions exhibit substantial photoreactivity (Figure 6).

The lipophilic extract of lipofuscin exhibits a broad absorption spectrum monotonically increasing with decreasing wavelength from 640 to 280 nm [172]. This mixture of numerous components includes potent photosensitizer(s), which, upon photoexcitation, form an excited triplet state with the ability to efficiently transfer the excitation energy to oxygen leading to the formation of singlet oxygen [223,224]. The quantum yields of singlet oxygen generation by photoexcited lipophilic extract of lipofuscin depend on the excitation wavelength: excitation with ultraviolet light of 355 nm wavelength or blue light from the range of 420 to 440 nm results in about 8 and 5% of absorbed photons utilized for singlet oxygen production, respectively. Superoxide is a minor product generated by the photoexcited lipophilic extract of lipofuscin. The quantum yield of blue light-induced generation of superoxide is only ~0.1%, which is about 50 times smaller than that of singlet oxygen [225].

#### 4.6.3. Negligible Contribution of A2E to the Photosensitizing Properties of Lipofuscin

One of the lipophilic components of lipofuscin is A2E [127,226]. Based on the quantification of A2E in lipofuscin granules and the absorption spectra of A2E and lipophilic extract of lipofuscin, it has been estimated that A2E provides only 0.8% contribution to the absorption of visible light by the lipophilic fraction of lipofuscin [107,129]. Considering A2E contribution to the absorption spectrum of lipofuscin extract and its very weak photosensitizing properties, it has been evaluated that (i) A2E contributes at most one singlet oxygen molecule per 300 singlet oxygen molecules generated by lipofuscin; and (ii) A2E contributes at most one superoxide molecule per 384 superoxide molecules generated by lipofuscin [173].

#### 4.6.4. Lack of Evidence of the Deleterious Effect of A2E to Cultured Cells While Incorporated into Lipofuscin

A2E has attracted a great deal of attention due to the simple way it can be synthesized so its effects on RPE cells can be studied in vitro [226]. Most of the studies on cultured cells used A2E solubilized in dimethylsulphoxide (DMSO), and added to the culture medium, so A2E was free to diffuse to various organelles (reviewed in [94,107,173]). Under physiological conditions, A2E is present in the lipofuscin granule, and it appears to be strongly anchored within that granule, so even treatment with SDS, which leads to the removal of most identifiable proteins from the granule, does not affect the concentration of A2E remaining in the granule [6]. Therefore, it is questionable whether some of the deleterious effects of A2E on mitochondria, DNA, or transport proteins, observed when A2E is delivered in solution, are relevant to the situation in vivo where A2E is encapsulated within the lipofuscin granule.

A2E is susceptible to oxidation and, as a result, it forms various oxidation products, which include epoxides, furanoid oxides, cyclic peroxides, and carbonyls [130,131,227,228,229,230,231,232,233,234]. Several of these products have been identified in the RPE from human cadavers [132,134,175,235]. Studies on cultured RPE cells, where solubilized oxidation products of A2E were delivered to cells in solution, or where A2E-laden RPE cells were exposed to light to cause its oxidation in situ, have demonstrated several detrimental effects of A2E oxidation products, including DNA damage, induction of pro-angiogenic factors, activation of the complement cascade and other pro-inflammatory pathways (reviewed in [127,236,237]). It remains to be shown whether these A2E oxidation products can stimulate these deleterious effects in vivo or whether they are safely trapped in the lipofuscin granule. It has been reported that oxidized A2E can react with other oxidized A2E molecules or A2E itself forming high-molecular weight products, which are more hydrophobic than A2E, and therefore more likely to remain in the granule [238,239].

#### 4.6.5. Potential Role of Oxidized DHA in Photosensitizing Properties of Lipofuscin

Interestingly, oxidation products of DHA, an abundant component of POS membranes and lipofuscin, exhibit potent photosensitizing properties upon exposure to ultraviolet or blue light [240]. The absorption spectrum of a mixture of products of DHA oxidation exhibits an increasing absorption with decreasing wavelength in a range of 280–600 nm (Figure 7A). Photoexcitation of oxidized DHA with 355 nm or blue light leads to the formation of a triplet state similar to the triplet state of lipophilic extract of lipofuscin. The triplet state is quenched by oxygen, which leads to the photosensitized generation of singlet oxygen. Exposure of oxidized DHA to visible light leads to the photosensitized generation of superoxide. The quantum yields of generation of singlet oxygen and superoxide are 2.4- and 3.6-fold greater, respectively than those of the lipophilic extract of lipofuscin. This is consistent with lipofuscin containing a mixture of chromophores contributing to the absorption of light, some of which, such as A2E, have very weak photosensitizing properties, whereas others, such as oxidized DHA are very potent photosensitizers. The action spectrum of photooxidation of oxidized DHA exhibits a monotonic increase with decreasing irradiation wavelength (Figure 7B).

#### 4.6.6. Neglected Components of Lipofuscin Exhibiting High Photoreactivity

Chloroform-insoluble components of lipofuscin also exhibit the ability to photosensitize the generation of singlet oxygen, superoxide, and oxidation of exogenous lipids and proteins [172]. Interestingly, when studied at the same concentration of dry mass, both soluble and insoluble fractions of lipofuscin demonstrate no age-related changes in photoreactivity, even though lipofuscin granules become more photoreactive with age. What changes with age is the ratio of insoluble to soluble components per lipofuscin granule, which increases with age (Figure 4). Therefore, the age-related increase in the photoreactivity of lipofuscin granules can be attributed to the increase in the insoluble part (Figure 6). However, to date, no successful attempts have been reported to identify the visible light-absorbing chromophores of the insoluble part, which could be responsible for the observed photoreactivity.

#### 4.6.7. Circumstantial Pieces of Evidence Suggesting That Lipofuscin Contributes to Retinal Phototoxicity In Vivo

As discussed above, numerous studies in vitro point to the potential phototoxic effects of lipofuscin in the RPE in vivo, which is exposed daily to light in the presence of high oxygen tensions of about 70 mm Hg [241], thus providing ideal conditions for the photoexcitation of photosensitizers of lipofuscin, and subsequent formation of reactive oxygen species, which can cause oxidation of the lipofuscin components and damage cellular proteins, lipids and nucleic acids. Without a doubt, these experiments in vitro add valuable information on the mechanisms that could be involved in the effects of lipofuscin. While it is clear that lipofuscin exhibits photosensitizing properties and can affect cell function and viability in vitro upon exposure to light, the evidence of the contribution of lipofuscin to light-induced retinal injury in vivo is rather limited. The main difficulty with the interpretation of the results of in vivo studies is in distinguishing whether light-induced injury to the retina is caused by lipofuscin or by retinaldehydes. Nevertheless, the studies in vivo with the potential involvement of lipofuscin are discussed to show mainly the circumstantial evidence that is available.

One of the circumstantial pieces of evidence pointing to lipofuscin contribution in light-induced injury is the age-related increase in susceptibility to retinal photodamage in rats reared under dim cyclic light as opposed to rats reared in dark [242]. It can be suggested that rats reared under dim cyclic light may have an increased lipofuscin content in the RPE but this was not evaluated in the study.

It has been shown in numerous studies that mice with abnormal trafficking of retinaldehydes and accelerated lipofuscin accumulation, such as *abca4(-/-)rdh8(-/-)* double knockout, are very susceptible to light-induced retinal injury [139,140,149,150,210,243,244,245,246,247,248,249,250,251,252,253,254,255,256,257,258,259,260,261,262]. In most of these studies, 30–60 min exposures of *abca4(-/-)rdh8(-/-)* double knockout mice with dilated pupils to light from a desk lamp equipped in a fluorescent bulb led to a substantial photoreceptors loss [139,140,149,150,210,245,246,247,248,249,250,251,252,253,254,255,256,257,258,259,261]. This loss of photoreceptors was effectively prevented by inhibitors of the synthesis of 11-*cis*-retinaldehyde and/or scavenging of retinaldehydes by amine compounds such as retinylamine. Therefore, these studies provided solid evidence of retinaldehydes being responsible for light-induced retinal injury but have not provided evidence of the lipofuscin contribution to that injury.

It has been recognized very early in the studies on light-induced injury to the retina that RPE is the primary target for the threshold injury induced by light of 441 nm and longer wavelengths applied to the macaque retina [217]. More recent studies on macaques demonstrated that exposure of the retina to 460, 488, 544, 568, and 594 nm laser light can lead to bleaching of RPE fluorescence and, at higher retinal radiant exposures, varying from about 30 to 300 J/cm^2^ for the shortest and longest wavelengths, respectively, to the disruption of the RPE mosaic, suggesting that photoreactivity of lipofuscin could contribute to the RPE injury [263,264,265,266]. Interestingly, these studies showed that light-induced retinal injury occurs at much lower radiant exposure thresholds for the cyan, green, yellow, and orange lasers than expected based on previous reports by Ham and colleagues [219]. A possible explanation for the discrepancy in the observed radiant exposure thresholds for injury could be the different concentrations of RPE lipofuscin, making macaques with more lipofuscin more susceptible to light injury; however, lipofuscin was not quantified in any of these studies.

To date, two reports suggest that RPE cells can be the primary target of light-induced injury in *abca4-/-* knockout mice with an increased accumulation of lipofuscin [260,262]. One of them demonstrated that a 15-min exposure to blue light (430 nm wavelength, 50 mW/cm^2^ irradiance) of pigmented wild-type and *abca4-/-* knockout mice results in a loss of some photoreceptors and no significant loss of RPE cells in the wild-type mice, whereas the knockout mice exhibit a massive loss of RPE cells in the central retina with the loss of photoreceptors not greater than that in the wild-type mice [260]. It was interpreted that the damage to the RPE in the knockout mice was caused by lipofuscin. While this can be true, it cannot be excluded that the abnormal trafficking of all-*trans*-retinaldehyde due to the absence of ABCR protein in the POS and RPE, affects RPE cells more than photoreceptors. Such a possibility is supported by experiments of Wu et al. on susceptibility to light damage of albino wild-type and *abca4-/-* knockout mice of different ages [267]. They have shown that 8-month-old *abca4+/+* mice have similar concentrations of A2E isomers as 2-month-old *abca4-/-* mice but are much less susceptible to RPE cell loss induced by exposure of the retina to 430 nm light than the young knockout mice. However, lipofuscin has not been quantified in this study so it is not clear whether its levels were the same in both types of mice.

A follow-up study by Fang et al. provided another circumstantial piece of evidence by demonstrating that upon pharmacological removal of RPE lipofuscin from 12-month-old *abca4-/-* mice, the exposure to blue light is less damaging [262]. The mice were pre-treated with soraprazan, a drug developed to treat gastroesophageal reflux, which, due to its ability to decrease the lipofuscin content, was renamed Remofuscin. Remofuscin was injected intravitreally and, after 28 days, led to decreased levels of lipofuscin and melanosomes in the RPE when quantified by fluorescence and transmission electron microscopy. Exposure to blue light was also performed 28 days after Remofuscin injection. Mice pre-treated with Remofuscin retained about twice more RPE and photoreceptor nuclei than those treated with DMSO used as a vehicle or without pre-treatment. Still, some doubts remain regarding the role of lipofuscin in mediating light-induced injury because Remofuscin has been shown to generate reactive oxygen species, such as superoxide, upon exposure to blue light. Therefore, it can be suggested that mice pre-treated with Remofuscin may have upregulated antioxidant and detoxification defences, making them more resistant to light-induced injury than mice pre-treated with the vehicle.

Teussing et al. have demonstrated in a small clinical study on five Stargardt’s disease patients with identified disease-related mutations in *ABCA4* gene that protecting one eye from light by wearing for at least 11 months a black contact lens during waking hours results in a smaller percentage of hypofluorescent pixels in the fundus fluorescence image, corresponding to focal losses of the RPE, in comparison with the unprotected fellow eye [268]. Like in studies on rodents described above, it is not clear whether the protective effect was due to preventing the deleterious effects of retinaldehydes and/or lipofuscin.

#### 4.6.8. Circumstantial Pieces of Evidence Suggesting That Lipofuscin Contributes to Retinal Degeneration In Vivo in Dark-Reared abca4(-/-)rdh8(-/-) Double Knockout Mice

Another study on the potentially deleterious effects of lipofuscin accumulation has been performed on *abca4(-/-)rdh8(-/-)* double-knockout mice reared in the dark [17]. Rearing in the dark slows down the light-induced retinal degeneration in these double knockout mice while still enabling the massive accumulation of lipofuscin in the RPE and preventing its photodegradation, as opposed to rearing under low-intensity cyclic light (below 10 lx), which can cause a total loss of photoreceptors and RPE degeneration by the age of 6 months as shown by Maeda et al. [17,140]. Interestingly, *abca4(-/-)rdh8(-/-)* double knockout mice used by Pan et al. have demonstrated similar loss of photoreceptors and RPE nuclei at 12 months of age independently whether the mice were housed in dark or in cyclic light of about 10 lx [17]. At the old age of 26 months, the thickness of the layer with photoreceptor nuclei in these double knockout mice decreases by about 40% in comparison with the wild-type mice, whereas the number of RPE nuclei decreases by about 30%. The remaining RPE cells in the 24-month-old *abca4(-/-)rdh8(-/-)* mice are significantly larger and with more nuclei per cell than in wild-type mice and, unlike healthy-looking wild-type RPE cells, exhibit stress fibres.

The fluorescence of lipofuscin in 3 months old *abca4(-/-)rdh8(-/-)* mice is significantly greater than in 33-month-old wild-type mice [17]. At 12 months, the double-knockout mice reared in the dark have about five-fold greater fluorescence than their counterparts raised in cyclic light. At 20 months, the dark-reared double knockout mice exhibit massive accumulation of RPE lipofuscin, and, in some RPE cells, anti-galectin-3 positive staining, which, in some small areas, colocalizes with staining for the lysosomal marker Lamp1, suggesting loss of lysosomal membrane integrity [17,269].

Interestingly, the flat-mounts of the *abca4(-/-)rdh8(-/-)* eyecups with removed neural retinas exhibit anti–phospho-Ser358 MLKL staining in RPE already at 2 months, and that staining density strongly increases with age, whereas, in the wild type RPE, it is virtually undetectable even at 27 months [17]. MLKL stands for mixed lineage kinase domain-like protein, which is a protein essential for triggering necroptosis and is also involved in facilitating endosomal trafficking and generation of extracellular vesicles [270]. In 12- and 26-month-old *abca4(-/-)rdh8(-/-)* mice, the anti–phospho-Ser358 MLKL staining is present not only in the cytoplasmic areas but also in the plasma membrane [17]. Retinal cross-sections from 20-month-old double knockouts exhibit the anti–phospho-Ser358 MLKL staining not only in the RPE layer but also in retinal layers occupied by photoreceptors: the outer and inner segment layers and outer nuclear layer, where the staining appears punctate. Interestingly, the anti–phospho-Ser358 MLKL staining in 26-month-old RPE can be completely prevented by a single intravitreal injection of necroptosis inhibitor, necrostatin 7 (Nec7) a week prior to the dissection. However, injecting another necroptosis inhibitor, namely necrostatin-1 (Nec1; an inhibitor of receptor-interacting serine/threonine-protein kinase 1 (RIPK1)), or a vehicle has no effect. Nec7 is not an inhibitor of RIPK1 and its mechanism of action as a necroptosis inhibitor is unclear. Nevertheless, it also protects from RPE cell loss in these double knockout mice when injected once a month for 6 months starting from the age of 7 months.

The double knockout *abca4(-/-)rdh8(-/-)* neural retinas include large clusters of lipofuscin and melanosomes enclosed in structures of about 5–10 µm in diameter [17]. The photoreceptor layer in proximity to such structures appears more disorganized with fewer nuclei than elsewhere, and the remaining nuclei stain positive for terminal deoxynucleotidyl transferase dUTP nick end labelling (TUNEL), suggesting that these photoreceptors are undergoing an apoptotic type of cell death. Lipofuscin-containing debris outside RPE detected in these double knockout mice vary in diameter from about 1–20 µm to much larger structures forming subretinal deposits, which usually are surrounded by anti–phospho-Ser358 MLKL staining, suggesting that phosphorylated MLKL may promote shedding parts of photoreceptors and RPE as a way to minimize necroptotic cell death.

The anti–phospho-Ser358 MLKL staining has been also detected in the human retinas from 74-, 80- and 86-year-old donors affected by dry AMD, but not in the normal retina from a 42-year-old donor [17]. The staining in AMD retinas spreads from the RPE to the outer nuclear layer and appears either punctate or covers larger areas, some of which co-stain for Iba1, a marker of microglia. It is stated by the authors that there is a correlation between the accumulated lipofuscin and staining of anti–phospho-Ser358 MLKL and Iba1, but no data supporting that statement are provided. The authors suggest that lipofuscin induces the permeability of the lysosomal membrane and subsequent leakage of lysosomal enzymes, which triggers the formation of atypical necrosome, not dependent on RIPK1 nor RIPK3, which can phosphorylate MLKL. Phosphorylation of MLKL enables its oligomerization and insertion into the lysosomal or plasma membranes forming pores, which lead to membrane permeability, with subsequent shedding of cell fragments or cell death by atypical necroptosis. However, another possibility worth considering is that the shedding of lipofuscin-containing cell fragments promoted by phosphorylated MLKL is an independent process from cell death and that the latter could be caused by the reactivity of retinaldehydes. Such a possibility could be tested by treating the double knockout mice with the RPE65 inhibitors from the age when substantial amounts of lipofuscin are present but with no substantial loss of RPE or photoreceptor cells.

#### 4.6.9. Protective Effect of Deuterated Vitamin A on A2E and Lipofuscin Accumulation, Complement Activation and Retinal Degeneration in Mice, and on Slowing Down Geographic Atrophy Progression in Stargardt’s Disease Patients

Another study on a*bca4(-/-)rdh8(-/-)* double knockout mice, which provides another piece of evidence of the deleterious effect of lipofuscin, used supplementation with a form of vitamin A, where a hydrogen atom at carbon 20 is substituted by deuterium to prevent the formation of bisretinoids [271]. This strategy leads to a substantial decrease in lipofuscin fluorescence, which in 3-month-old mice is about 75% less intense than in the retinas of mice fed with normal vitamin A. The effect is similar for A2E, isoA2E, and their oxidation products, the accumulation of which was quantified between 1 and 18 months and has been shown to be decreased by about 75% in comparison with mice fed normal vitamin A. Interestingly, in both treatment groups, the bisretinoids increased up to the age of 8 months and then plateaued. There were no differences at the ages of 7 and 18 months between the normal and deuterated vitamin A-supplemented mice in electroretinogram amplitudes of a- and b-waves, which reflect functions of photoreceptors and bipolar cells, respectively. The age-related decrease in a- and b-waves was similar in both groups. Supplementation with deuterated vitamin A was associated with improved dark-adaptation in 12-month-old mice in comparison with mice supplemented with normal vitamin A, where 30 min after photobleaching, mice recovered 71% and 53% of the fully dark-adapted b-wave amplitude for deuterated and normal vitamin A, respectively. It has been stated by the authors that the deuterated vitamin A partly prevented age-related loss of retinal thickness measured at 12 and 18 months when compared with retinal thickness at 3 months. The retinal thickness in mice supplemented with deuterated vitamin A decreased only by 5% and 3% at 12 and 18 months, respectively. The retinal thickness in mice on normal vitamin A decreased by 17% and 15% at 12 and 18 months, respectively. The thickness was measured 1 mm from the optic disc, but it is not specified whether or how the eyes were oriented before enucleation, so it is not clear whether these measurements were taken from the same location in each retina. This is important because there is a different susceptibility to photoreceptor loss in the superior and inferior retina of rodents. Interestingly, there were no statistically significant differences in the thickness of the outer nuclear layer between the treatment groups. The outer nuclear layer is made of cell bodies of photoreceptors and its thickness can reflect their density. Importantly, deuterated vitamin A completely prevented focal degenerative changes starting to appear at 12 months in double knockout mice supplemented with normal vitamin A, where RPE cell loss or hypertrophy, and the appearance of pigmented cells in the photoreceptor layers were detectable in every eye examined from twelve 18-month-old mice, suggesting that lipofuscin bisretinoids can lead to deleterious effects in the retina in vivo.

The beneficial effects of deuterated vitamin A were also shown in experiments on a*bca4(-/-)* single-knockout mice [272]. In comparison with 9-month-old animals supplemented with normal vitamin A, age-matched animals supplemented with deuterated vitamin A had about a 50% decrease in both the cytoplasmic volumes occupied by lipofuscin granules in the RPE and the intensities of fundus fluorescence excited by 488 nm laser, as well as a five-fold decreased level of A2E. The eyecups from knockout mice supplemented with normal vitamin A had 2.2-fold increased mRNA levels of complement component C3 and 2.0- and 2.3-fold decreased mRNA levels of complement factor B and complement factor properdin in comparison with the wild-type mice on the same supplement. Importantly, supplementation of the *abca4-/-* mice with deuterated vitamin A resulted in the same levels of mRNA for all three complement factors as those in the eyecups from the wild-type mice supplemented with normal vitamin A. These findings are significant considering the associations of complement proteins with AMD [273,274].

The most convincing piece of evidence of the deleterious effects of A2E and lipofuscin comes from the initial results of a multicentre clinical trial testing the effects of deuterated vitamin A on the progression of geographic atrophy in Stargardt’s disease patients [275]. This double-masked, placebo-controlled trial randomized 50 patients with Stargardt’s disease caused by the *ABCA4* mutation into the treatment and placebo groups at a 2:1 ratio. The treatment group received pills with deuterated vitamin A, C20-D3-retinyl acetate (known as ALK-001). Supplementation in the treated group led to about 90% vitamin A in its deuterated form, suggesting that the dose of dietary vitamin was low in comparison with that from ALK-001. The growth rate of the square root of atrophic lesions in the treated group was 21% slower than in the placebo group (*p* < 0.001).

#### 4.6.10. Is There an Association between Light Exposure and the Development or Progression of AMD?

Due to the photosensitizing properties of lipofuscin, it is reasonable to expect that age-related accumulation of lipofuscin together with life-long exposure to light can have detrimental effects and increase the risk of development of AMD. In the natural environment, sunlight is the strongest light source that can be focused on the retina [216], therefore several epidemiological studies have attempted to determine if there is an association between AMD and exposure to sunlight or specific parts of the solar spectrum.

The meta-analysis of 14 studies published up to December 2017 performed by Zhou et al. led to a pooled odds ratio for sunlight exposure and AMD of 1.10 (95% CI = 0.98–1.23) and no significant association between sun-avoidance behaviour and AMD (OR = 1.12, 95% CI = 0.76–1.67) [276]. The analysis did not attempt to compare the values of sunlight exposure in different studies. Most of the studies included in that meta-analysis did not ask participants about their sun avoidance behaviours [277,278,279,280,281,282,283,284,285], so it is not clear how Zhou and colleagues obtained these data for all 14 studies to perform the meta-analysis for the association between AMD and sun avoidance, given that such data were included in only six of them [286,287,288,289,290,291]. Another meta-analysis of pooled 14 studies published up to March 2012 has shown a statistically significant relationship between sunlight exposure and AMD (OR = 1.379; 95% CI: 1.091 to 1.745) [292]. The inclusion and exclusion criteria of these two meta-analyses were different, contributing to the difference in the selection of the studies used for the meta-analyses.

Altogether, there are more studies showing a statistically significant positive association of early and/or late AMD with sunlight exposure [277,278,280,281,282,283,284,285,289,290,293,294,295,296,297], than studies finding no association or negative association [279,286,287,288,291,298,299,300,301]. Importantly, how the exposure to sunlight was evaluated needs to be considered. Some of the studies attempted to evaluate the ocular exposure to sunlight by a more or less detailed structured questionnaire asking about the percentage of time wearing sunglasses outdoors, indoor/outdoor places or work, summer leisure time outdoors, and winter leisure time outdoors [282,283,284,285,290,291,296,297,300,301]. Other studies relied on a brief questionnaire and evaluated sunlight exposure based on working/living in a sunny climate for 5 years or longer, treatment for skin tumours, the mean sun exposure index, and sun avoidance with no attempt to evaluate the ocular exposure to sunlight [288], or just by two questions about living in a sunny area and using “usual sun protection”, without specifying the period in life [286]. In other studies, sunlight exposure was evaluated by a short questionnaire as more or fewer than a certain number of hours per day of outside activities during working life, which could vary from less than 2 h to more than 8 h, and include, or not, an option of sun avoidance [277,278,279,280,281,287,293,295,296].

While some studies evaluated sunlight exposure based on both the questionnaire and meteorological data for UV exposure [283,284,285,295,299], there is a study where such meteorological data together with the locations where a person lived up to the age of 65 years were the only source of data with no attempt to evaluate the time spent outdoors or use of sun protection for the eyes [298].

Only one study attempted to evaluate the ocular exposure to sunlight by another method than a questionnaire [294]. Hirakawa et al. assessed sunlight exposure by measurement of the facial hyperpigmentation and total length of facial wrinkles per surface area of the upper cheek and temporal areas next to the eyes in 67 controls, 75 patients with early AMD, and 73 patients with late AMD (all neovascular). All patients were male farmers with similar lifestyles with life-long residence in the rural area of Kagoshima, Japan. They found a significant positive association between late AMD and facial wrinkles (*p* = 0.047, OR = 3.8; 95% CI: 1.01–13.97), which correspond to cumulative exposure to UV.

Some studies analysed the results taking into account age and gender [286] or more confounding factors, such as education level, occupational classification, household income, alcohol consumption, current smoking status, physical activity, obesity, hypertension, and diabetes mellitus [293], or smoking and genetic susceptibility variants [299]. After adjusting for the confounding factors, in some cases, the association of sunlight exposure and AMD has remained statistically significant [293,299], while in other cases it has not [286]. Another important factor affecting the susceptibility to light-induced retinal injury are adaptive changes occurring in the retina in response to ambient levels of light it is exposed to. Several experiments on animals have shown that long-term adaptation to environmental light modulates the susceptibility to light-induced retinal injury [242,302,303,304,305], which at least in part is due to lower rhodopsin concentration and higher antioxidant activities in retinas of animals raised at higher light intensities in comparison with animals raised at low levels of light [303,306,307,308,309,310,311].

When considering the effect of UV light on the retina, it is worth considering it together with the effects of UV on the transmission properties of the lens and synthesis of vitamin D. Increased exposure to UV can be expected to accelerate the synthesis of lenticular chromophores, which then can act as a sunscreen for the retina by absorbing UV and later also blue light [312,313,314,315]. Evaluation of the absorption properties of the lens was not considered in any of the epidemiological studies of the association between sunlight exposure and AMD.

Exposure of 7-dehydrocholesterol in the skin to UV-B results in the formation of cholecalciferol, which then can be used for enzymatic synthesis of calcitriol, known as vitamin D [286,316,317]. Out of the multiple roles of vitamin D, those that are relevant to AMD include its immunoregulatory, anti-inflammatory, and antiangiogenic properties. It has been shown in some studies that low serum levels of vitamin D, especially if they indicate vitamin D deficiency, are associated with an increased risk of AMD, and that there is a positive correlation between the lower dietary intake of vitamin D and severity of AMD [300,318,319,320,321,322,323,324,325]. One study of postmenopausal women in the Carotenoids in Age-Related Eye Disease Study (CAREDS) found a positive association between serum levels of vitamin D and average ocular sun exposure in the past 20 years, and a negative association between risk of early AMD and both serum levels of vitamin D and vitamin D dietary intake, but found no association between the risk of AMD and exposure to sunlight [300].

Interestingly, the European Eye Study EUREYE found no overall association between sunlight exposure and AMD but found a significant association between exposure to blue light and neovascular AMD in participants in the lowest quartile of blood levels of antioxidants vitamin C, vitamin E, and zeaxanthin [283]. The study used a questionnaire asking about time spent outdoors throughout a participant’s working life and in retirement up to the time of the eye examination. Information was collected on the use of hats and eyewear (glasses, contact lenses, and sunglasses) and about the terrain in which outdoor exposures took place. The geographical coordinates of residence were used to generate estimates of individual years of exposure to different wavelengths of light.

There are some studies that did not attempt to evaluate sunlight exposure but investigated the association between AMD and absorption properties of intraocular lenses (IOLs) implanted during cataract surgery [326,327,328,329,330]. The largest study was performed in Taiwan and involved an impressive number of 186,591 patients older than 50 years who underwent bilateral cataract surgery and implantation into both eyes of either colourless IOLs (165,465 patients) or blue-light-absorbing IOLs (21,126 patients) [326]. The patients were monitored until AMD diagnosis, death, loss to follow-up, or the end-of-study date, so patients were followed for up to 10 years, with an average of 6.1 years. The incidence rate of AMD was calculated as the number of AMD cases, categorized as nonexudative and exudative, after cataract surgery, divided by the person-years involved. AMD developed in 13,586 patients, with the nonexudative, exudative, and both forms in the same patient being 12,533, 1655, and 602, respectively. The AMD rate per 1000 person-years was 10.99 for nonexudative AMD and 1.42 for exudative AMD. A total of 1513 and 12,675 patients developed AMD in the blue-light-absorbing IOL and colourless IOL groups, respectively. The AMD rate per 1000 person-years was significantly smaller in the blue-light-absorbing IOL group (nonexudative: 9.95; exudative: 1.22) than in the colourless-IOL group (nonexudative: 11.13; exudative AMD: 1.44) with *p* < 0.0001 and *p* = 0.0097 for nonexudative and exudative AMD, respectively. However, the baseline characteristics of patients with UV-blocking and blue-light-absorbing IOLs were different, and after adjusting for age, gender, socioeconomic status, and comorbidities, the differences became nonsignificant. The blue-light-absorbing IOLs were considered premium IOLs, which were not reimbursed by the Taiwan National Health Insurance. Therefore, there is a possibility that patients who had colourless IOLs were protecting their eyes more from excessive sunlight exposure than patients with blue-light-absorbing IOLs; however, such data were not collected. It is also possible that patients with blue-light-absorbing IOLs overestimated the protection offered by their IOLs and did not use any additional protection for their eyes. There are different types of blue-light-absorbing IOLs with different absorption characteristics, but this was not considered either.

No consideration was given to wearing sunlight protection for the eyes and spectral characteristics of the implanted blue-light-absorbing IOLs in other studies as well except for one where the type of the blue-light-absorbing IOL was given.

In a Finnish study, where 11,397 patients were followed after cataract surgery involving implantation of blue-light-absorbing IOLs in 47.6% of eyes and colourless IOLs in the rest, no significant differences were found between the effects of blue-light-absorbing IOLs and colourless IOLs on the development of neovascular AMD and, in case of pre-existing neovascular AMD, on various markers of its progression (including numbers of anti-VEGF injections and treatment interval) [327].

Another study investigating if blue-light-absorbing IOLs have a protective effect on AMD, in comparison with clear IOLs, was performed in Canada and involved 196 patients with wet AMD and an equal number of control patients, who were matched with respect to age at the time of cataract surgery and gender [328]. The proportions of blue-light-absorbing IOLs in the AMD and control groups were similar: 62.8% and 63.3%, respectively. The AMD group had higher blood pressure, aspirin use, hyperlipidemia, and a higher ratio of positive smoking history. After adjusting for these variables, blue-light-absorbing IOLs had no significant effect on wet AMD. Interestingly, the mean time between the cataract surgery and the first anti-VEGF injection was significantly shorter in the blue-light-absorbing IOLs group than in the colourless IOLs group: 5.76 years and 6.62 years, respectively.

There are two relatively small studies showing a protective effect of blue-light-absorbing IOLs on AMD [329,330]. One of these studies compared the development of abnormal fundus fluorescence and/or fluorescence changes, drusen, the development of late AMD (neovascular or geographic atrophy) in patients with implanted blue-light-absorbing or only UV-absorbing IOLs and examined immediately after and then again two years after the surgery [330]. The analysis of fundus fluorescence was performed on images from 52 eyes with a blue-light-absorbing IOL and 79 eyes with a colourless IOL. The two groups of patients did not differ significantly in terms of age, sex, smoking, diabetes, hypertension, or supplement use. Abnormal fundus fluorescence developed or progressed in 12 eyes (15.2%) in the colourless IOL group (mostly as reticular) and in none of the eyes in the blue-light-absorbing IOL group, and the difference was statistically significant (*p* = 0.0016). In the blue-light-absorbing IOL group, the abnormal fundus fluorescence decreased in three eyes whereas, in the colourless IOL group, it decreased in two eyes. Late AMD developed in one eye (1.9%) in the blue-light-absorbing IOL group and in nine eyes (11.4%) in the colourless IOL group, which was statistically significant (*p* = 0.042). Out of the nine eyes with late AMD in the colourless IOL group, six eyes had geographic atrophy and three eyes had wet AMD. The only eye that developed late AMD in the blue-light-absorbing IOL group had geographic atrophy.

Another study showing protection by blue-light-absorbing IOLs involved 66 eyes of 40 patients with geographic atrophy, 39 of which had implanted blue-light-absorbing IOLs and 27 had IOLs absorbing only UV light [329]. All IOLs absorbing blue light used the same chromophore, which, for 20 D lens, transmitted 8, 34, 49, and 68% of light of 400, 425, 450 and 475 nm wavelength, respectively (product information for SN60WF IOL available from the manufacturer website accessed on 8 August 2023, https://pdf.medicalexpo.com/pdf/alcon/model-sn60wf/80586-137180-_3.html). There was a statistically significant difference in the progression of geographic atrophy after one year between the blue-light-absorbing IOL group and the group with IOLs not absorbing blue light (0.72 ± 0.39 mm2 and 1.48 ± 0.88 mm^2^, respectively; *p* = 0.0002).

The above examples show how the methods for assessing sunlight exposure varied between the different studies and that none of them assessed exposure to sunlight of the retina. Assessing exposure of the retina to sunlight over the entire life is not easy. Even if it involves a detailed questionnaire, it relies on an elderly person to recall how much time they used to spend outdoors and what ocular protection they used, sometimes from early childhood, so it is not surprising that the accuracy may vary. None of these studies considered the transmission properties of the lens, which can affect the light reaching the retina. People with lenses exposed to more UV light in early life can develop the lenticular chromophores that block access of UV and blue light to the retina before substantial accumulation of RPE lipofuscin occurs.

There is great interindividual variability in the transmission properties of human lenses [79]. It can be expected that people wearing UV-blocking glasses or contact lenses, or spending most time indoors from early life are more likely to have smaller concentrations of lenticular chromophores blocking UV- and short-wavelength visible light. For such a person, moving into a sunny place and spending more time outdoors after retirement, may expose their retina to greater fluxes of short-wavelength light than those reaching the retina of a person whose lens blocks that light.

Finally, it needs to be kept in mind that AMD is a multifactorial disease with various genetic, clinical, and environmental risk factors identified, including smoking, diet, physical activity, body mass index, cardiovascular disease, hypertension, cataract, and several genetic variants involved in complement activation, lipid transport, remodelling of extracellular matrix, and angiogenesis [331,332]. Being outdoors may be associated not only with increased exposure to sunlight but also with increased physical activity, and consequently, with decreased body mass index, cardiovascular disease, and hypertension.

Sunlight has beneficial effects of great importance. In addition to the synthesis of vitamin D in the skin, other effects rely on light absorption by the retina and include photoentrainment, which is important in adjusting the body’s circadian clock to the day/night cycle [333,334]. Therefore, it is important to determine the exposure that is damaging to the retina so such exposures can be avoided while not compromising the beneficial effect of sunlight.

#### 4.6.11. How Much Sunlight Reaches the Retina?

Under physiological conditions, the radiance across the visual field varies considerably; consequently, there are topographical changes in the irradiance across the retina. Overall, the central retina is exposed to much greater levels of light than other parts [216] but no attempt has been undertaken to quantitatively evaluate the topography of retinal irradiance encountered during different daily activities at different times of the day at different seasons, atmospheric conditions, and latitudes.

When outdoors, as long as the Sun is in the visual field, it is the source of the greatest irradiance on the retina. It has been estimated that, in an emmetropic eye, direct gazing at the Sun, with a constricted pupil of 2 mm in diameter, produces an image of the Sun on the retina of 0.16 mm in diameter, and the irradiance in that small area is about 11 W/cm^2^ [80,335]. Obviously, that irradiance depends on the latitude, season, time of day, and atmospheric conditions. Other estimates of retinal irradiance in a human eye viewing the mid-day Sun vary from 1.6 W/cm^2^ for 400–500 nm range [217] to 18–122 W/cm^2^ for the full spectral range (including UV and infrared) [80,218]. The calculations of Pflibsen et al. suggest that if the Sun is within the 50° eccentricity to the visual axis, its projection on the retina exposes the retina to irradiance that is only 10% to 20% smaller (depending on age) than that due to gazing at the Sun directly; it is just that in this case, the image of the Sun is focused at a different retinal area than the centre of the fovea [336].

The visual field extends up to 50° superior to the visual axis. At the latitude of Cardiff, Wales (latitude of 51.48°), when walking and looking straight ahead, the Sun is in the upper part of the visual field throughout the whole day throughout the entire year except for a few weeks around the summer solstice when the Sun is above the visual field for up to 5 h around midday (https://www.sunearthtools.com/dp/tools/pos_sun.php?lang=en; accessed on 8 August 2023). The Sun in the upper part of the visual field can be expected to be focused on the inferior part of the retina. In contrast, in areas close to the Equator, such as Quito, Ecuador (latitude of −0.22°), the Sun is above the visual field of a person looking straight ahead throughout the substantial portion of the day throughout most of a year. Therefore, going for a walk at midday close to the Equator may be safer for the retina than going for a midday walk in Cardiff in early spring.

These considerations do not take into account the eyelid position, which can limit the access to the pupil of bright light from above and can restrict the superior visual field to only 25–30° or even less when squinting the eyelids [216]. The bright reflection of sunlight from water, sand, or snow is likely to be focused on the superior part of the retina (when the head is in an upright position and a person is looking straight ahead). Fresh snow can reflect as much as 85% of short-wavelength light, whereas grass reflects only 1–2%. Therefore, depending on the surroundings, the retinal irradiance can vary greatly. Considerations like that should be taken into account when evaluating retinal exposure to sunlight, which certainly is not easy, let alone if it is meant to be evaluated nearly for someone’s entire life, as was attempted in some population-based and case-control studies of AMD. There is no doubt that the evaluation of someone’s lifelong exposure to sunlight is challenging. However, current technology (spectrophotometers, eye trackers, spectroradiometers) enables the evaluation of topographical irradiance of the retina during various activities in various geographical locations. Future longitudinal studies could combine such data not only with meteorological data but also with data from mobile phone apps, such as Google Maps Timeline to give more accurate estimates of retinal exposures to light.

#### 4.6.12. Does Lipofuscin Contribute to Light-Induced Injury of the Retina In Vivo?

Altogether, while there is a large body of evidence on the deleterious effects of RPE lipofuscin in cultured cells exposed to light, the evidence suggesting that lipofuscin can exert deleterious effects in the retina in vivo is still rather limited and requires further exploration. It is well documented that gazing at the Sun can cause focal retinal injury after exposures lasting as little as one or a few minutes [80,217,218]; however, it is unclear what relative contribution to that injury is caused by retinaldehydes and what by lipofuscin. It can be expected that it depends on the age and spectral transmission of the lens.

Yellowing of the lens provides protection from photoexcitation of both retinaldehydes and lipofuscin. However, the absorption spectrum of photosensitizers present in lipofuscin extends further towards longer wavelengths than those of retinaldehydes [94] (Figure 6). Therefore, it may suggest that the age-related increase in the lipofuscin content and age-related decrease in the short-wavelength light reaching the retina can increase the contribution of lipofuscin to phototoxicity. However, the aged human retina is characterized not only by increased concentrations of lipofuscin but also by misalignment and convolutions of rod outer segments [337]. This suggests a possibility that the trafficking of retinaldehydes is impaired, so both the enzymatic reduction of all-*trans*-retinaldehyde as and the incorporation of 11-*cis*-retinaldehyde into opsin to create rhodopsin can be delayed. Regeneration of all visual pigments is required to regain the amazing retinal sensitivity to light, which can be achieved in the dark [338]. It has been reported that dark adaptation observed in elderly patients, which requires regeneration of all visual pigments, is delayed [339,340]. This supports the hypothesis that trafficking of retinaldehydes is impaired in the aged retina, which may increase the chances of their photoexcitation. Therefore, it is still unclear what the contribution of lipofuscin to light-induced retinal injury is.

The retina is protected from exposure to excessive fluxes of light by various mechanisms (reviewed in [94]). They include limiting fluxes of light reaching the retina by the eyelid, eyelashes and eyebrows, and the action of the iris regulating the pupil size. Moreover, the high concentrations of enzymatic and low-molecular-weight antioxidants, detoxification enzymes, and efflux transporters can prevent oxidative damage to biomolecules and/or remove damaged products. Thus, it may be argued that the retina is well equipped l to defend itself from the noxious species generated by lipofuscin and retinaldehydes, and the deleterious effects occur only when the capacity of these cellular defences is exceeded.

## 5. Fluorescence of RPE Lipofuscin

When excited with light, isolated RPE lipofuscin emits a characteristic golden-yellow fluorescence with a broad maximum at about 580–630 nm for lipofuscin in suspension and at about 610–630 nm for lipofuscin dried on a microscope slide [64]. The emission peaks are at similar wavelengths, irrespective of the excitation wavelengths of 364 nm or 476 nm. For lipofuscin isolated from 5–29-year-old donors, the emission spectra exhibit shoulders at 470, 550, and 680 nm. The contributions of 470 and 550 nm shoulders to the emission spectrum are less pronounced for lipofuscin from 30–49-year-old donors, and are no longer noticeable for lipofuscin from the group of donors above 50 years of age. In contrast, the contribution from the 680 nm shoulder seems to increase with age.

Spectrally resolved confocal microscopy with a greater sensitivity of fluorescence detection than in experiments of Boulton et al. [64] enabled to show that there is a large variation in the emission maxima between individual lipofuscin granules with the wavelengths corresponding to the maxima varying from 513 to 645 nm [176,179,341]. It has been demonstrated that varying the excitation wavelength from 532 to 400 nm does not affect the long-wavelength downward slope of the emission, and the emission maximum remains the same for the excitations with 400, 457, or 488 nm, and shifts slightly towards longer wavelengths for excitations with 514 or 532 nm. With the increasing excitation wavelength, the emission spectrum becomes narrower, suggesting a decreasing number of excitable fluorophores as the excitation wavelength increases. Studies of fluorescence emission and polarization spectra upon excitation with 400 nm light have shown that the emission is significantly polarized for 430 nm wavelength and that polarisation decreases rapidly with increasing wavelength until reaching a relatively steady low value at 575 nm, which remains at that level till at least 725 nm (the end wavelength of the detection range used in the study). These changes in polarization for different emission wavelengths suggest that the fluorophores contributing to the emission in the blue and green spectral ranges are different than the fluorophores contributing to the emission in the yellow-red spectral range. It appears that at least some fluorophores emitting in the blue-green spectral range are also the chromophores absorbing the excitation light, whereas at least some fluorophores emitting in the yellow-red spectral range are excited via energy transfer from the blue-light-absorbing chromophores. The simultaneous collection of NSOM images of fluorescence and topography of lipofuscin granules shows that orange-emitting fluorophore(s), are not uniformly distributed within the granule [176,177,180].

A2E and other bisretinoids are considered the major emitters of yellow and red fluorescence from photoexcited RPE lipofuscin [127,342,343]. Quantum yields of A2E fluorescence have been studied in Triton X-100 micelles in aqueous solution as well as in various organic solvents and shown to be the greatest in Triton X-100 where 1.1% of absorbed photons are used for emission of fluorescence [344,345]. A direct comparison of fluorescence quantum yields of A2E with the lipophilic extract of lipofuscin shows that fluorescence emission from the lipophilic extract of lipofuscin is 70-fold greater than for A2E [224,346]. Thus, the small contribution to the absorption of lipofuscin and the findings of Haralampus-Grynaviski et al. suggest that to be the major emitter of fluorescence in the lipofuscin granule, A2E must act as an energy acceptor from other blue-light-absorbing chromophores [179]. While the blue-light-absorbing chromophores, which can act as energy donors for A2E have not been identified, it can be suggested that some products of oxidation of DHA can be the potential candidates [75,240]. Lipofuscin contains proteins modified by products of lipid oxidation, such as 4HNE, MDA, and therefore it can be expected that they contribute to lipofuscin fluorescence excited by UV light [69,70,71,166,347].

It has been demonstrated that exposure of lipofuscin isolated from human RPE to visible light results in substantial changes in the fluorescence spectra [75,348]. For experiments shown in Figure 8, lipofuscin granules were suspended in PBS, exposed to 9.76 mW/cm^2^ visible light, and their fluorescence emission spectra were measured at selected times for up to 10.5 h [75]. As a result of exposure to visible light, there is a dose-dependent decrease in the intensity of fluorescence above 560 nm. The emission in the range of 530–550 nm exhibits changes dependent on the excitation wavelength: an excitation with 360 nm light results in a small, dose-dependent increase in fluorescence followed by a plateau, whereas an excitation with 488 nm causes an initial increase in the fluorescence emission in that range followed by a decrease. The decrease in the long-wavelength fluorescence during the photodegradation of lipofuscin is accompanied by a concomitant increase in the emission in the range of 380–520 nm observed upon excitation with 360 nm light.

Because bisretinoids are the major emitter of lipofuscin fluorescence [127,128,179], it is plausible that some of the observed changes in fluorescence during lipofuscin photodegradation [75,348] can be ascribed to light-induced oxidation of bisretinoids, which are prone to oxidative degradation.

To compare the changes of fluorescence upon photodegradation of lipofuscin and A2E, a suspension of lipofuscin granules and a suspension of proteoliposomes made of cardiolipin, bovine serum albumin and A2E were exposed to visible light under the same conditions [348]. Photodegradation of A2E proteoliposomes led to a decrease in fluorescence above 540 nm, which appears to be an isosbestic point below which an increased fluorescence occurs. Some of the A2E photodegradation products are hydrophilic enough to diffuse out of the liposomes and exhibit that short-wavelength fluorescence. The emission maximum of the A2E photodegradation products appears at a shorter wavelength than the emission maximum of the lipofuscin photodegradation products. Another source of short-wavelength fluorescence appearing during the photodegradation of lipofuscin can be oxidized DHA [75].

An addition of A2E to the suspension of photodegraded lipofuscin granules results in an expected increase in the long-wavelength emission and in a decrease in the short-wavelength emission, providing additional evidence of energy transfer to A2E from the chromophores absorbing 365 nm light used for photoexcitation [348].

Spectral changes of fluorescence emission similar to those observed during photodegradation of lipofuscin suspension have been observed upon exposure to visible light of lipofuscin-laden RPE cells, ARPE-19 (Figure 9) [75]. Cells were exposed to the same irradiance as the suspension of lipofuscin but, unlike the exposure of lipofuscin in suspension, which was continuous, the radiant exposure of cells was fractionated into 45-min daily exposures delivering the same total dose after 14 days as that delivered after 10.5 h of irradiation of lipofuscin in suspension. This approach was needed to ensure the maintenance of cell viability and function. Interestingly, upon exposure to light, the cells assembled vesicles containing multiple lipofuscin granules, which appeared floating above the cell monolayer. As a result of the removal of lipofuscin from cells, both - the density of lipofuscin granules inside cells and the fluorescence intensity of cells solubilized in Triton X-100 considerably decreased. There was a marked increase in the ratio of blue-green emission to yellow-red emission.

## 6. Fluorescence of the Retina

### 6.1. Sources of Fluorescence in the Retina

In addition to lipofuscin, the retina contains a number of other fluorophores. Like all other cells from different tissues, retinal cells contain NAD(P)H, flavins, and porphyrins, which emit fluorescence upon absorption of light. NAD(P)H requires excitation with ultraviolet light and emits broad-band fluorescence ranging from 400 up to 650 nm with maxima at ~440 and 460 nm for the protein-bound and free state, respectively [349]. Flavins can be excited by ultraviolet light and blue light and emit in the range of 470–650 nm with maxima at about 540 nm, respectively [350,351]. The fluorescence of flavins and NAD(P)H has been investigated as a way of monitoring the metabolic status of cells. Porphyrins appear transiently in the multistep biosynthetic pathways of haems, which serves as a prosthetic group of many proteins, including cytochromes, cytochrome c oxidase, catalase, peroxidases, and normally porphyrins, which do not accumulate in substantial concentrations [352].

Photoreceptors, Müller cells and RPE contain considerable concentrations of retinoids, which also exhibit fluorescence properties [353,354]. In particular, retinyl palmitate, which serves as a storage form for vitamin A from which the chromophores for visual pigments are synthesized, can accumulate in particularly high concentrations in the RPE and emits broad range fluorescence between 400–650 nm with a maximum at ~490 nm when excited with ultraviolet light [353,354,355].

In the RPE, in addition to lipofuscin, there are other pigment granules, which can emit fluorescence: melanosomes and melanolipofuscin [64,356]. The amount of melanosome in human RPE cells decreases with age, whereas melanolipofuscin increases [77,357].

Studies of pigment granules isolated by differential centrifugation from human RPE revealed that melanosomes, when excited with 364 nm light, emit broad-band fluorescence with emission peaks at 440 and 560 nm and a tail extending beyond 700 nm [64,356]. A comparison of the intensity of fluorescence at their corresponding emission maxima showed that the fluorescence intensity of RPE melanosomes is 3.7- to 6.7-fold smaller than for lipofuscin. The excitation maxima of the isolated melanosomes appear at 350 and 450 nm. Isolated melanolipofuscin granules also emit broad-band fluorescence, extending beyond 700 nm, with the main emission peak at about 560 nm and shoulders at 460 and 680 nm. The excitation spectrum of melanolipofuscin is similar to that of lipofuscin with a broad maximum at about 400 nm and a relatively sharp peak at 460 nm.

In contrast to the findings on isolated pigment granules described by Docchio et al. and Boulton et al. [64,356], the evaluation of pigment granules in RPE cells by high-resolution structured illumination microscopy (SIM) and detection of fluorescence excitable by 488 nm laser, allowing for the lateral resolution improvement to 110 nm, demonstrated that human RPE cells do contain melanosome granules, both spherical or spindle-shaped, which do not emit fluorescence when excited with blue light [358]. These findings are based on an analysis of an impressive number of 193,096 individual pigment granules in 450 RPE cells. Bermond and colleagues also identified different types of melanolipofuscin granules, some of which exhibit nonfluorescent cores similar to melanosomes surrounded by a fluorescent coat, others appear as fluorescent inclusions inside the nonfluorescent granules, while some other granules appear highly fluorescent throughout. It is possible that some of these granules classified as melanolipofuscin by Bermond and colleagues have a high enough density to pass through 2 M sucrose together with melanosomes but do not appear as different to melanosomes by transmission electron microscopy, the techniques used by Boulton and colleagues for isolation and testing the purity of pigment granules, respectively.

Unlike lipofuscin, for which photooxidation results in an increase in the short-wavelength fluorescence and a decrease in the long-wavelength fluorescence, photooxidation of melanosomes leads to an increase in their fluorescence throughout the visible and near-infrared [75,359,360].

Due to the high concentration of polyunsaturated fatty acids, especially in photoreceptor outer segments, lipofuscin, drusen, and Bruch’s membrane, it can be expected that products of lipid oxidation and their adducts with proteins can contribute to retinal fluorescence [69,70,71,75,166,347].

In addition to the fluorophores present in the normal retina, certain pathologies result in the accumulation of additional fluorophores. In the case of impaired phagocytosis of POS by RPE, such as that occurring in the Royal College of Surgeons (RCS) rats, there is an accumulation of material with lipofuscin-like fluorescence at the interface of POS and RPE [361]. RCS rats carry a mutation in *MERTK* gene coding a receptor Mer tyrosine kinase (MerTK) [362]. MerTK is a widely expressed receptor involved in phagocytosis of shed POS by the RPE as well as apoptotic bodies by macrophages and microglia [362,363,364,365]. Mutations in human *MERTK* can cause early-onset retinal degenerations including rod-cone dystrophy (also known as retinitis pigmentosa) and cone-rod dystrophy [366]. There are a number of other conditions affecting the human retina where the accumulation of non-phagocytosed POS can occur and has been documented in a number of case reports, such as detachment of the neural retina from the RPE [367], central serous chorioretinopathy [368], vitelliform macular dystrophy [369] or optic nerve pit maculopathy with serous macular detachment [370].

In the cases of intraretinal or subretinal haemorrhages, the metabolic degradation of haemoglobin can result in the accumulation of biliverdin and bilirubin, which, when excited by UVA or blue light, exhibit broad-band fluorescence emission with maxima at about 470–480 and 520–540 nm, respectively, and the long-wavelength tail extending well above 600 nm, especially for bilirubin bound to serum albumin or erythrocyte ghosts [371,372,373,374].

Subretinal and sub-RPE deposits, which accumulate with age and some of them are a hallmark of AMD, such as reticular pseudodrusen, basal laminar deposits, basal linear deposits and drusen, can emit fluorescence [375,376,377]. The fluorescent properties of drusen appear highly variable, ranging from homogenous fluorescence to highly irregular with hypo- and hyper-fluorescent areas, and some with no fluorescence.

### 6.2. Imaging of Fluorescence in the Retina

Fluorescence of the retina can be detected in the human eye noninvasively in vivo [353,378,379]. Because it does not involve exogenous fluorophores, it is often referred to as autofluorescence. In an elegant series of experiments, Delori et al. have demonstrated that fundus autofluorescence is mainly due to the fluorescence of retinal structures behind photoreceptors and exhibits characteristics of RPE lipofuscin [379]. These experiments involved 30 White volunteers ranging in age from 21 to 67 years, with a distribution of 6 individuals per decade. The fluorescence excitation was obtained from a xenon arc lamp and one of the fluorescence filters transmitting light centred at 430, 450, 470, 490, 510, 530 or 550 nm. The area of the retina exposed to the excitation was a 3° in diameter centred either at the fovea or at 7° temporal to the fovea, while the emission was collected from an area of 2° (corresponding to 585 µm) in diameter. The emission intensity was corrected for the spectral characteristics of the detection system. The emission spectra from the human fundus exhibited maxima at 620–630 nm for all excitation wavelengths except for 550 nm, which caused a small shift of the maximum of about 10 nm towards longer wavelengths. The excitation spectrum recorded for the emission at 620 nm exhibited a maximum at 510 nm. For the excitation with 430 or 470 nm, some emission spectra showed additional fluorescence with the broad emission between 520–580 nm, indicating that more fluorophores contribute to the emission induced by blue light than by green light.

Delori and colleagues compared the spectral characteristics of the fluorescence emission from human fundus 7° temporal from the fovea in vivo with the spectral characteristics of RPE explants taken at 26–33° from the fovea of post-mortem donor eyes fixed in 4% paraformaldehyde and demonstrated that they are similar [379].

They also demonstrated that the topographical distribution of fundus fluorescence and an age-related increase in fluorescence intensity are consistent with the quantification of lipofuscin granules in the RPE sections examined by TEM [77,379]. Fundus fluorescence in vivo excited by light of 510 or 550 nm wavelength and monitored at 620 nm shows an age-related statistically significant increase both in the fovea and at 7° temporal to the fovea. For excitation with 470 nm light, only the fluorescence at 7°exhibits a statistically significant age-related increase whereas fluorescence at the fovea does not. For the emission obtained by excitation with 430 nm light, there is no correlation with age in that age range of 21–67 years.

For all excitation wavelengths, the fluorescence at the fovea is 41 to 75% less intense than at 7° away from the fovea [379]. This effect is more pronounced for excitations with blue light than for green light of 510 or 550 nm wavelengths, indicating that, apart from lower lipofuscin content in the fovea, additional factor(s) are involved. Macular pigment present in the inner retina can efficiently absorb light of 430 and 470 nm wavelengths, but much less green light of 510 nm wavelength, and its absorption of 550 nm light is negligible. The difference spectrum obtained by subtracting the fluorescence excitation spectrum from the fovea from the spectrum for the 7° allowed the authors to calculate the absorption spectrum of the chromophores absorbing blue light and demonstrate that it matches the absorption spectrum of the macular pigment, thus providing an additional explanation for the lower fluorescence emission from the fovea than outside this area for the excitation with blue light.

On the temporal side from the fovea, the fluorescence intensity remains high up to 15° and then gradually declines towards the equator [379].

As an additional test of the site of origin of retinal fluorescence, Delori and colleagues compared fundus fluorescence at 10–15° temporal to the fovea from dark-adapted and light-adapted volunteers and demonstrated that the difference in emission spectra can be explained by the visual pigment of rods absorbing partly the 450 nm light used for excitation and emitted fluorescence [379]. This provided evidence that the fluorescence originates behind POS.

To examine the origin of fluorescence in retinal layers behind POS, Marmorstein et al. investigated fluorescence properties of nonfixed cryo-sections from human retina-choroid-sclera complex isolated postmortem from normal eyes of 81, 83, and 87 years of age. They used 364, 488, 568, and 633 nm as the excitation wavelengths of the laser scanning confocal microscope [375]. The emission spectra were not corrected for the changes in detection sensitivity at different wavelengths, but as they were collected with a Hamamatsu R6357 detector, which exhibits similar radiant sensitivity and quantum efficiency for different wavelengths from the range of 400–800 nm, it can be expected that the reported wavelengths of the emission maxima are close to the true values. They have demonstrated that RPE emits strong fluorescence when excited with any of the four wavelengths, and the emission spectra maxima shift towards the longer wavelength as the excitation wavelength increases to yellow (568 nm) or red (633 nm) light [375]. For the excitations with 364 or 488 nm light, the emission maxima for the RPE are at the same wavelength of 555 nm. For the excitations with 568 or 633 nm, the emission maxima are at 615 and 655 nm, respectively. The Bruch’s membrane and sub-RPE deposits: basal laminar, basal linear deposits, and drusen, also emit fluorescence but with the emission maxima at shorter wavelengths than for the RPE. The most pronounced difference has been seen for the excitation with 364 nm, where the emission maximum for Bruch’s membrane/sub-RPE deposits is at 485 nm, which is 70 nm shorter than the wavelength for the emission maximum from the RPE. For other excitation wavelengths, the differences are only 5–15 nm.

The intensities of the emission maxima of the Bruch’s membrane/sub-RPE deposits are about 10-fold smaller than for the RPE for excitations with 633 and 568 nm [375]. For the excitation with 488 nm, the emission maximum of the RPE is still 40 and 60% greater than for Bruch’s membrane and sub-RPE deposits, respectively. However, the emission intensity maximum of Bruch’s membrane is similar to that of the RPE for the excitation with 364 nm. Because the emission spectrum of Bruch’s membrane is narrower than that of the RPE, it can be expected that, when measuring total fluorescence intensity integrated over a wide spectral range, the contribution of Bruch’s membrane can be smaller than that judged by the intensity of its spectral maximum. The emission spectrum of sub-RPE deposits excited by 364 nm light is similar to that of Bruch’s membrane but its maximum is about 15% smaller.

All in all, Marmorstein and colleagues have demonstrated that RPE fluorescence strongly dominates for excitations with 568 or 633 nm light; excitation with 488 nm brings about strong emission of not only RPE but also considerable contributions from Bruch’s membrane and sub-RPE deposits, which become comparable to the RPE emission upon photoexcitation with 364 nm light [375]. These studies were conducted on 8 µm thin tissue cross-sections, therefore there was no issue with the RPE pigments preventing the excitation light from reaching the Bruch’s membrane and sub-RPE deposits, as well as filtering the emitted light. This issue is encountered when imaging the fluorescence en face mainly due to the filtering effect of RPE pigment granules [77,357,380]. Moreover, this filtering effect is likely to change with ageing because (i) there is an overall loss of melanin [81]; (ii) there is an accumulation of lipofuscin and complex granules containing both melanin and lipofuscin [77,182]; and (iii) melanosomes redistribute from their initial location in the apical portion of the RPE and lipofuscin redistributes from its initial location at the basal site to become distributed more evenly throughout the cell [77,182].

Another study of fluorescence of RPE and sub-RPE deposits has been conducted on paraformaldehyde-fixed retinal cross-sections from Caucasian eyes and employed multiphoton laser scanning microscopy using 960 nm wavelength from 140 femtosecond laser as the excitation source [377]. Following such multiphoton excitation, sub-RPE deposits exhibit a broad emission maximum at about 560 nm whereas the emission maximum from the RPE is at about 610 nm.

In a study on twenty 20 µm thick flat-mounts from human RPE-Bruch’s membrane complexes without retinal pathologies, Smith et al. used narrow-band excitation from mercury arc lamp and dichroic filters with transmission centred at either 436 or 480 nm, and measured fluorescence emission spectra en face at the fovea, at perifovea (2 mm superior to the fovea), and at the periphery (10 to 12 mm superior to the fovea) using a hyperspectral camera, which recorded a fluorescence emission spectrum every 10 nm in the range of 420–720 nm from each pixel in the imaged area [381]. The emission spectra were corrected for the spectral changes in the quantum efficiency of the detector. The areas devoid of RPE allowed for the collection of emission spectra from Bruch’s membrane, which exhibited a broad flat maximum at about 550 nm when excited with either 436 or 480 nm. The position of that maximum was similar to the emission maximum from the Bruch’s membrane/sub-RPE deposits at about 540–545 nm obtained by photoexcitation with 488 nm laser reported by Marmorstein et al. on nonfixed cryo-cross-sections from human retinas-choroid-sclera complex described earlier [375]. The authors did not provide information on the age of donors but did provide some examples where the donor’s age was stated [381]. This included an emission spectrum from the perifovea of a 47-year-old donor, which exhibited a global maximum at about 560–570 nm and the whole spectrum could be fitted by a combination of four Gaussian curves, with the maxima at about 525, 555, 600 and 640 nm. The fitting procedure was performed for the emission spectra from each pixel. As stated by the authors, the most consistent emission maxima obtained by excitation with 436 nm were between 525–540 nm and 550–575 nm. The emission maxima obtained by excitation with 480 nm were 20–30 nm shifted towards longer wavelengths. Local maxima or shoulders could be seen at 600 and 640 nm in the perifovea and periphery for both excitation wavelengths. A small maximum appeared also at 700 nm. Importantly, the authors developed a mathematical procedure for finding the best-fitting combination of four Gaussian curves and an emission spectrum of Bruch’s membrane to the emission spectrum from each pixel across a considerable retinal area. Absorption by RPE melanin of the excitation light and emitted light was considered when evaluating the contribution of Bruch’s membrane emission to the total emission of the RPE-Bruch’s membrane complex. This then allowed for mapping the contributions of individual fluorophores, or rather groups of chromophores, across the imaged area.

Using SIM on RPE-Bruch’s membrane flat-mounts from 15 Caucasian donors, Bermond et al. demonstrated that, in the fovea, melanolipofuscin is more abundant than lipofuscin, lipofuscin predominates in the perifovea, whereas in the near periphery, the abundance of both types of pigment granules is similar [358,382]. Overall, the granule densities and intensities of RPE fluorescence appear rather uniform, suggesting that RPE cells can control granule numbers [382]. RPE cells with the lowest granule numbers and low fluorescence intensity excited by 488 nm light contain more melanolipofuscin than lipofuscin, whereas lipofuscin predominates over melanolipofuscin in cells with intense fluorescence and high granule load.

The spectrofluorometer developed by Delori et al. mentioned earlier collected emission spectra from the human eye in vivo from a retinal area of about 2° in diameter [379]. In parallel, Delori et al. and Ruckmann et al. have developed a confocal scanning laser ophthalmoscope (SLO), where the fluorescence was excited by laser light, and allowed for detailed mapping of fluorescence topography across relatively large square retinal areas of 30° in length [383,384,385,386].

Since the pioneering work of Delori and von Rückmann’s groups, fundus autofluorescence imaging has been further improved, commercialized, and has become a part of routine clinical evaluation of the retina [378,387,388,389]. Typical current imaging systems employ either scanning lasers or lamps equipped with fundus cameras and vary with excitation wavelengths from 435 to 610 nm for imaging of lipofuscin fluorescence, and 780–795 nm excitation for imaging of melanosomes fluorescence. Both lipofuscin and melanosomes can emit fluorescence upon excitation with 633 nm light.

While the collection of the whole emission spectra was performed in the initial instrument developed by Delori et al. [379], this option has not been made available in the commercial instruments. Later studies demonstrated that using photoexcitation with 450 nm-light emitting diode (LED) and recording emission in two spectral channels: green (510–560) and yellow-red (560–700) allows seeing that retinal areas vary in the ratios of short-wavelength to long-wavelength emission [390,391,392,393]. Therefore, the development of a hyperspectral camera with the ability to collect the emission spectra for every pixel has been of particular importance [394]. The recent applications of the hyperspectral camera for ex vivo imaging of the RPE-Bruch’s membrane explants and human eyes in vivo have demonstrated its great potential for differentiating emission from different retinal fluorophores [353,395,396].

Combining scanning laser ophthalmoscopy with adaptive optics (AOSLO) to minimize the aberrations of the anterior segment of the eye allows for a selective collection of fluorescence emission from RPE cells excited with blue or near-infrared light [397]. Further information can be extracted by adding confocal laser scanning reflectance imaging and optical coherence tomography (OCT) to visualize individual RPE cells and their morphometry, identify areas with drusen or subretinal drusenoid deposits that can help in the explanation of the apparent disruption of the RPE cell mosaic seen in these areas with fluorescence emission imaged with AOSLO. This is important so to discriminate fluorescence originating from the RPE from fluorescence from other areas (Table 1).

Another fluorescent technique, which can add complementary information on fluorophores is imaging of fluorescence lifetimes [353,395,398]. Upon a short excitation pulse, in the range of pico- or even femtoseconds, and absorption of photons by fluorophores of the same type, the intensity of emission from fluorophores in, a dilute solution, decays in an exponential manner, with a rate of decay being a reciprocal of the fluorescence lifetime. Each type of fluorophore has its own fluorescence lifetime, which can change due to the interaction of the fluorophores with the solvent or other solutes. Oxygen is a well-recognized quencher of fluorescence because it can act as an energy acceptor from the excited electronic states. As a result of energy transfer from a fluorophore to an energy acceptor, the fluorescence lifetime is decreased, and consequently the total integrated intensity of fluorescence is decreased.

As described earlier, energy transfer can occur from UV- and blue-light-absorbing chromophores of lipofuscin to A2E [179]. If these chromophores exhibit fluorescence, the loss of A2E in lipofuscin due to photooxidation can increase their fluorescence lifetime. This is consistent with the experimental results of Semenov et al., who have shown that exposure of lipofuscin-laden ARPE-19 cells to sublethal doses of visible light results in an increase in the lipofuscin fluorescence lifetime and this effect can be prevented by supplementation of cells with zeaxanthin [399]. The effect of zeaxanthin can be due to its ability to quench excited triplet states of photosensitizers and singlet oxygen thereby preventing photooxidation of A2E.

Moreover, melanin is a well-recognized acceptor of energy from excited electronic states (reviewed in [81]). Therefore, it can be expected that fluorophores in proximity to melanin within the melanolipofuscin granule will exhibit shorter fluorescence lifetimes.

The quenching effect of melanin on A2E fluorescence was demonstrated by Guan et al., who isolated pigment granules from RPE cells, quantified A2E in melanosomes, melanolipofuscin and lipofuscin per mg of dry granule mass and showed that, despite having a substantial content of A2E of 3.5 ± 0.1 µg/mg, which was similar to A2E content in lipofuscin (3.7–6.7 µg/mg), melanosomes did not emit detectable fluorescence upon photoexcitation with 488 nm laser [174].

Due to the numerous fluorophores present in the retina, and their different environments, it can be expected that the fluorescence decay is a superposition of tens, if not more, of different exponential decays. For the lipofuscin granules, there is a need to fit with at least four exponential decays to obtain a reasonably good fit [225,356]. The same is true for melanolipofuscin and melanosomes.

In currently used clinical fluorescence lifetime imaging systems, the fluorescence is excited by a pulsed laser-emitting visible or near-infrared light, and the decay is measured in two or more different spectral channels [395,400,401].

When measuring the fluorescence from the fundus, the age-related changes in absorption properties and fluorescence of the lens need to be taken into consideration [402,403,404,405]. With age, there is a progressive build-up of lenticular chromophores absorbing initially ultraviolet, then violet, blue light, and eventually even longer-wavelength light [79]. Some of these chromophores have fluorescence properties, which can interfere with measurements of the fluorescence from the fundus by absorbing some excitation/emission light and by emitting their own fluorescence, thereby modifying the acquired emission spectrum and altering the fluorescence decay kinetics [402,403,404,405]. Moreover, with age, there is an increased risk of aggregation of lens crystallins, causing light scatter that eventually becomes visible as opacities known as cataracts, all of which affect both the intensity and spectrum of detected fluorescence [402]. While in their initial instrument, Delori et al. measured the actual spectral transmission of the anterior segment and accounted for it to extract the fundus fluorescence [379], the later studies, where the effects of the anterior segment were considered, used the normative data of van de Kraats and van Norren to account for the age-related decrease in spectral transmission of the lens, and, therefore, did not account for the great variability in the lens chromophores/fluorophores between individuals [79,406]. A recent clinical study clearly demonstrated that lens opacities can affect the measurements of quantitative fundus autofluorescence [404].

By increasing the excitation wavelength to near-infrared, the effects of absorption and scatter by the lens can be minimized. To enable the excitation of retinal fluorophores like lipofuscin, the photon density needs to be high enough to allow for two-photon excitation. This can be achieved by employing short pulses of high-intensity near-infrared laser. The most recent development in clinical imaging of fundus fluorescence is spectral imaging of fluorescence emitted in the range of 400–700 nm upon intense pulse excitation with two photons of near-infrared wavelengths and the use of adaptive optics [252,353,354,395,407,408,409,410,411,412,413,414,415,416,417,418,419,420,421]. The emitted light can be spectrally analysed, which enables the detection of specific fluorophores. For example, *abca4(-/-)rdh8(-/-)* double knockout mice with accumulated lipofuscin show the greatest intensity of fundus fluorescence collected in 594–646 nm range upon two-photon excitation with 780 nm pulsed laser. *Rpe65(-/-*) knockout mice, which accumulate relatively low amounts of lipofuscin but many retinyl esters in the RPE, exhibit less intense emission in 594–646 nm range than *abca4(-/-)rdh8(-/-)* but more intense emission in the 400–550 nm range [395]. This technique has been tested for safety in imaging of the retinas of human volunteers with healthy eyes, and it can be expected to be used soon to image diseased retinas (Figure 10). In addition to retinyl esters and lipofuscin, there are a number of fluorophores and photosensitizers in the RPE, which can be excited by absorption of two photons of 780 nm wavelength, including retinoids, flavins, and porphyrins.

Summing up, fundus autofluorescence techniques allow visualizing fluorescence distribution, its spectral characteristics and lifetimes, and are widely used for mapping the areas of geographic atrophy in various retinal diseases, including Stargardt’s disease and age-related macular degeneration, where the loss of RPE cells is followed by their clearance resulting in distinct hypofluorescent areas visible with excitation with blue, green, and near-infrared light [422,423,424,425,426,427,428,429,430,431,432].

### 6.3. Age-Related Changes in Retinal Fluorescence

Lipofuscin-like autofluorescence of the fundus appears very early during development. Fundus autofluorescence excited with 532 nm laser from confocal SLO has been imaged in 13 premature infants who were screened for retinopathy of prematurity and showed that fluorescence is detectable at that early age, appearing first in the posterior pole and developing centrifugally towards the periphery over the course of the following 30 weeks [433]. Importantly, using 532 nm as the excitation wavelength avoids the confounding effect of the macular pigment.

In addition to the previously discussed studies by Feeney-Burns, Delori and colleagues [77,379], a number of other studies in vivo and ex vivo demonstrated an age-related increase in the fluorescence of the fundus and RPE-Bruch’s membrane complex [181,182,385,434,435,436,437] (Table 2). Wing et al. used 44 human eyes from 35 donors, ranging in age from a 6-week premature newborn to 88 years, and measured fluorescence from retinal cross-sections through the entire retina including the fovea and optic disc [181]. The excitation was provided by a xenon arc lamp and a filter with transmission centred at 380 nm. The emission was collected using a band-pass filter transmitting blue light from the range of 460–480 nm. The results have shown that the RPE fluorescence increase occurs particularly fast in the first and second decades of life, then it appears to slow down followed by an increase in people above the age of 60 years. They also showed that there is an age-related increase in emission intensity across the whole retina when results were analysed based on four age groups: premature-12-months, 17 months-30 years, 31–60 years, and 61–88 years. The fluorescence emission from tissues of the youngest age group appeared green suggesting that it was mostly due to retinoids and not lipofuscin. There was about a 40% increase in fluorescence emission intensity in the oldest age group in comparison with the 31–60-year-old group.

Weiter and colleagues evaluated lipofuscin fluorescence at 470 nm excited by 365 nm light in the RPE across the retina post-mortem from 19 White donors, ranging in age from 2 weeks old to 88 years old, and from 19 Black donors, from 6.5 years old to 90 years old [182]. A linear fit of the data showed a statistically significant increase in emission intensity with age only for Whites, with r^2^ ranging from 0.45 to 0.65 depending on the location in the retina, and *p* = 0.0001. The actual data were not shown.

Measurements of fundus fluorescence in vivo excited by 488 nm laser and collected from the fovea and an area 7–15° temporal to the fovea in healthy retinas of 33 White subjects 6–78 years of age demonstrated a linear increase with age up to about 60 years, above which the emission appeared to plateau [385]. The emission was collected using a long-pass filter transmitting light above 521 nm to at least 650 nm.

A couple of studies of fundus fluorescence in vivo excited by 488 nm laser were performed on 156 participants, 5 to 77 years of age [436,437]. The fluorescence was collected from 500–750 nm and quantified in a grid centred at the fovea. Two different types of grids were used, each subdivided into a different number of rings and subdividing segments. The outer edge of the outermost grid bordered the edge of the optic nerve head. Overall, fluorescence monotonically increased with age both in the fovea and extrafoveal circle extending to the optic nerve head, with an initial rapid linear increase up to the age of 20 years, possibly reaching a plateau around the age of 60 and further increase after the age of 65. The most intense fluorescence had a hot spot in the temporal and/or temporal-slightly superior area of the perifovea.

Okubo et al. measured fluorescence in 5 µm-thick sections from the formalin-fixed 8 mm in diameter circles centred on the fovea of 88 donors ranging in age from 1–98 years [438]. Excitation was provided by a mercury lamp equipped with a filter transmitting 450–490 nm light, and emission was collected after passing through a 520 nm long-pass filter. In this study, the emission also showed a linear increase up to the age of 60 years, followed by a plateau. A similar increase followed by a plateau was obtained from morphometric quantification of the cytoplasmic volume occupied by lipofuscin on other sections from the same retinas imaged with TEM.

Consistent findings were reported by Delori and colleagues from a study on 145 participants ranging in age from 15 to 80 years where the fundus fluorescence was excited by 470 nm and the emission integrated above 520 nm [434]. The emission intensities reached a maximum for the age group in their seventh decade and remained at the same level in the eighth decade. In contrast, when the excitation of the fundus of the same subjects was performed with a xenon arc lamp equipped with a broad-band transmission filter centred at 550 nm, and the emission was collected in the range of 650–750 nm, a linear increase in fluorescence occurred up to the age of 70 years, followed by steep decreases in all five locations examined: the fovea, and at 7° temporal, nasal, superior and inferior to the fovea. The rate of age-related decrease in fluorescence at the temporal site of elderly fundi was about 2.5-fold greater than the rate of the fluorescence increase in younger fundi.

The largest study of fundus fluorescence as a function of age was conducted by Greenberg and colleagues on 277 subjects of different ethnicities aged 5–60 years and involved the examination of 374 healthy eyes [439]. The excitation was with a 488 nm laser and emission was collected from 500–680 nm. A fluorescence standard was incorporated into the imaging instrument to allow for image analysis to compensate for changes in the laser power and detector gain. To account for the age-related changes in the transmission of the anterior segment of the eye, the normative data of van de Kraats and van Norren were used [406]. The age-related increase in fundus fluorescence was the greatest for Whites, followed by Indigenous Americans, Hispanics, Blacks, and Asians.

Ach et al. analysed fluorescence from RPE-Bruch’s membrane flat-mounts from 20 donors divided into two age groups: one with 10 donors from 16 to 51 years of age (average age of 40 years), and another with 10 donors from 82 to 90 years of age (average age of 85 years) [435]. The fluorescence was excited by light from a mercury arc lamp and filter transmitting 460–490 nm wavelengths. The emission was collected above 505 nm. The emission intensity significantly increased in the elderly group in comparison with the younger group by 51% in the central fovea (a circle centred in the fovea with a 0.6 mm radius), by 45% in the areas of parafovea, perifovea, and perimacula (0.601–3 mm from the foveal centre), and by 56% in the periphery (more than 3 mm from the centre of the fovea), respectively. These differences between the two age groups are similar to those in measurements in vivo by Delori et al. using similar excitation and emission wavelengths in age groups of about 55 and 80 years of age, where the intensity of fluorescence in the eldest group is about 60% greater than in the 55-year-old group [434].

In a follow-up study to the studies of Ach and Smith et al. [381,435] described above and in Section 6.2 on 20 flat-mounts from human RPE-Bruch’s membrane complexes, the data from 6 eyes were excluded due to the loss of RPE cells and/or the inability of the mathematical algorithm to recover sufficient spectra for the fitting procedure, and the analysis was performed on tissues from 14 donors ranging in age from 16 to 90 years [440]. The acquisition of hyperspectral data and processing was the same as described in their previous report with the excitation wavelengths of either 436 or 480 nm and emission collected using a hyperspectral camera collecting emission from the range of 410–720 nm passing through the long-pass emission filters transmitting light above 460 or 510 nm, respectively [381]. The data were analysed based on dividing the flat-mounts into two age groups: below 51 years of age and above 80 years of age, with seven donors in each of them [440]. Interestingly, there were no significant differences between the wavelengths at which maxima of fitted individual curves appeared for the two age groups. The comparison of the relative contributions of the fitted curves to the overall fluorescence was not the subject of the analysis; nevertheless, the two examples, where the graphs were provided, show that spectra with the emission maximum at about 570 nm dominate in the emission in a flat-mount from 16-year old, whereas in the 36-year-old, the dominant spectrum exhibits a maximum at about 520–530 nm [440]. It would be of great interest to see if there is an age-dependent change in the contributions of spectra with the short- and long-wavelength maxima.

The evaluation of the emission maxima as a function of age in vivo was conducted by Schultz et al. who compared the fluorescence emission maxima in 44 persons below 40 years of age (average age of 24 years) and 18 persons above 40 years of age (average age of 67.5 years) [441]. The fluorescence was excited by a picosecond laser diode emitting 473 nm wavelength with a repetition rate of 80 MHz and it was coupled with a commercial laser scanning ophthalmoscope. Fluorescence emission was collected from each pixel by the time-correlated single photon counting in two spectral channels: green (498–560 nm) and yellow-red (560–720 nm). The ratio of photons collected in each channel was calculated and the emission maximum was evaluated based on the calibration curve [442]. The calibration was prepared by measuring the fluorescence emission ratio from both channels of healthy human fundus and lens in a cohort of healthy volunteers and plotting it against the actual emission maxima from the human fundus and lens published by Delori [379] and Zuclich [443], respectively [442]. The emission peaks were evaluated for the fovea, and the inner and outer rings surrounding the fovea in a standard grid [441]. For the young group, the emission maxima were at 602 ± 16, 614 ± 12, and 621 ± 11 nm for the 1 mm in diameter area inside the fovea, inner ring (1–3 mm in diameter) and outer ring (3–6 mm in diameter), respectively. For the elderly group, the emission maxima were at 599 ± 17, 611 ± 11, and 614 ± 11, respectively. While it appears that there is a trend of slight decreases with age in the wavelength corresponding to the fluorescence maxima in all three locations, the differences are not statistically significant. Based on spectral changes in lipofuscin fluorescence upon photodegradation, it can be expected that a shorter excitation wavelength and monitoring short-wavelength emission in about 470–490 nm spectral range is more likely to show a statistically significant age-related decrease in the calculated emission maxima.

Altogether, there is an age-related increase in fundus fluorescence intensity at least to the age of 60 years and is most pronounced in people of White ethnicity [385,434,438,439] (Table 2). Emission in the blue range 460–480 nm shows an age-related increase even in the age range of 60 to 88 years [181]. At least three studies demonstrated that the emission collected >520 nm reaches a plateau above the age of 60 years [385,434,438]. Fluorescence emission excited by 550 nm light and collected in the red range of 650–750 nm exhibits an age-related increase up to the age of 70 years followed by a steep decrease [434]. Therefore, it appears that at old age there is a decrease in fluorophores emitting long-wavelength fluorescence and an increase in fluorescence from fluorophores emitting short-wavelength light [181,182,434,441].

**Table 2 antioxidants-12-02111-t002:** Studies of the relationship of fundus fluorescence with age.

Study on Normal Human Eyes In Vivo and Ex Vivo	Excitation (nm)	Emission (nm)	Age-Related Changes in Fluorescence Intensity or Spectra
Cross-sections from eyes of 19 White donors, **2 weeks–88 years of age**, and 19 Black donors, **6.5–90 years old**; 5 sites per eye: fovea, parafovea (half a distance from the fovea to the disc and on the other side, two equatorial sites [182]	365	470	Age-related increase for Whites No correlation with age for Blacks
Cross-sections from 44 human eyes from 35 donors, **6-week premature newborn–88 years**; from ora serrata via optic disc and fovea to ora serrata on opposite site; lipofuscin from all the length of RPE was quantified in 29 eyes [181]	380	460–480	A fast increase in the first and second decades of life, then slowing down followed by **an increase in people above the age of 60 years**; about a 40% increase in fluorescence emission intensity in the oldest age group 61–88 in comparison with 31–60 years group
30 participants, **21–67 years of age**; excitation area of 3° in diameter; fluorescence measured at the fovea and at 7° temporal to the fovea from an area of 2° in diameter [379]	430	620	No significant correlation with age
Sections from formalin-fixed 8 mm in diameter circles centred on the fovea of 88 donors ranging in age from **1–98 years** [438]	450–490	>520	A linear **increase up to the age of 60 years, followed by a plateau**; supported by TEM quantification of lipofuscin
RPE-Bruch’s membrane flat-mounts about 20 × 20 mm including optic disc and macula from 20 donors divided into two age groups: **16–51 years of age** (10 donors, average age of 40 years), and **82–90 years of age** (10 donors, average age of 85 years) [435]	460–490	>505	Increased in the 82–90 year-old group in comparison with the 16–51 year-old group
145 participants, **15–80 years of age**; retinal field of 13° circle centred on the fovea and quantified at the fovea and at 7° eccentricity temporal to the fovea; individually corrected for the absorption of light by the lens [434]	470	>520	Intensities reached **a maximum for the age group in their 7th decade and remained at the same level in the 8th decade**
33 White participants **6–78 years of age**; fluorescence imaged over 40° field-of-view and quantified at the fovea and at the site of maximum intensity 7–15° of eccentricity [385]	488	>521	A linear increase with age from 6 to about 60 years, **above 60 the emission appears to plateau**
277 participants of different ethnicities from **5–60 years of age**; fluorescence imaged over 30° × 30° and quantified in a ring at about 8.4° of eccentricity [439]	488	500–680	The age-related increase in fundus fluorescence was the greatest for Whites, followed by Indogenous Americans, Hispanics, Blacks, and Asians
53 White participants, **5–18 years of age** [437] and 103 White participants, **18–77 years of age** [436]; fluorescence imaged over 30° × 30° area centred on the fovea	488	500–750	Overall a monotonic increase with age in the fovea and extrafoveal circle extending to the optic nerve head, with an initial rapid linear increase up to the age of 20 years, possibly reaching a plateau around the age of 60 and further increase after the age of 65
30 participants, **21–67 years of age**; other details as for excitation with 430 nm [379]	470, 510 or 550	620	Positive correlation with age
145 participants, **15–80 years of age**; other details as for excitation with 470 nm [434]	550	650–750	A linear increase in fluorescence occurred up to the age of **70 years, followed by a steep decrease**
44 participants **below 40 years of age** (average age of 24 years) and 18 participants **above 40 years of age** (average age of 67.5 years); calculated emission maxima based on emission of fluorescence in two spectral channels [441]	473	498–560 and 560–720	For the younger group, the emission maxima were at 602 ± 16, 614 ± 12, and 621 ± 11 nm for the fovea (1 mm in diameter), inner ring (1–3 mm in diameter) and outer ring (3–6 mm in diameter), respectively. For the older group the emission maxima were at 599 ± 17, 611 ± 11, and 614 ± 11, respectively

### 6.4. Fundus Autofluorescence in Age-Related Macular Degeneration (AMD)

AMD is the leading cause of vision loss in people above the age of 50 in developed countries with an estimated prevalence of 288 million worldwide by 2040 [332,444,445,446,447]. Multiple factors are involved in its development and progression including the genetic component, cardiovascular risk factors, and lifestyle. AMD can be classified into several stages with an early stage considered when drusen appear of medium size, between 63 and 125 µm, but there are no pigmentary abnormalities in the retina [447]. Intermediate AMD is diagnosed when there are drusen larger than 125 µm and/or pigmentary abnormalities. These stages of AMD are predictors for its progress into late forms, which affect vision: (i) dry (atrophic) form, characterized by loss of RPE and photoreceptors; (ii) wet (neovascular) form, characterized by the growth of abnormal blood vessels from the choroid into the retina, which can leak fluid and other blood components (exudates); or (iii) a combination of both. More recently, reticular pseudodrusen has been identified as another risk factor for progression to late AMD, particularly the atrophic form [448]. The atrophy initially appears in a small retinal area and is known as geographic atrophy, which expands as the disease progresses. There are several studies investigating the effect of AMD on fundus autofluorescence ex vivo and in vivo and using the findings to monitor the progression of the disease in vivo.

#### 6.4.1. Sources of Fluorescence in the AMD Retina Examined Ex Vivo

Marmorstein et al. compared fluorescence properties of nonfixed cryosections from human retina-choroid-sclera explants isolated post-mortem from three normal eyes (age 81, 83 and 87 years) and three eyes with AMD (age 74, 84, and 92 years) [375]. All AMD eyes had extensive drusen and one eye had choroidal neovascularization. The cryosections were cut from 2 mm wide strips centred on the fovea. Like in cryosection from normal eyes, excitation with 568 and 633 nm laser of cryosections from AMD eyes induced fluorescence in the RPE, Bruch’s membrane, and sub-RPE deposits (drusen, basal linear deposis, and basal laminar deposits); however, the ratio of fluorescence from Bruch’s membrane and sub-RPE deposits to the RPE fluorescence was about twice greater than in normal eyes. For the excitation with 488 nm, the intensity maximum of Bruch’s membrane fluorescence was similar to that of the RPE and about twice greater than for sub-RPE deposits.

Upon excitation with 364 nm light of AMD cryosections, Bruch’s membrane/sub-RPE deposits emitted strong fluorescence with intensity maxima at similar wavelengths of 485–490 nm as for normal retinas [375]. However, in contrast to normal eyes, where the intensity maxima were similar for Bruch’s membrane and RPE, in AMD eyes, the intensity maximum was about 1.8-fold greater for Bruch’s membrane than for RPE. The emission spectrum for Bruch’s membrane was similar in shape but the emission maximum was about twice greater than for sub-RPE deposits. The emission maximum of the RPE fluorescence induced by excitation with 364 nm was at 540 nm, which is 15 nm shifted towards shorter wavelengths in comparison with the normal retina.

The number of tissue donors was small, yet the images of fluorescence excited by 488 nm light in the cryosections showed strikingly greater fluorescence emission at 545nm, 555 nm, and 605 in the RPE and sub-RPE deposits in AMD eyes than in normal eyes [375]. This difference between AMD and normal eyes was even more apparent when fluorescence was excited by 364 nm light and emission was monitored at 485, 555, 625, and 695 nm.

Rudolf et al. examined sections of the AMD retinas with geographic atrophy from 10 donor eyes, ranging in age from 73–93 years (average age of 87 years), and demonstrated that the areas adjacent to the geographic atrophy exhibit nonuniform fluorescence in the RPE monolayer with some cells exhibiting substantially decreased fluorescence excited by 488 nm laser with emission collected after passing via filter transmitting light above 500 nm [449]. They also showed some fluorescent cells present in the neural retina and RPE cells stacked one on top of another in the areas adjacent to geographic atrophy areas. This stacking of cells can explain the increased fluorescence seen at the edges of geographic atrophy when imaged en face [450]. Moreover, Zanzottera et al. showed that edges of the geographic atrophy contain RPE cells or their parts in the neural retina, thick basal laminar deposits, and some flattened RPE cells adjacent to Bruch’s membrane, which are separated from the photoreceptors by basal laminar deposits [450].

Bonilha et al. examined fundus fluorescence of 32 eyes from 17 donors, 65–97 years of age, with atrophic AMD and 10 eyes from 10 donors with no eye disease, 57–95 years of age, which was followed by analyses of histologic sections in the areas of geographic atrophy [451]. The fundus fluorescence was imaged upon excitation with 488 nm and 787 nm lasers from a commercial laser scanning ophthalmoscope. The fluorescence of histological sections was acquired with filters with excitation and transmission properties close to those of typical clinical instruments: the emission of fluorescence excited with 480 nm light was collected in the range of 500–680 nm; whereas the emission of fluorescence excited by 750 nm light was collected above 800 nm. The RPE present away from the border of geographic atrophy had nonfluorescent melanosomes present in the apical portions of the cells, a high concentration of fluorescent granules excited by blue and near-infrared light and appeared similar to the RPE in the age-matched control eyes. Close to the atrophy border, the RPE cells were hypertrophic and filled with granules emitting in the green-yellow-red spectral range as well as in the near-infrared. In those cells, the infrequent nonfluorescent melanosomes were dispersed throughout the cell. The morphology and pigmentation of the RPE close to the border were frequently nonuniform. Next to the intact RPE, cells appeared sloughed into the subretinal area or aggregates of granules shed into the basal laminar deposit. Some RPE cells appeared stacked into a bilayer or appeared as individual cells with no contact with other RPE cells or even translocated into the neural retina.

In a series of studies of human RPE flat-mounts from AMD and normal eyes examined by SIM and confocal fluorescence microscopy, Bermond et al. demonstrated that AMD-affected RPE cells show an increased variability in size and a significantly increased granule load, and this can be seen in all retinal locations examined: the fovea, perifovea, and near periphery [189,358,382]. In comparison with normal retinas, the relative contribution of lipofuscin to all pigment granules decreases in the AMD retina, whereas the melanolipofuscin contribution increases, particularly in the fovea. AMD-affected RPE cells exhibit decreased fluorescence collected within the range of 510–750 nm when excited by 488 nm laser in comparison with normal RPE [189]. These findings can explain the results of quantitative measurements of fundus autofluorescence intensity in AMD patients and healthy controls, which revealed that there is lower overall fluorescence in AMD retina induced by photoexcitation with 488 nm laser [188]. This effect was pronounced particularly in patients with reticular pseudodrusen, which are lipid-rich deposits between POS and RPE.

In another study where the imaging of fluorescence was performed en face on RPE-Bruch’s membrane-choroid flat-mounts, it has been observed that RPE cells from aged or AMD retinas sequester fluorescent granules–lipofuscin and melanolipofuscin into packets, 2.5 to 20.9 µm in diameter leaving some cytoplasmic areas, or even entire RPE cells, almost devoid of fluorescent granules [20]. Cross-sections of these flat mounts revealed that some cells form mushroom cap-like structures filled with lipofuscin, which protrude into the area of POS. Another study showed lipofuscin/melanolipofuscin granules shed into basal laminar deposits [450]. It has been proposed that the removal of the lipofuscin-laden packets from cells is a mechanism of control of intracellular levels of lipofuscin but it may contribute to the sub-RPE deposits [20,449,450,452,453]. Interestingly, similar packets with lipofuscin were observed in vitro above lipofuscin-laden confluent ARPE-19 cell monolayer after exposure to visible light, which caused spectral changes in lipofuscin fluorescence [75].

In another study where the fluorescence imaging of RPE-Bruch’s membrane was performed en face on flat mounts, five AMD eyes were used: one with early AMD, one with geographic atrophy, and three with exudative AMD [454]. Twelve areas with drusen were imaged in total. Out of the total of 25 drusen analysed, 12 were small (≤63 μm in diameter), eight were intermediate (64–124 μm), and five were large (≥125 μm). As before [435], the explants were excited with 436 or 480 nm and emission filters were transmitting light above 460 nm and 510 nm, respectively. The hyperspectral camera was used for image acquisition every 10 nm from 420–720 nm. The fluorescence of the RPE from AMD eyes was similar to that from normal eyes. Using the same fitting procedure as the one used previously on explants from normal eyes, a new spectrum specific for drusen has been identified. With the excitation of 436 nm, the emission maxima of that spectrum were at 510 nm for 10 drusen, one druse exhibited a maximum at 500 nm, and one druse at 490 nm. Excitation with 480 nm led to a less intense emission with the maximum at 520 nm for the majority of drusen, followed by some drusen emitting with the maximum at 540 nm and one druse at 530 nm. Other sub-RPE deposits (basal laminar and basal linear deposits) emitted similar fluorescence as drusen.

Subsequent studies by Mohammed, Tong, and colleagues imaged hyperspectral autofluorescence of RPE-Bruch’s membrane-choroid explants in pre-selected retinal locations (fovea, perifovea, mid-periphery) as well as in all identified drusen [396,455]. The explants were obtained post-mortem from AMD patients 81–88 years of age. For the excitation, they used four dichroic filters transmitting narrow-band light from a mercury arc lamp centred at 436, 450, 480, or 505 nm. The emission spectra were collected from each pixel by the hyperspectral camera as in their previous study described above [454]. Using their mathematical fitting procedure, they showed that contribution from drusen fluorescence dominated the emission in AMD RPE-Bruch’s membrane flat-mounts excited by 436 nm, and its contribution decreased as the excitation wavelength increased [396,455]. This approach allowed for even better differentiation of drusen and sub-RPE deposits than the one based on the use of only two excitation wavelengths. Importantly, the authors showed that excitation with 450 nm light provides similar spectral characteristics of the drusen spectrum as that obtained with 436 nm excitation. This is an important finding because using a longer excitation wavelength is safer for the retina in vivo considering the light-induced reactivity of retinaldehydes, flavins, porphyrins and lipofuscin [6,81,94,106,108,240,456,457,458].

Clearly, there are distinct changes in the AMD retina in comparison with the age-matched control retina, which affect the fluorescence. The contribution from Bruch’s membrane and sub-RPE deposits to retinal fluorescence increases in AMD; there are changes in the distribution of pigment granules in RPE cells, with an overall increase in fluorescent granule concentration, but with decreased concentrations of lipofuscin and increased melanolipofuscin, which results in a decreased fluorescence excited by 488 nm light in the affected RPE cells. In the AMD retina, some RPE cells are hypertrophic, cells/cell fragments filled with fluorescent granules are present in the neural retina, while some sub-RPE deposits include fluorescent granules. Some RPE cells are stacked on other RPE cells, while in the areas of atrophy, there is a complete absence of RPE cells.

#### 6.4.2. Fluorescence Characteristics of AMD Retina In Vivo

It has been observed already in the initial studies of fundus autofluorescence, that the emission spectrum in areas with drusen is shifted toward shorter wavelengths [441,459,460,461]. Hammer et al. analysed 43 eyes of 39 White patients with AMD characterized by drusen and absence of choroidal neovascularization or atrophy [461]. As mentioned earlier, pigmentary changes and drusen, which are visible in a typical fundus examination and recorded with colour fundus camera, are considered as early or intermediary stages of AMD [447]. Consistently with the findings on RPE-Bruch’s membrane explants and histological sections of the retina described above, they demonstrated that drusen and hyperpigmentation were associated with shorter emission wavelengths for fundus autofluorescence. Retinal areas with reticular pseudodrusen (also known as subretinal drusenoid deposits, SDD) as determined by OCT also exhibit shorter peak emission wavelengths than the surrounding areas [462].

Borrelli et al.’s study on 25 eyes of 18 patients diagnosed with an atrophic form of AMD has shown that some areas devoid of RPE (as seen by OCT) exhibit substantial green fluorescence when excited with 450 nm light [391,463]. From the OCT images included in their paper, it appears that these green-fluorescent areas correspond to areas of severely damaged photoreceptors and absent RPE: there is no characteristic hypo-reflective band corresponding to the outer nuclear layer and no characteristic hyper-reflective bands corresponding to the outer limiting membrane and the ellipsoid area of the inner segment. Some of the areas of increased green fluorescence appear in OCT as hyper-reflective in locations corresponding to the inner retinal layers: the inner nuclear layer or even the inner plexiform layer; some other areas are hyper-reflective at the area corresponding to the nonexisting outer nuclear layer through RPE. It has been concluded that the residual debris overlying Bruch’s membrane are the origins of the green-emitting fluorophores.

Several studies compared the emission spectra in normal and AMD eyes in vivo including one that demonstrated a statistically significant shift in the emission maximum towards shorter wavelength [376,441,459]. Schultz et al. compared the wavelength of the fluorescence emission maxima in 18 persons above 40 years of age (average age of 67.5 years) and 63 patients with early AMD (average age of 74 years) [441]. As described before for the comparison of emission maxima in two age groups, the fluorescence was excited by a picosecond laser diode emitting 473 nm wavelength, the fluorescence emission was collected in two spectral channels: green (498–560 nm) and yellow-red (560–720 nm), and the calibration was used to determine the emission maxima [442]. AMD retinas exhibited maxima at 571 ± 26, 596 ± 17, and 602 ± 16 nm for the fovea, inner and outer ring of a standard grid, respectively [441]. These values are 28, 15, and 12 nm shifted towards shorter wavelengths in comparison with the emission maxima of the healthy retinas at the corresponding locations, and the differences are statistically significant.

In another study by the same group, fundus fluorescence lifetimes and emission intensities in the two spectral channels were examined on three patients with an atrophic form of AMD [398]. It has been determined that the area of complete RPE and outer retinal atrophy (cRORA) had a stronger emission in the green (498–560 nm) channel than in yellow-red (560–720 nm) channel, with a particularly strong emission from drusen. Some drusen were surrounded by hypofluorescent areas. Areas with drusen without overlying RPE exhibited longer fluorescence lifetimes than other areas of cRORA. This is consistent with the previous studies by Hammer, Schweitzer et al., demonstrating that fluorescence lifetimes in AMD patients are longer than in controls [464,465]. In these studies, the fluorescence was excited with a 448 nm laser, and emission was monitored in two spectral ranges: 490–560 nm and 560–700 nm.

#### 6.4.3. Current Evidence for the Prognostic Value of Fundus Fluorescence Characteristics for AMD Progression

The characteristics of fundus fluorescence as a potential predictive biomarker for AMD progression have been investigated in several studies. A longitudinal study of 121 eyes of 71 patients with intermediate AMD followed over a median period of 15 months (interquartile range of 9–21 months) has demonstrated that a decrease in autofluorescence in a standardized ring surrounding the fovea precedes the progression to the atrophic, but not to the neovascular, form of AMD [466]. The quantification of fundus fluorescence was performed with excitation with a 488 nm laser and the emission was collected between 500 and 680 nm using a commercial instrument combining SLO and OCT. Another study from the same group and using the same instrument demonstrated on 43 eyes of 26 patients with atrophic AMD that there is a positive correlation between the progression of the geographic atrophy area over a period of 12 months and the intensity of fluorescence at the junctional zone in the inferior retina but not elsewhere [467]. The junctional zone was defined by the area of the retina adjacent to the area of geographic atrophy, which had still a distinguishable ellipsoid layer in OCT images. Interestingly, the expansion of geographic atrophy occurred towards the outside areas with decreased fluorescence and this effect was statistically significant also only for the inferior retina.

The prognostic value of fluorescence lifetimes in the area of large soft hyperfluorescent drusen with overlying RPE has been demonstrated by Schwanengel et al. in a study of 40 eyes from 38 patients with intermediate AMD examined by colour fundus camera, OCT, and fluorescence excited by 473 nm laser and emission collected in two spectral channels: green (498–560 nm) and yellow-red (560–720 nm) [442,468] (Figure 11). Over the follow-up of up to 6 years, the retinas, which progressed to choroidal neovascularisation or cRORA, exhibited longer fluorescence lifetimes of the drusen areas at baseline than the retinas with no disease progression. These differences were statistically significant when monitored in either green or yellow-red channels.

The peak emission wavelength of the drusen at baseline was shorter in eyes that exhibited AMD progression than in eyes with no progression, but this difference was not statistically significant [468]. In another study by the same group, 39 AMD patients were followed for up to 6 years to determine the progression of the disease. During the follow-up visits, a shortening of the peak emission wavelength was observed and reached statistical significance in the fovea and inner ring in months 37–72 of the follow-up [442]. Interestingly, 10 AMD patients who progressed to the outer retina atrophy or to cRORA in the follow-up had the fluorescence emission maxima at significantly shorter wavelengths in all three retinal locations, the centre, inner and outer rings of a standard grid, than patients who did not progress in their disease.

It could be argued that the spectral shift of the emission maximum to shorter wavelengths is due to SDD, drusen, and other deposits, such as basal linear deposit, which accumulate in the Bruch’ membrane and are already known as prognostic factors for AMD progression [469,470]. While the contribution from such sub-RPE deposits can indeed contribute to the observed decrease in a wavelength corresponding to the fluorescence emission maximum, there is some evidence that this decrease is due to spectral changes in fluorescence of the RPE [471,472,473,474]. Feldman et al. reported fluorescence spectra of suspensions of macular RPE cells from two normal eyes and two AMD eyes [471]. All four samples showed broad emission upon excitation with 430 nm, overall similar in shape, but with a hypsochromic shift in the emission maxima from AMD-affected RPE. Normal RPE exhibited maxima at 563 and 564 nm, whereas the maxima for the RPE from AMD eyes were at 528 and 540 nm [471]. A follow-up study using excitation with 488 nm laser of RPE cell suspensions from 19 normal eyes (age 27–74 years) and 12 AMD (age 59–88 years) demonstrated substantial spectral differences between the emission spectra of macular RPE from normal and AMD-affected eyes [472,473,474]. There was a clear emission maximum at 592 nm for normal RPE. RPE from eyes with AMD showed a flat maximum extending from about 550 to 590 nm. The ratios of integrated green-yellow fluorescence (530–580 nm) to orange-red fluorescence (600–650 nm) were 1.04 ± 0.11 and 1.73 ± 0.16 for RPE from normal and AMD eyes, respectively.

## 7. Retinal Spectral Fluorescence Characteristics as a Potential In Vivo Biomarker of Oxidative Damage and Efficacy of Potential Antioxidant Therapies

### 7.1. Current Evidence for Photooxidation of Lipofuscin In Vivo

As discussed in the previous sections, lipofuscin is the major contributor to fundus fluorescence excited by UVA and visible light, but photooxidation of lipofuscin in vitro leads to spectral changes in its emission spectrum, namely a decrease in its long-wavelength yellow-orange fluorescence and an increase in its blue-cyan fluorescence.

While exposure to light can accelerate lipofuscin accumulation in the retina and its fluorescence, there is a growing body of evidence suggesting that photooxidation can affect fluorescent components of lipofuscin in animal and human retinas also in vivo and, as a consequence, can lead to a decrease in lipofuscin fluorescence. For example, raising pigmented wild-type and *abca4-/-* knockout mice under cyclic light initially, up to 18 weeks, accelerates the accumulation of A2E in comparison to mice raised in the dark, with the knockout mice having 12-fold greater levels of A2E than their wild-type counterparts [230,475]. However, this difference becomes smaller over longer rearing times and depends on the intensity of light the animals are exposed to [230,475]. This is particularly apparent in albino mice: 7-month-old albino *abca4-/-* mice raised under 30 lx cyclic light exhibit similar levels of A2E as pigmented counterparts but have about 30% greater levels of A2E oxidized to oxiranes [230]. The level of A2E-derived oxiranes in pigmented 7-month-old *abca4-/-* mice is almost two-fold greater in mice raised under 120 lx than under 30 lx, while the levels of intact A2E are similar.

In another study, it has been shown that the fundus fluorescence excited with 488 nm laser of 9-month-old albino BALB/cJ mice decreases by about 20% after seven daily two-hour exposures to 430 ± 30 nm light from a 1 mW halogen source [476]. The decrease in fluorescence emission is accompanied by about an 18% decrease in A2E extracted from the eyecups. A similar decrease in A2E of about 24% has been also observed in 8-month-old albino C57BL/CJ^c2j^ mice subjected to the same light exposure protocol in comparison with not-exposed control mice. A comparison of A2E levels in pigmented C57BL/CJ and albino C57BL/CJ^c2j^ mice raised in dark or under cyclic light (40–60 lx from white LEDs) has shown that the six-month-old albino mice raised under cyclic light have almost twice lower levels of A2E than their counterparts raised in the dark whereas the difference between pigmented mice raised under cyclic light or in dark is smaller than 10%. It has been also shown that the levels of A2E and other bisretinoids are several-fold smaller in albino *abca4-/-* knockout mice raised for 8 months under cyclic light than in mice raised in the dark but this light-dependent decrease, together with a decrease in fundus fluorescence, can be partly prevented by supplementing the animals with vitamin E. It has been shown in a follow-up study that daily administration of an iron chelator deferiprone from the age of 2 to 6 months, can partly inhibit loss of bisretinoids and fluorescence decrease in albino *abca4-/-* knockout mice and bisretinoids loss in agouti 129/SvJ mice [477].

A rearing light-dependent fluorescence decrease has been also observed by Pan and colleagues in pigmented mice, both wild type and mice with deleted genes coding for two proteins involved in clearance of retinaldehydes, *abca4(-/-)rdh8(-/-)* [17]. The double knockout mice raised under a cyclic 10 lx light for 12 months exhibit a 2.8-fold lower intensity of RPE fluorescence emission in comparison to mice of the same genetic background raised in the dark, but it is not clear what the spectral range of the emission was measured because the excitation and emission details have not been provided.

Moreover, it has been shown that RPE fluorescence in the macaque retina in vivo decreases by about 20% as a result of a single radiant exposure of 210 J/cm^2^ to the light of 568 nm wavelength with no indication of any damage to the RPE monolayer [263,264,265]. It has been also demonstrated that exposure of fixed human RPE ex vivo to a much smaller dose of only 30.6 J/cm^2^ of 568 nm light leads to fluorescence photobleaching. The decrease in fluorescence in the macaque retina in vivo without damaging the RPE has been also observed after exposure to 460, 488, 544, and 594 nm lasers [266].

There are reports on changes in retinal fluorescence dependent on light exposure in human retinas as well. It has been shown that translocation of the reattached neural retina after detachment reveals areas of increased fluorescence matching the shape of blood vessels, which, after the translocation, no longer prevent the incident light from reaching the RPE behind them suggesting that the fluorescence in the other areas has been partially bleached by exposure to light [478,479].

As mentioned previously, Teussing et al. studied the effects of wearing a black contact lens in one eye of Stargardt’s disease patients during waking hours and compared it with the unprotected eye [268]. They imaged fundus fluorescence excited by a 488 nm laser and have shown that the average increase in the percentage of hypofluorescent pixels in the covered eye was 2.5%/year whereas the average increase in the unprotected eye was 8.1%/year, which is 3.2-fold greater.

### 7.2. Age-Related Changes of A2E Content in the Macula and Periphery

Bhosale et al. reported age-related changes in A2E content in the macular areas and mid-periphery in monkey and human retinas [175]. They determined that young monkeys *Macaca fascicularis*, 2 ± 2 years of age, have higher concentrations of A2E in the macula than in the mid-periphery: 75 ± 5 ng A2E per g of wet RPE/choroid extracted from 8 mm in diameter circular trephine punch from the macular area, and 53 ± 85 ng A2E from the mid-peripheral area (not specified exactly where) [175]. At this young age, the macular area contains a 1.4-fold higher concentration of A2E than the mid-periphery suggesting that A2E accumulates faster in the macula than in the mid-periphery.

At 9 years of age, the A2E content increases to 86 ± 15 and 93 ± 85 ng in macular and peripheral punches, respectively. This corresponds to a 15% increase in A2E in the macula and a 75% increase in the mid-periphery, and shows that the accumulation of A2E slows down with age and/or A2E accumulation is accompanied by its degradation. Clearly, these effects are more pronounced in the macula than in the mid-periphery.

The slowing down of A2E accumulation could be due to slower accumulation of lipofuscin once lenticular chromophores are formed and strongly absorb UV and partly blue light reaching the retina. Such chromophores are already present in the lenses of a 2-year-old macaque and, at similar levels, in a 12-year-old macaque [480] suggesting that light-induced formation of lipofuscin and A2E could be similar at the ages of 2 and 9 years.

In 9-year-old monkeys, the A2E content is already greater in the mid-periphery than in the macula [175]. The ratios of A2E content in the mid-periphery to macular A2E increase from 0.7 to 1.1 for 2-year-old and 9-year-old monkeys, respectively.

Interestingly, RPE/choroid tissue from human cadavers, 23 ± 5 years of age, contains only 26 ± 12 ng of A2E per gram of wet weight in the macular punch of 8 mm in diameter, and 102 ± 16 ng A2E in the mid-peripheral punch, which means a 3.9-fold decrease in A2E between the mid-periphery and macula already in young adulthood [175]. The ratio of A2E in the mid-peripheral and macular punches in the age group of individuals 60 ± 6years is 3.4 (53 ± 13 ng of A2E in the macular punch and 182 ± 23 ng of A2E in the mid-peripheral punch).

Another set of data presented by Bhosale et al. includes an evaluation of A2E and iso-A2E in the macular and mid-peripheral punches not normalized to the wet weight and also shows their smaller levels of 1.4 and 5.6 ng in the macula and periphery, respectively, in the age group of 37 ± 10 years than 3.5 and 10.5 ng, respectively, in the age group of 74 ± 9 years [175]. These data are consistent with data on A2E per gram of wet weight by demonstrating that ratios of A2E and its isomer in the mid-peripheral to macular area decrease with age from 4.0 to 3.0 in these two age groups.

These results suggest that there is faster accumulation and degradation of A2E in the macula than in the periphery possibly because the central retina is exposed to higher intensities of light than the peripheral retina.

The differences in A2E content between the macula and periphery do not reflect the distribution of lipofuscin granules reported by Feeney-Burns [77] and described in Section 4.5.1, which shows much smaller differences between the macula and equator. It appears that A2E is very susceptible to degradation, particularly in the macula.

It appears that the very early oxidation products of A2E formed by the addition of one or two oxygen atoms resulting in monofuranoid oxide of A2E and presumably 7,8,7′,8′-bis-epoxide track the distribution of A2E [132,134,175,235]. Ben Shabat et al. and Gaillard et al. [130,228,233] reported various other A2E photooxidation resulting from the sequential addition of more oxygen atoms to A2E and the formation of various epoxides, including nonaepoxide. Gaillard group also reported A2E photooxidation and autooxidation leading to the formation of various scission products, including reactive carbonyls, bis-furanoid oxide of A2E, mono-furanoid oxide of A2E with the second oxygen addition to the cyclohexynyl ring, as well as high-molecular-weight aggregates [232,234,238,239]. The spatial distribution in the retina of these products has not been reported. Due to increased exposure to light, the environment of the central retina is more oxidizing than in the peripheral retina, so it can be expected that the central retina will be depleted from A2E and its early oxidation products faster than the peripheral retina.

Because A2E is believed to be the major emitter of yellow-orange fluorescence in lipofuscin, it can be expected that its degradation can affect the topographical distribution of that fluorescence with age.

### 7.3. Are There Spectral Changes in Retinal Fluorescence with Age?

As the retina ages, the cumulative exposure to light increases, and, therefore, it can be expected that at a certain age, the effects on fluorescence of oxidative degradation of lipofuscin can overshadow the effects of accumulation of fresh lipofuscin. Indeed, changes in fundus fluorescence consistent with oxidative degradation of lipofuscin have been observed in studies investigating the effect of age on the fluorescence of the human fundus (Table 2). For fluorescence excited with UV light (365 or 380 nm) and emission detected in the blue range of the visible spectrum (at about 470 nm), there is an age-related increase in the emission intensity even in the eldest eyes studied, at the age of 88 or 90 years [181,182]. In studies where fluorescence was excited by long-wavelength blue light (470–490 nm) and the emission was collected at above 505 or 520 nm, an age-related increase in fluorescence was observed until the sixth or seventh decade when it appeared to plateau [385,434,438]. The long-wavelength fluorescence emission collected in the range of 650–750 nm exhibits a linear increase up to the age of 70 years followed by a steep decrease after that age [434]. These studies on age-dependence of fluorescence intensity have not assessed the spectral characteristics of the emission so it is not clear whether the decreased emission was due to removal of lipofuscin, spectral changes in the emission spectra, or both. However, the study by Delori showing that there is an age-dependent decrease in fluorescence emission in the range of 650–750 nm between the age of 70 and 80 years, while the same group of individuals exhibited a plateau of emission collected above 520 nm suggest that spectral changes in the fluorescence emission with age do occur [434]. Schultz and colleagues also demonstrated a small spectral shift of the emission maximum towards shorter wavelengths in a group of 18 subjects above the age of 40 years in comparison with a group of 44 subjects below the age of 40 but the differences did not reach statistical significance [441]. This study was based on collecting emission in two spectral channels 498–560 nm and 560–720 nm and the fluorescence was excited by 470 nm light.

Based on the spectral changes observed during photodegradation of RPE lipofuscin, it can be suggested that by using a shorter excitation wavelength, such as 460 nm, and collecting emissions in the short wavelength channel where the fluorescence of degraded lipofuscin is increased, such as 470–490 nm (Figure 8) [75], the age-related differences could be more pronounced.

### 7.4. Current Evidence for Increased Oxidative Stress and Oxidative Damage in AMD or Stargardt’s Retina

It has been shown that the AMD retina is under increased oxidative stress in comparison with the age-matched retina and there is a growing body of evidence suggesting that the oxidative stress and products of lipid and bisretinoid oxidation may contribute to the development and progression of AMD by activation of the complement cascade, and other proinflammatory and proangiogenic signalling (reviewed in [173,481,482,483,484,485,486]). Firstly, RPE affected by early, atrophic, and exudative AMD exhibit about five-fold increased content of total iron and iron chelateable by desferrioxamine in comparison with RPE in age-matched normal retinas [487]. It has been shown in experiments on mice that the retinal iron overload, caused by intravitreal injection of ferric ammonium citrate into the eye, results in (i) the modification of proteins by a characteristic product of DHA oxidation forming CEP; (ii) the upregulation of interleukin-1β (IL-1β) and cluster of differentiation 68 (CD68) expression; and (iii) the photoreceptor and RPE atrophy [100,488]. Inhibiting lipid peroxidation by supplementing animals with a form of DHA where the hydrogen atoms are replaced by deuteriums, thereby making it substantially less susceptible to the propagation of lipid peroxidation than normal DHA, can effectively prevent the CEP formation, upregulation of both inflammatory markers and cell loss [488,489,490,491,492]. Notably, upregulation of IL-1β can create a vicious cycle leading to excessive accumulation of intracellular iron because it has been shown to upregulate RPE iron importers and downregulate iron exporters [493]. Sterling et al. have shown that high-fat-diet-induced obesity, a risk factor for AMD, drives systemic and local inflammatory circuits upregulating IL-1β [493]. Moreover, with regard to the pro-inflammatory signalling, it has been shown that an 8-h exposure of ARPE-19 cells to IL-1β or A2E results in the upregulation of a potent pro-inflammatory enzyme, cyclooxygenase-2 [494,495]. As stated previously, A2E is strongly embedded inside the lipofuscin granule, therefore it is unclear whether effects observed in cultured cells where A2E is injected from its solution in an organic solvent can occur under more physiological conditions.

Secondly, in comparison with age-matched normal human retinas, AMD-affected retinas exhibit a greater level of protein modifications by oxidized lipids, such as CEP, which has been detected diffusely distributed in photoreceptor outer segments, RPE, Bruch’s membrane and drusen [496,497]. Moreover, it has been shown that Western blots of proteins extracted from RPE-Bruch’s membrane-choroid complexes isolated from 11 AMD and 11 normal eyes show positive staining of numerous proteins for anti-CEP antibody, and this staining is more pronounced in AMD tissues than in normal tissues [496].

It has been shown that CEP can stimulate angiogenesis in the chick embryo, rat cornea, and in a mouse model of laser-induced chorioretinal neovascularization [498,499]. Interestingly, administration of anti-VEGF antibodies only partly blocks CEP–protein adduct-induced angiogenesis in vivo, while it completely blocks angiogenesis induced by VEGF, suggesting an involvement of an additional, VEGF-independent, angiogenesis pathway. While CEP does not upregulate VEGF in ARPE-19 cells in vitro, another product of lipid oxidation, 4HNE does so at a concentration as low as 1 µM [500]. Also, it has been shown that exposure to blue light of A2E-laden ARPE-19 leads to an increased expression of VEGF mRNA and protein, but it is unclear whether these effects of A2E delivered in organic solvent are physiologically relevant [501,502]. VEGF is a potent angiogenic factor that promotes choroidal neovascularization and anti-VEGF treatments have been established as effective in preventing neovascularisation and its deleterious consequences in the neovascular form of AMD [503]. In addition to triggering pro-angiogenic signalling, oxidized lipids can also trigger pro-inflammatory signalling. For example, the phagocytosis of oxidized POS by cultured ARPE-19 causes 2.8- and 3.2-fold increased expression of proinflammatory cytokines: interleukin 8 (IL-8; also known as the neutrophil chemotactic factor) and monocyte chemoattractant protein-1 (MCP-1, also known as CCL2), respectively, in comparison to cells fed naïve POS [504]. IL-8 and MCP-1 are potent chemoattractants for phagocytes and are elevated in the vitreous of patients with retinal neovascularization [505]. Intravitreal injection of 4HNE into the mouse eye results in an increased formation of C3a in the RPE in comparison with the vehicle-treated mice suggesting that 4HNE can activate the complement cascade [506]. Notably, C3a accumulation is present in Bruch’s membrane in AMD-affected eyes [507].

Products of A2E oxidation, such as peroxy-A2E and furano-A2E, and A2E-laden cells exposed to blue light can activate the complement cascade in the human serum, as observed by monitoring the cleavage products of complement component C3, which are known to trigger inflammatory responses and formation of the membrane attack complex (MAC), which can cause cell death [227,508,509]. Again, caution is needed with extrapolating the results obtained on cultured cells with A2E injected from a solution in an organic solvent to physiological situation where A2E and its early oxidation products are inside the lipofuscin granules [227]. However, MAC deposition was also observed in cultured cells supplemented with bovine retinal extract where RPE cells differentiated from intrinsically pluripotent stem cell line from the Stargardt’s disease patient and from an unaffected human donor [508]. RPE cells derived from Stargardt’s patient had a seven-fold decreased protein expression of ABCA4 in comparison with RPE cells derived from the unaffected donor. The level of bisretinoids was not reported in this study. After 10 months of retinal extract supplementation, RPE cells from the Stargardt’s patient had a two-fold increase in fluorescence excited by 488 nm laser, a massive increase in immunofluorescence staining against 4HNE, and about 30% increase in MAC deposition in comparison with RPE cells from the unaffected donor. Increased immunoreactivity against 4HNE and MAC deposition was also observed in RPE cells in RPE flatmounts and retinal sections, respectively, from Stargardt’s disease donor eyes in comparison with unaffected eyes [509].

Thirdly, several reports demonstrate that in RPE from AMD retinas, the damage to mitochondrial DNA is greater than in age-matched normal RPE and the mitochondrial DNA damage increases with the disease progression [510,511,512,513,514,515,516]. Interestingly, the mitochondrial DNA damage is significantly higher in RPE from AMD retinas with the homozygous complement factor H (CFH) variant Y402 (due to single nucleotide polymorphism rs1061170), which is associated with an increased risk for AMD, than in retinas with H402 variant of CFH [513].

Moreover, RPE from AMD retinas exhibits upregulation of two important antioxidant enzymes, glutathione peroxidases (GPXs): extracellular GPX3 and mitochondrial isoform of GPX4, which catalyse the reduction of hydrogen peroxide and lipid hydroperoxides by glutathione [517].

Finally, the role of oxidative stress in the progression of AMD is underscored by the results of two large clinical trials Age-Related Eye Disease Study (AREDS) and AREDS2 showing that patients with moderate AMD receiving a mixture of antioxidants including vitamin C, vitamin E, zinc and carotenoids were at a smaller risk of progression to advanced AMD, especially if their dietary intake of these antioxidants was low [518,519,520,521].

The increased accumulation of iron and oxidative damage to biomolecules in AMD retina provide evidence of an increased oxidative environment in the AMD retina. Therefore, it can be expected that lipofuscin can be oxidized in such an environment, and this can be detected by measurement of fluorescence emission spectra or a ratio of cyan (about 470 nm) to orange-red (620–650 nm) fluorescence.

Considering the above-mentioned products of oxidation, which can activate the complement cascade and upregulate VEGF, an effective antioxidant therapy can decrease the risk of the formation of such oxidation products, and therefore act upstream of these deleterious effects thereby preventing complement activation and excessive secretion of VEGF, both of which are held responsible for the development of AMD. There is room for improvement in AREDS2 supplementation with Vitamin E, vitamin C, lutein, zeaxanthin, zinc and copper. Other approaches to inhibit oxidative damage to the retina include inhibitors of the retinoid cycle, such as Emixustat (known also as ACU-449), which is tested for slowing down the progression of AMD and Stargardt’s disease in several clinical trials (ClinicalTrials.gov Identifiers: NCT03033108, NCT03772665, NCT02130531, and NCT01002950), while improved derivatives of Emixustat are being developed [522]. Another approach to decrease the oxidative damage induced by all-*trans*-retinaldehyde upon release from visual pigments is by inhibiting delivery of vitamin A to the retina by compounds such as Tinlarebant tested in clinical trials NCT05266014 and NCT05244304, STG-001 (NCT04489511), and Fenretinide (NCT00429936).

Previously mentioned Remofuscin, which has been shown in animal studies to stimulate lipofuscin removal from the RPE, is also tested in a clinical trial (EudraCT registration: 2018-001496-20 STARTT) and is likely to affect the fluorescence of the retina.

Also, the deuterated form of vitamin A (C20-D3-Vitamin A also known as ALK-001) discussed earlier has been shown in animal studies to slow down the formation of bisretinoids, which otherwise can facilitate lipofuscin formation by inhibition of lysosomal degradation, is currently tested in three clinical trials: NCT03845582, NCT02402660, NCT04239625), and is likely to affect the fluorescence of the retina.

Another approach to decrease the oxidative damage in the retina is the use of deuterated polyunsaturated fatty acids such as DHA, where hydrogen atoms at bis-allylic sites are replaced by deuteriums [488,489,490,491,492]. Supplementation with deuterated DHA is expected to inhibit lipofuscin accumulation by inhibiting the propagation steps in lipid peroxidation, which can be initiated not only by reactive oxygen species generated by lipofuscin, retinaldehyde and its derivatives, but also by reactive oxygen species generated during phagocytosis, and by other compounds, such as iron, which accumulate in the AMD retina. It is thought that the huge inhibition of lipid peroxidation by deuterated DHA occurs at the propagation step: the deuterium on the methylene group is less susceptible to abstraction by a lipid-derived peroxyl radical than a hydrogen on such a methylene group. As a result, it gives time for antioxidants such as vitamin E to scavenge the lipid-derived peroxyl radical or for the peroxyl radical to interact with another peroxyl radical resulting in the termination of the chain reaction.

### 7.5. RPE Lipofuscin Fluorescence: Intensity and Spectral Characteristics as a Potential Biomarker of Oxidative Damage to the Retina In Vivo

There is a growing body of evidence demonstrating that: (i) RPE lipofuscin is the major fluorophore in the retina; (ii) oxidation of RPE lipofuscin results in characteristic changes in its emission spectrum; (iii) oxidation of RPE lipofuscin can occur in vivo; (iv) there is increased oxidative damage in the AMD retina; (v) there is a spectral shift of fluorescence emission maximum towards shorter wavelengths in AMD retina; (vi) a shorter wavelength of the emission maximum is a risk factor for the AMD progression. Therefore, it can be expected that spectral changes in fluorescence emission due to lipofuscin oxidation in the AMD retina can be a contributing factor to the observed changes in fluorescence in the AMD retina. It is tempting to suggest that the characteristics of lipofuscin fluorescence can be used in clinics to monitor the oxidative damage in AMD and other retinal degenerations associated with lipofuscin accumulation and to evaluate the efficacy of potential antioxidant therapies.

It appears that the recent developments in imaging techniques, which include combined OCT and fluorescence imaging of the fundus with subcellular resolution make such monitoring in vivo of the spectral changes in fluorescence emission of lipofuscin feasible [353,395,421]. The combined use of OCT and SLO allows for the identification of areas with SDD or drusen, which can also be a source of fluorescence. The subcellular resolution of fluorescence imaging is enabled by TPEF-SLO, which limits the area of the excitation, and imaging of both fluorescence and reflectance [353,395,421]. The latter is used to identify the retinal layer where the laser light is focused so it is the origin of the fluorescence. Moreover, due to excitation with near-infrared lasers, the artefacts introduced by the anterior segment of the eye are minimized. Importantly, it has been demonstrated that such imaging can be performed safely for the human eye in vivo [353,395,421].

However, current protocols for excitation and collection of emitted fluorescence from the human fundus in vivo are not optimal to detect the spectral changes occurring in lipofuscin fluorescence due to oxidation. Lipofuscin oxidation results in a decrease in its yellow-red fluorescence with a concomitant increase in the emission of its blue fluorescence (Figure 8) [75]. Optimizing the excitation wavelength and the spectral ranges of collected emission can improve the specificity for monitoring these changes and thereby improve the sensitivity with which the progression of oxidation occurs. The safety aspects need to be considered as well. All-*trans*-retinal, which transiently accumulates in the photoreceptor outer segments after photoexcitation of visual pigments, is a potent photosensitizer absorbing visible light up to 460 nm. Absorption of flavins extends even more towards longer wavelengths.

Boguslawski and colleagues have demonstrated that safe imaging of the human fundus in vivo by TPEF-SLO provides a sufficient signal-to-noise ratio for informative imaging of the reflectance and fluorescence emission when collected in distinct spectral windows obtained by bandpass filters: 400–600 nm, 594–646 nm, 500–540 nm, and 400–550 nm [353,395,421]. However, the excitation source used in this instrument providing 76 fs pulses of 780 nm wavelength excites multiple retinal fluorophores and photosensitizers, including not only lipofuscin but also retinoids, flavins, and porphyrins. While this has its own advantages, including the ability to monitor the retinoid cycle in vivo, it can be argued that for imaging of oxidation-induced changes in lipofuscin fluorescence, TPEF-SLO using excitation with femtosecond pulses of 920 nm near-infrared laser [413] could provide a better option by being more specific for excitation of lipofuscin.

Such an excitation with 55 fs pulses of 920 nm light fired at 80 MHz repetition rate combined with AO-SLO with imaging of both reflectance and fluorescence of various retinal layers was used in a study by Sharma and colleagues on anaesthetised macaques in vivo [411,413]. This elegant study demonstrated the great potential of such a technique for visualizing fluorophores in the individual layers of the retina but, despite the emitted fluorescence was collected over a broad range of 400–680 nm, to achieve the sufficiently high signal-to-noise ratio required radiant exposure 3.5-fold greater than the maximal permissible exposure for that wavelength given by the American National Standard for Safe Use of Lasers ANSI Z136.1–2014. 2014.

To make their TPEF radiant exposures safe, Boguslawki and colleagues employed a dispersion precompensation unit, which enabled the two-photon excitation to be more efficient and reduced the pulse repetition rate from the typically used 80 MHz to 6 MHz, which enabled to increase the peak power and therefore increase the yield of fluorescence [395,421]. Therefore, it appears that TPEF with 920 nm light can also be amended to be safe for the retina. The spectral channels for the acquisition of emitted fluorescence would also benefit from collecting short-wavelength fluorescence from the 470–500 nm range, mid-wavelength fluorescence around 540 nm, and long-wavelength fluorescence from 560–700 nm range. The intensity of mid-wavelength fluorescence would provide information on the presence of lipofuscin, whereas the ratio of the emissions in the short- to long-wavelength spectral channels would provide information on the oxidation state of lipofuscin.

## 8. Conclusions

In conclusion, despite substantial research efforts, lipofuscin remains rather enigmatic. Numerous components derived from retinoids have been identified in RPE lipofuscin, but not many other components of interest, such as those that exhibit strong photosensitizing properties and may be derived from the oxidation of polyunsaturated fatty acids and/or are not soluble in organic solvents. The contribution of such components to lipofuscin fluorescence is also not known. Oxidized DHA, which is a likely component of lipofuscin, fluoresces upon excitation with UV or 488 nm blue light [75]. When excited with UV light, oxidized DHA exhibits its fluorescence emission maximum in blue and extends its tail into the yellow-red spectral range. It can be expected that at least some of these not-identified components derived from DHA oxidation can be common between lipofuscin from RPE and other tissues rich in DHA, such as the brain.

It is well-established that oxidative stress and lysosomal dysfunction play a major role in lipofuscin accumulation. Less is known about the effects lipofuscin exerts on cells under physiological conditions. The protective effect on the progression of geographic atrophy in Stargardt’s disease patients supplemented with deuterated vitamin A, which has been shown in vitro and animal models to slow down the formation of bisretinoids and accumulation of lipofuscin [275], suggests that prevention of lipofuscin accumulation may be an effective treatment for Stargardt’s and other diseases associated with increased levels of lipofuscin.

It appears that cells can accumulate substantial amounts of lipofuscin before reaching a certain threshold above which the cells can no longer cope, and the deleterious properties of lipofuscin manifest themselves. Alternatively, when lipofuscin starts exhibiting toxic properties, which could be induced by exposure to light, the cells remove it but it is unclear what mechanisms are involved in the removal of packets of lipofuscin from RPE cells, and whether it contributes to SDD, drusen, or other sub-RPE deposits in Bruch’s membrane.

Lipofuscin is the major contributor to the fluorescence of the retina. Imaging of RPE lipofuscin fluorescence is widely employed in clinics to monitor the progression of atrophic AMD, Stargardt’s disease, bestrophinopathies, and some forms of retinitis pigmentosa associated with lipofuscin accumulation and RPE cell loss. Oxidation of RPE lipofuscin results in spectral changes in its fluorescence emission: a decrease in its yellow-red fluorescence with a concomitant increase in the emission of its blue fluorescence. It has been shown that lipofuscin oxidation can occur in vivo. However, the excitation wavelength in currently used fluorescence imaging instruments in clinical settings is too long so the fluorescence emission cannot be collected in the blue spectral range where the most pronounced fluorescence increase occurs upon oxidation.

## 9. Future Research Directions

### 9.1. Elucidation of the Role of Oxidized DHA in Photosensitizing and Fluorescence Properties of Lipofuscin

The numerous products of DHA oxidation include whole families of compounds such as neuroprostanes, oxidized neuroprostanes, neuroketals and oxidized neuroketals [111,112,113,117,120,523]. Neuroketals are a family of 256 γ-keto aldehydes formed via neuroprostane pathway and deserve particular attention. Neuroketals are highly reactive and form adducts with lysine residues on proteins, which then can cause cross-links and aggregation, protein dysfunction, and ultimately cell death [114]. Neuroketal-protein adducts have been detected in post-mortem tissues from 18 normal brains of donors 40–79 years and showed an age-related increase when tissues from the 40–48-year-old group were compared with tissues from the 70–79-year-old group [119]. The proteins that were modified by neuroketals include some that are essential for energy metabolism, cytoskeleton, proteostasis, neurotransmission, and haem metabolism. Surprisingly, despite numerous studies on the deleterious effects of oxidized DHA in the brain, the studies on the retina seem to be limited to CEP and no attempts of neuroketals detection in the retina have been reported. Research in that direction may not only identify important mechanisms for cell dysfunction and death in several retinal diseases but also lead to potential treatments, which may involve the use of scavengers of γ-keto aldehydes [121] or inhibiting lipid peroxidation in the first place [488,490,524].

While none of the reported neuroketals appear to have sufficient conjugation system to absorb blue or longer wavelength light, some of the neuroketal-lysyl lactam adducts have five or even seven conjugated double bonds [111,112] suggesting that they can absorb short-wavelength visible light and should be considered as potential photosensitizers and/or fluorophores of lipofuscin.

### 9.2. Relative Contribution of Retinaldehydes and Lipofuscin to Light-Induced Retinal Injury

To elucidate the relative contributions of lipofuscin and retinaldehydes to light-induced retinal injury, animals with accumulated lipofuscin could be exposed to damaging light in the absence and presence of inhibition of the synthesis of 11-*cis*-retinaldehyde by RPE65. Moreover, the susceptibility to light-induced retinal injury could be compared for animals with decreased accumulation of lipofuscin by deuterated vitamin A and animals fed with normal vitamin A.

### 9.3. Determination of Topography of Retinal Irradiance under Various Daily Activities in Different Geographical Locations and Atmospheric Conditions

In all epidemiological studies attempting to investigate a relationship between sunlight exposure and AMD discussed earlier, only chronic exposure to sunlight was considered. Another possibility worth consideration is that acute exposure to light that exceeds the antioxidant capacity of photoreceptors and RPE cells causes damage, which may happen only over a small portion of the retina, but can trigger focal oxidative damage, activation of complement and/or pro-angiogenic response. Evaluations of retinal exposures like that are now feasible due to available technology such as an eye-tracker and a miniature wide-angle camera, which can be mounted close to the eye and capture the scene within the visual field, and spectroradiometry to measure spectral radiance across the visual field. It would be extremely useful to determine the risk of focal retinal damage when walking without sunglasses with the Sun in the visual field, staying on a beach, or walking/skiing in the snow with no protective eyewear. Such data would provide valuable information on whether such activities can put the retina at risk of light-induced injury. The feasibility of such measurements with an eye-tracker and a camera capturing the images in the visual field was shown already in 2012 by Bone et al. but also revealed the challenges of quantitative measurements of the absolute illuminance values using a scene camera with a small dynamic range [525,526]. Overcoming these challenges and adding spectroradiometric measurements and the measurements of the transmission of the lens, would enable the calculation of the spatial distribution of the retinal spectral irradiance in the absolute values. In the future, such data combined with mobile phone apps, such as Google Maps Timeline, and meteorological data for visited places provide means to evaluate exposure to sunlight in longitudinal studies more accurately.

### 9.4. Comparison of the Effects of Deuterated Vitamin A and Deuterated DHA on Lipofuscin Accumulation, Susceptibility to Light-Induced Retinal Injury, and Progression of Geographic Atrophy in Animal Models of Stargardt’s Disease and AMD

In the clinical trial, the supplementation of patients with deuterated form of vitamin A, C20-D3-retinyl acetate led to about 90% vitamin A in the blood in its deuterated form suggesting that the supplement provided the vast majority of vitamin A and exceeded about 10-fold its dietary intake [275]. The recommended dietary intake of vitamin A is 0.9 and 0.7 mg retinol per day for men and women, respectively [527]. These requirements can be easily exceeded by eating products rich in vitamin A and its precursors, which are abundant in dairy, fish, cantaloupe, broccoli, squash, peas, and spinach. Toxicity of vitamin A is usually associated with intakes of at least 30 mg per day but there are reports of liver abnormalities after chronic intake of vitamin A well below that amount suggesting that achieving the desired ratio of deuterated vitamin A over the dietary vitamin A may require serious restrictions to the diet and removing some products, which are otherwise considered healthy and their consumption has been shown to be associated with a decreased risk of AMD.

In contrast, the beneficial effects of deuterated DHA have been shown when it replaced only 30% of normal DHA in the mouse retina [488]. Surprisingly, there are no clinical trials testing the effects of deuterated DHA on major diseases of the retina or brain, which are associated with increased oxidative stress. There are numerous clinical trials with supplementations with DHA up to 6 g/day demonstrating its safety [528,529]. The mean dietary intake of DHA in the USA is rather low, on average less than 100 mg/day but can be increased by eating fatty fish, such as mackerel or salmon, which can increase the daily intake for up to 542 mg/day [530,531]. The latter amount would still be 10-fold smaller than deuterated DHA coming from a safe dose of supplement. As mentioned earlier, the advantage of deuterated DHA is its ability to inhibit lipid peroxidation, which can be initiated not only by reactive oxygen species generated by lipofuscin, retinaldehydes and their derivatives but also by reactive oxygen species generated during phagocytosis, and by other compounds, such as iron, which was shown to accumulate in the retina with ageing and even more so in AMD.

### 9.5. Stimulation of Lipofuscin Removal by Light

Observations from TEM of human and monkey eyes suggest that RPE cells can remove lipofuscin into the extracellular space. Based on TEM observations of RPE and Bruch’s membrane from human eyes from macular and extramacular areas, Burns and Feeney-Burns suggested that RPE cells shed parts of their cytoplasm, which can penetrate the basal lamina and contribute to drusen [532]. Gouras et al. observed what appears to be exocytosis of lipofuscin from the RPE towards Bruch’s membrane in monkey eyes [533,534]. Recent observations of lipofuscin removal from cultured ARPE-19 cells suggest that lipofuscin removal can be stimulated by light [75].

It can be argued that stimulation of lipofuscin removal by well-controlled exposure of selected RPE areas to light may be a safer approach than stimulating lipofuscin removal by Remofuscin, which accumulates long term in the RPE and increases oxidative stress [262,535,536]. Lipofuscin and melanosomes exocytosed as a result of treatment with Remofuscin are taken up by macrophages. Therefore, it is worth investigating whether light-induced exocytosis of lipofuscin is also accompanied by exocytosis of melanosomes and whether it happens only on the basal side or it can happen on the apical side. Such experiments could be performed on polarised cultures of RPE grown on inserts and in animal models of human diseases in vivo. Finally, it is worth investigating whether lipofuscin exocytosis is accompanied by the release of chemoattractants to phagocytes, such as monocytes or choroidal macrophages, which can clear not only the exocytosed lipofuscin but also other debris such as drusen. The latter possibility is supported by the disappearance of drusen following laser-induced photocoagulation of the retina used to limit retinal detachment or choroidal neovascularisation [537].

Photocoagulation was tested in clinical trials as a prophylactic treatment of AMD and demonstrated its effectiveness in drusen removal but also in increasing the inflammatory response and risk of choroidal neovascularisation. The approach was later refined by using shorter laser pulses (nanoseconds to microseconds) than those used for photocoagulation (milliseconds), to limit the thermal damage to adjacent cells and cause selective damage to RPE cells by photomechanical disruption of melanosomes, as opposed to massive damage to RPE and adjacent cells caused by photocoagulation [537,538]. A clinical trial using 3 ns laser showed statistically significant drusen regression 24 months post-treatment in comparison with the natural history records of nontreated patients. It can be suggested that the drusen clearance following the treatment with the 3 ns laser was due to monocytes/macrophages infiltrating the retina in response to signalling from damaged cells. It is of interest to determine whether the removal of lipofuscin induced by light is accompanied by similar signals attracting phagocytes for clearance of lipofuscin as well as drusen. If so, this approach would offer a safer drusen removal than treatments with pulsed lasers.

### 9.6. Lipofuscin Fluorescence as a Way of Monitoring Oxidative Damage in RPE

Recent developments in TPEF-SLO for safe imaging of the human retina suggest that this technique has the potential to be useful for imaging of the oxidation state of lipofuscin with two-photon excitation by 920 nm laser pulses and monitoring spectrally-resolved emission above 465 nm. TPEF allows for photoexcitation of a very small retinal area within the RPE layer therefore avoiding photoexcitation of fluorophores of the sub-RPE deposits and Bruch’s membrane and subretinal deposits and photoreceptor outer segments. It is expected that such an approach can be useful for monitoring the oxidative damage in AMD and other retinal degenerations associated with lipofuscin accumulation and for evaluation of the efficacy of potential antioxidant therapies.

### 9.7. Is RPE Lipofuscin Really So Much Different from Lipofuscins from Other Cells?

While bisretinoids may be a unique feature of RPE lipofuscin, polyunsaturated lipids can be common components for many types of lipofuscin, especially those from the brain, which is rich in DHA and other polyunsaturated fatty acids. It well may be that oxidation-induced spectral changes in lipofuscin from other tissues may be a biomarker of oxidative damage and the formation of highly harmful end products of lipid oxidation. Therefore, it would be of interest to compare the effects of oxidation on the fluorescence properties of RPE lipofuscin with lipofuscin from other tissues. Ideally, the fluorescence spectra should be corrected for the spectral sensitivity of the detection so they can be compared between different laboratories.

## Figures and Tables

**Figure 1 antioxidants-12-02111-f001:**
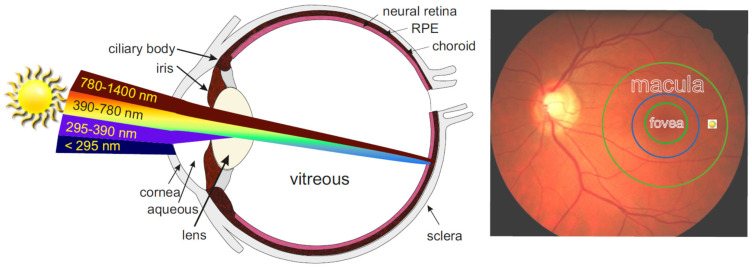
A diagram shows a cross-section of the human eye (not to scale) and a photograph of the fundus with depicted areas of the fovea and macula. The cornea absorbs light below 295 nm, and the crystalline lens in the young adult absorbs the remaining UVB and UVA light, although some individuals above 60 years of age still retain the transmission window at about 320 nm [79]. The image of the Sun focused on the retina (shown below the fovea) is about 0.16 mm in diameter [80]. Modified from [81]. The diameter of the fovea is 1.5 mm. A 0.5 mm-thick concentric ring surrounding the fovea is known as the parafovea (the outer border of which is shown as a blue circle), and a 1.5 mm-thick concentric ring surrounding the parafovea is known as the perifovea, whose outer border is the same as the border of the macula [82]. The diameter of the macula is 5.5 mm. The macula, and the fovea in particular, are the areas of the retina responsible for vision with the greatest acuity.

**Figure 2 antioxidants-12-02111-f002:**
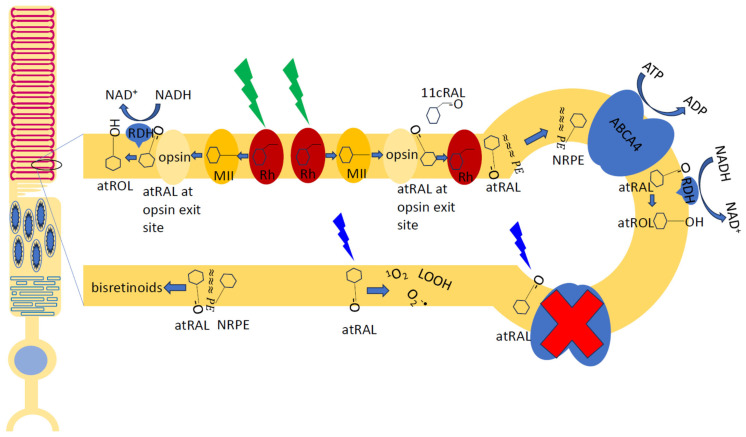
Schematic diagram of photoreceptor and a part of disc from a photoreceptor outer segment. Red ovals depict visual pigment, rhodopsin (Rh) in its inactive state where its chromophore 11-*cis*-retinaldehyde (11cRAL) is bound at the active site to Lys296 via Schiff base linkage. Absorption of a photon (green lighting arrow) by the chromophore results in its isomerization to the all-*trans* form followed by conformational changes of the protein creating a biochemically active state, metarhodopsin II (MII), with all-*trans*-retinaldehyde (atRAL) still bound to the active site. The hydrolysis of all-*trans*-retinaldehyde results in an apo-protein, opsin, with all-*trans*-retinaldehyde noncovalently bound to the opsin exit site, where it can be reduced to all-*trans*-retinol (atROL) by NADPH-dependent retinol dehydrogenase (RDH). Another scenario that may follow an absorption of a photon by rhodopsin is the regeneration of rhodopsin upon binding to opsin of 11-*cis*-retinaldehyde. Presumably, this allows atRAL to dissociate from the protein and diffuse into the lipid membrane. AtRAL on the intradiscal side of the lipid membrane cannot serve as a substrate for RDH. AtRAL can form a Shiff-base linkage with the ethanolamine group of phosphatidylethanolamine (PE), which accounts for about 40% of disc membrane phospholipids. The product, N-retinylidene phosphatidylethanolamine (NRPE) serves as a substrate of ATP-binding cassette transporter A4 (ABCA4) present in the rim of the disc. ABCA4 flips NRPE to the cytoplasmic side of the disc membrane where, upon hydrolysis from PE, atRAL becomes a substrate for RDH. This scenario is less likely in the retina where the synthesis of 11-*cis*-retinaldehyde is inhibited. Such inhibition can be induced by inhibitors of RPE65, isomerohydrolase, which converts in the RPE all-*trans*-retinyl esters into 11-*cis*-retinol or occurs naturally in mice with Met at position 450 of RPE65, which makes the enzymes less active than its variant Leu450. The bottom portion of the disc depicts processes that may happen when free atRAL accumulates. Interaction of NRPE with atRAL leads to the formation of bisretinoids. It can be expected that the likelihood of such a reaction is low because the reaction of atRAL with PE is fast and atRAL is in proximity to PE, which outnumbers the concentration of atRAL by a factor of at least 100. Absorption of light by atRAL (blue lighting arrow) in the presence of abundant oxygen results in the generation of reactive oxygen species: singlet oxygen (^1^O_2_) and superoxide radical anion (O_2_^−•^). Singlet oxygen can react with carbon–carbon double bonds in unsaturated lipids forming lipid hydroperoxides (LOOH). Photoexcitation of atRAL in the presence of ABCA4 results in the inactivation of the enzyme (red cross). Inhibition of RPE65 by pharmacological treatment or Met450 variant of RPE65 results in a slower delivery of 11cRAL to photoreceptors and consequently a substantial decrease in susceptibility to light-induced retinal injury. Modified from [94].

**Figure 3 antioxidants-12-02111-f003:**
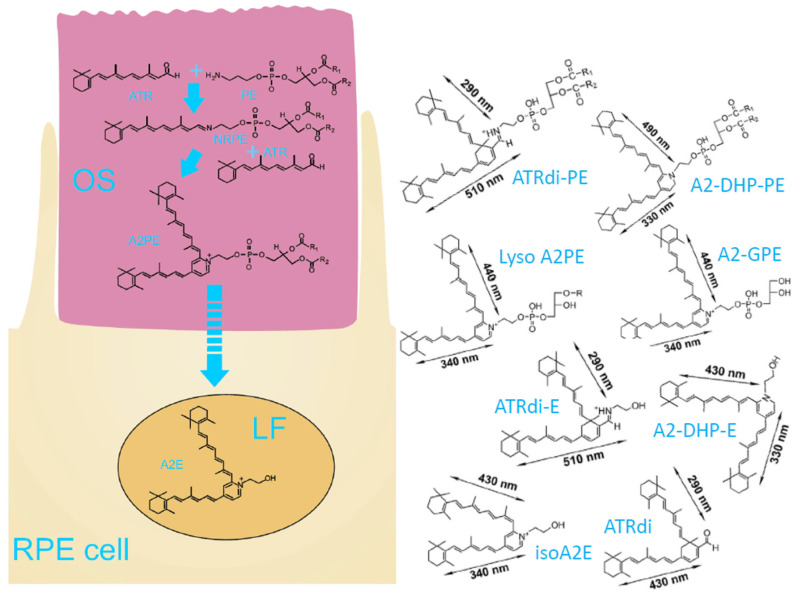
(**Left**) Initial biosynthetic pathway for the formation of a lipofuscin fluorophore A2E in photoreceptor outer segment (OS) and its structure as a component of lipofuscin (LF) in the RPE cell. All-*trans*-retinaldehyde (ATR) hydrolysed from opsin binds to a highly abundant outer segment phospholipid, phosphatidylethanolamine (PE), forming N-retinylidene phosphatidylethanolamine (NRPE). The binding of another molecule of ATR to NRPE results in the A2PE. Upon lysosomal cleavage of A2PE by phospholipase D in the RPE, A2E is formed. R1 and R2 stand for various fatty acids. (**Right**) Other bisretinoids that can be formed in the OS and RPE: ATR dimer-PE (ATRdi-PE), A2-dihydropyridine-PE (A2-DHP-PE), lyso A2PE, A2-glycerophosphoethanolamine (A2-GPE), ATR dimer-E (ATRdi-E), A2-dihydropyridine-E (A2-DHP-E), A2E isomer (isoA2E), and ATR dimer (ATRdi). Modified from [94,127].

**Figure 4 antioxidants-12-02111-f004:**
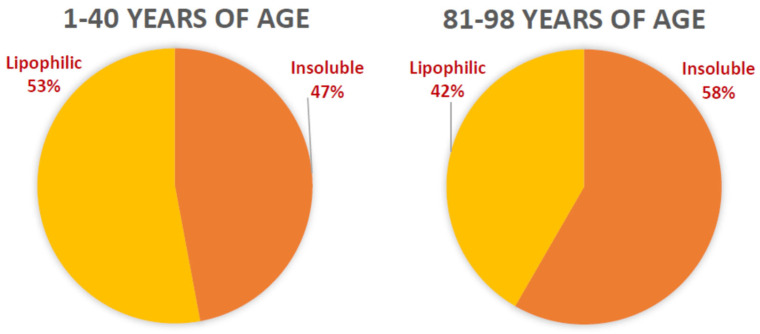
Components of lipofuscin (LF) granules isolated from two age groups: 1–40 and 81–98 years of age and extracted from its suspension in PBS by a mixture of chloroform/methanol. Lipophilic components enter the chloroform-enriched phase (Lipophilic) while the rest forms an insoluble interphase (Insoluble). Lipophilic components of lipofuscin include phospholipids, free fatty acids, bisretinoids, and their oxidation products (shown in Figure 3 and Figure 5). Chloroform-insoluble components include highly modified proteins, including modifications by carboxyethylpyrrole (CEP), iso[4]-levuglandin E2, malondialdehyde (MDA), 4-hydroxynonenal (4HNE), methionine sulphoxide, methione sulphone, advanced glycation end products, and 3-nitrotyrosine. Based on data from [94].

**Figure 5 antioxidants-12-02111-f005:**
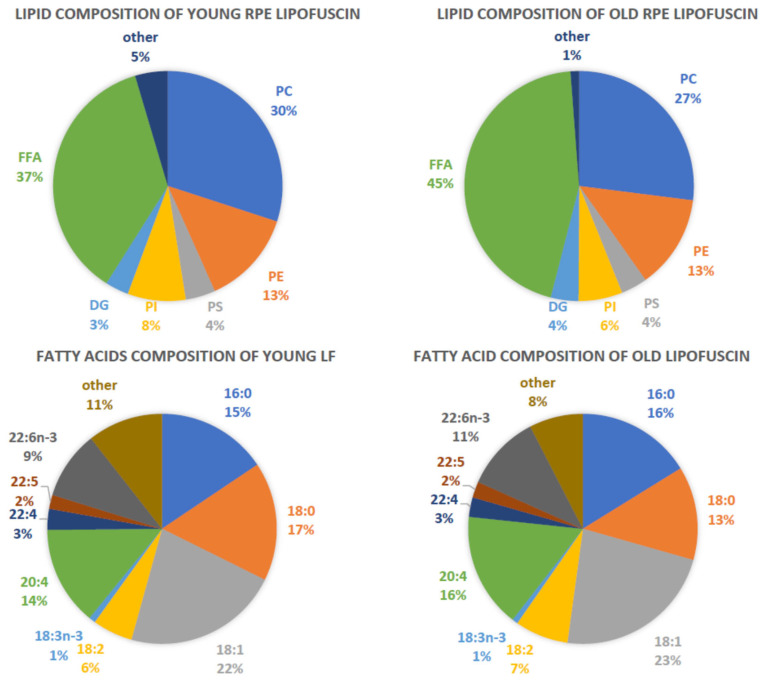
Lipids and fatty acids identified in chloroform extract from RPE lipofuscin isolated from two age groups: 6–39-year-olds (YOUNG) and 47–89-year-olds (OLD) composition. PC, phosphatidylcholines; PE, phosphatidylethanolamine; PS, phosphatidylserine; PI, phosphatidylinositol; DG, diacylglycerol; FFA, free fatty acid; 16:0, palmitic acid; 18:0, stearic acid; 18:1, oleic acid; 18:2, linoleic acid; 18:3n-3, omega-3 linolenic acid; 20:4, arachidonic acid; 22:4, docosatetraenoic acid; 22:5, docosapentaenoic acid; 22:6n-3, omega-3 docosahexaenoic acid (DHA). Based on data from Bazan et al. [110].

**Figure 6 antioxidants-12-02111-f006:**
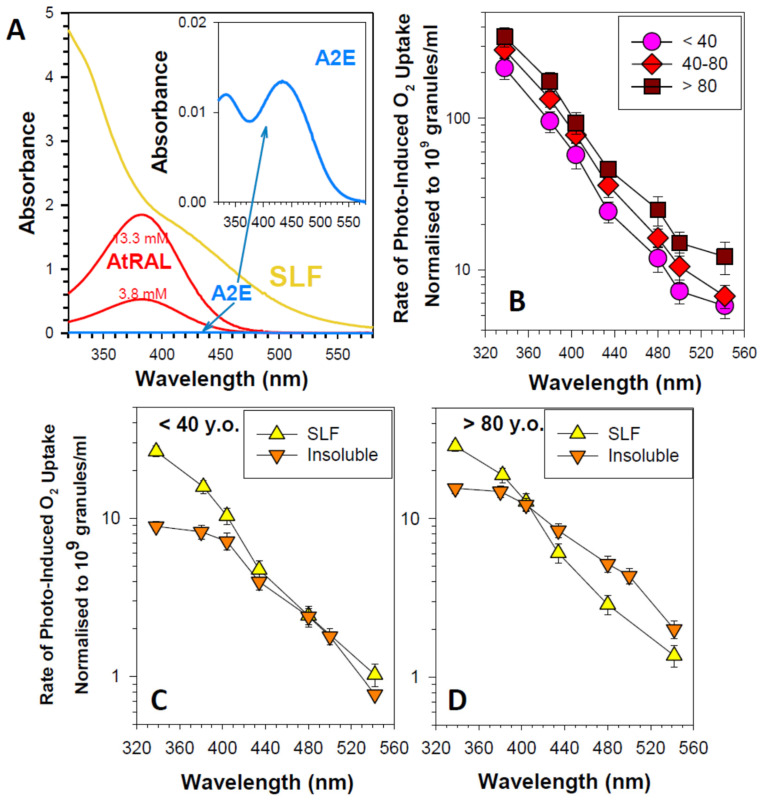
(**A**) Estimated upper limits for absorption of UV and visible light by all-*trans*-retinaldehyde (AtRAL), chloroform-soluble components of lipofuscin (SLF), and a component of lipofuscin called A2E in the retina. Absorption spectra of AtRAL correspond to 3.8 mM and 13.3 mM solutions of all-*trans*-retinaldehyde in optical pathlength of 31.2 µm corresponding to the length of photoreceptor outer segments in the perifovea. The concentration of 3.8 mM corresponds to the concentration of rhodopsin in dark-adapted outer segments and the concentration of 13.3 mM corresponds to the worst-case scenario where all stores of retinyl esters were mobilized and converted to 11-*cis*-retinaldehyde for regeneration of rhodopsin, which was subsequently photobleached and AtRAL was hydrolysed from opsin but no enzymatic reduction to all-*trans*-retinol has taken place. Absorption spectra of soluble components of lipofuscin (SLF) in the RPE are based on (i) absorption spectra of measured dry weight of these components solubilized in benzene; (ii) the content of chloroform-soluble components of lipofuscin per lipofuscin granule, 0.093 pg/granule; (iii) an estimated number of granules of 7966 per RPE cell where lipofuscin occupies 19%of cell volume, and a lipofuscin volume is calculated based on an average diameter of lipofuscin granule of 0.5 µm; and (iv) thickness of the RPE layer of 14 µm. Absorption spectra of A2E are based on the content of A2E in lipofuscin granule of 7.8 × 10^−20^ mol/granule, and other assumptions as above. This leads to an averaged A2E concentration in the RPE of 0.224 mM. The integrated absorption of the visible light (>390 nm) of lipofuscin soluble components is 120 times greater than for A2E. Inset: Absorption spectrum of A2E after blowing out the ordinate axis. Note that the calculations of the expected absorption of light by AtRAL, SLF, andA2E refer to chromophores in solutions. Under physiological conditions, these chromophores are present in photoreceptor discs or are encapsulated within lipofuscin granules, and therefore their absorption cross-section in photoreceptor outer segments or the RPE may be substantially smaller than that in solution. (**B**): Wavelength dependence of initial rates of the light-induced oxygen uptake normalized to equal numbers of incident photons (action spectra of photooxidation) for suspension of lipofuscin granules (LF) isolated from RPE cells pooled into three age groups: below 40-year-old, 40–80-year-old, and above 80-year-old. (**C**,**D**): Action spectra of the initial rates of light-induced oxygen uptake by chloroform soluble (SLF) and insoluble components of lipofuscin isolated from two age groups: below 40-year-old and above 80-year-old, and suspended in PBS. Modified from [94,172,173].

**Figure 7 antioxidants-12-02111-f007:**
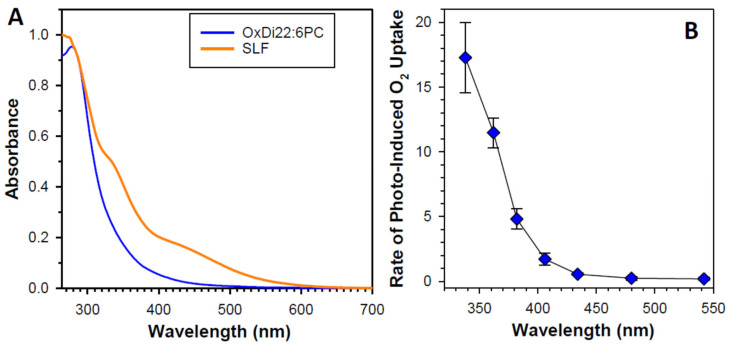
(**A**) Absorption spectra of oxidized di-docosahexaenoyl phosphatidylcholine (OxDi22:6PC) and chloroform soluble lipophilic components of lipofuscin (SLF). (**B**) Wavelength dependence of initial rates of the light-induced oxygen uptake normalized to the equal number of incident photons (action spectra of photooxidation) for suspension of OxDi22:6PC suspended in PBS. Modified from [240].

**Figure 8 antioxidants-12-02111-f008:**
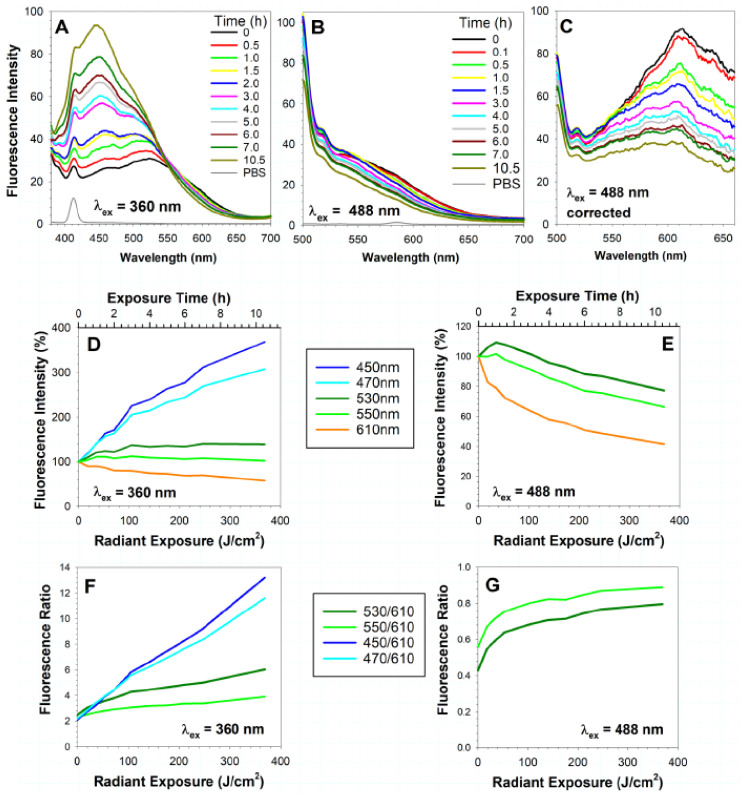
(**A**,**B**): Representative fluorescence emission spectra of lipofuscin granules isolated from human RPE and suspended in PBS before and after the indicated times of irradiation with visible light (9.76 mW/cm^2^). Fluorescence was excited with 360 nm (**A**,**D**,**F**) or 488 nm (**B**,**E**,**G**) light. (**C**): Fluorescence emission spectra from (**B**) corrected for the spectral changes in the detection sensitivity. (**D**,**E**): Kinetics of changes of fluorescence emission intensity monitored at the indicated wavelengths, normalized to the value before irradiation and expressed as a percentage. (**F**,**G**): fluorescence intensity ratios at the indicated wavelengths. The ratios of fluorescence intensities were calculated based on uncorrected spectra in the case of excitation with 360 nm and on corrected spectra in the case of excitation with 488 nm light. The fluorescence of the solvent, phosphate-buffered saline (PBS) was recorded to show its Raman emission peaks. From [75].

**Figure 9 antioxidants-12-02111-f009:**
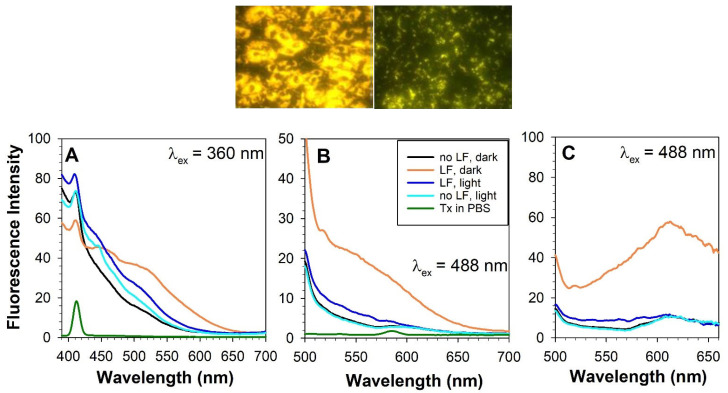
Top panel: fluorescence of lipofuscin-laden ARPE-19 cell monolayer kept in the dark (**left** panel) or exposed daily for 14 days to 45 min of 9.76 mW/cm^2^ visible light providing a total dose of 369 J/cm^2^ (**right** panel). Fluorescence images were acquired using a fluorescence microscope with blue light excitation, long-pass-emission filter and a colour camera capturing the colour of lipofuscin fluorescence as it appears to the human eye. The gain used for collecting fluorescence images was two-fold smaller for lipofuscin-laden cells not exposed to light than for cells exposed to light. (**Bottom**) panel (**A**,**B**): Representative fluorescence emission spectra from ARPE-19 cells with and without lipofuscin (LF) kept in dark or exposed to light as above and then solubilized in Triton X-100 (Tx). Fluorescence was induced by photoexcitation with 360 nm (**A**) or 488 nm (**B**) light. (**C**): Fluorescence emission spectra from (**B**), corrected for the spectral changes in the detector sensitivity. Modified from [75].

**Figure 10 antioxidants-12-02111-f010:**
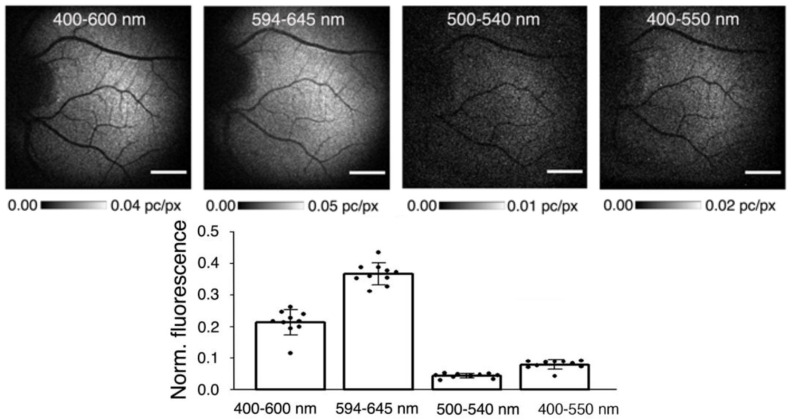
Top panel: imaging of two-photon-excited fluorescence (TPEF) by scanning laser ophthalmoscope (SLO) in the fundus of a healthy eye of a human volunteer. The laser system provided 76 fs pulses of 780 nm wavelength. The images of the scanned areas were acquired simultaneously in fluorescence channels at the indicated spectral ranges and in a reflectance channel. Bottom panel: the integrated fluorescence intensity in the four spectral ranges normalized to the integrated fluorescence intensity in the range of 400–700 nm. Adapted from Boguslawski et al. [395] licensed under a Creative Commons Attribution 4.0 International License. Copyright Boguslawski et al. 2022 [395].

**Figure 11 antioxidants-12-02111-f011:**
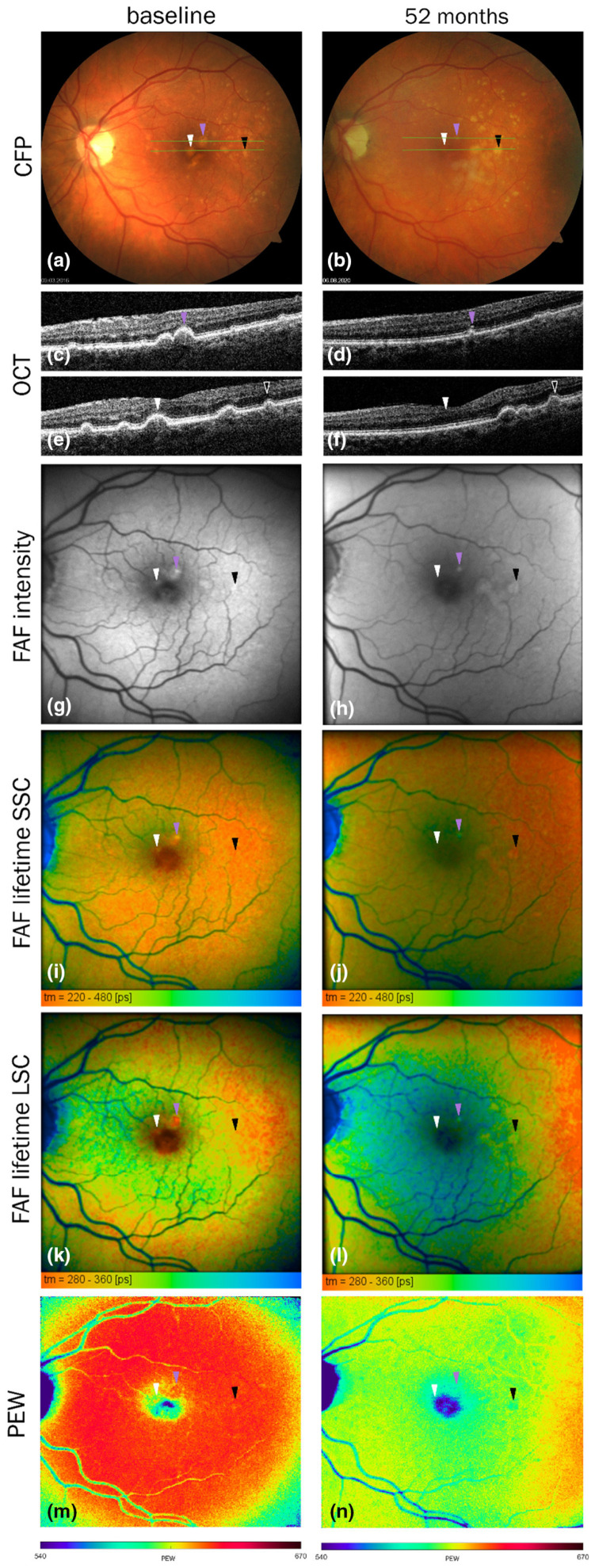
Multimodal imaging of a 72-year-old patient at baseline (panels on the **left**) and 52 months later (panels on the **right**); (**a**,**b**): colour fundus photography (CFP); (**c**–**f**): OCT at the green lines in (**a**,**b**); (**g**,**h**): fundus autofluorescence (FAF) intensity; (**i**,**j**): imaging of mean FAF lifetimes in the short-wavelength spectral channel (498–560 nm; SSC); (**k**,**l**): imaging of mean FAF lifetimes in the long-wavelength spectral channel (560–720 nm; LSC); (**m**,**n**): peak emission wavelength (PEW; calculated based on calibration in the range of 540–670 nm). Arrows point to drusen with different fates. Light purple arrowhead: druse with short mean fluorescence lifetime and indifferent PEW at baseline, which collapsed over follow-up leaving ectopic RPE hyperreflective foci in OCT. The lifetime changed from short to long and the PEW decreased. White arrowhead: hyperautofluorescent druse with a short lifetime and PEW, which disappeared over the follow-up not leaving RPE distortion. Black arrowhead: a druse growing over the follow-up with increasing mean fluorescence lifetime and decreasing PEW. From [468] under the Creative Commons Attribution License © 2022 Schwanengel et al. [468].

**Table 1 antioxidants-12-02111-t001:** Major fluorophores in the retina, their predominant location, and fluorescence characteristics. See text for further information.

Fluorophore	The Main Location in the Retina	Excitation	Emission	Effect of (Photo)Oxidation/Degradation
NAD(P)H	Photoreceptor inner segment	300–400 nm (maximum at 340 nm)	400–650 nm (maxima at 440 and 460 nm in the protein-bound and free state, respectively)	Decrease in fluorescence
Flavins	Photoreceptor inner segment	300–500 nm (maxima at 360 and 445 nm)	470–650 nm (maximum at 540 nm)	Decrease in fluorescence of the reduced form (due to interaction with superoxide); increase in 400–480 nm fluorescence of the oxidized form
Retinyl esters	RPE	300–360 nm (maximum at 330 nm)	400–650 nm (maximum at ~490 nm)	Decrease in fluorescence
Lipofuscin	RPE	Broad excitation spectrum covering 300–550 nm; 568 nm, 633 nm	Broad emission spectrum dependent on the excitation wavelength covering at least 420–725 nm range	An increase in the short-wavelength fluorescence and a decrease in the long-wavelength fluorescence
Melanolipofuscin	RPE	Broad excitation spectrum covering at least 300–500 nm	Broad emission spectrum covering at least 420–700 nm	Not determined
Melanosomes	RPE	Broad excitation spectrum covering at least 300–500 nm; 633 nm; 780–795 nm	Broad emission spectrum covering visible range 420–700 nm; near-infrared excitation and emission used in clinical settings	An increase in fluorescence
Drusen and other deposits in Bruch’s membrane	Bruch’s membrane	Excitation spectra not reported; excited by 364, 488, 545, 555, 568, 605, 633 nm; two-photon excitation with 960 nm	Broad emission spectrum 500–650 nm dependent on the excitation wavelength; emission with a maximum at 540–550 nm upon excitation with 436, 480, or 488 nm; emission maximum at 560 nm upon two-photon excitation with 960 nm	Not determined

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
