# Peer review of "Lipofuscin, Its Origin, Properties, and Contribution to Retinal Fluorescence as a Potential Biomarker of Oxidative Damage to the Retina"

_antioxidants, 2023, doi:10.3390/antiox12122111_

Round 1

Reviewer 1 Report (New Reviewer)

Comments and Suggestions for Authors

This is a very comprehensive review of lipofuscin, covering in detail the components, oxidation properties, medical significance, autofluorescence, and relationship to sun exposure.

In general, the topic is very interesting and the writing is high quality. I like the critical evaluation of the literature, with comments about difficulties in interpretation or deficiencies in methodology.

My main criticism is that the review is overly lengthy. It includes vast amounts of detail about individual studies, rather than trying to summarize the existing literature. At times, an entire page is devoted to describing one research article (e.g., page 17/18). It is a shame because there are very interesting summary comments, but they get lost in the noise. The reader would probably learn more from a document half as long. Why not add additional tables to summarize some of the study details and use the text to focus more on the author's analysis of the work.

At times, more background is needed to help the reader. Examples: in section 4.3, explaining that Vitamin A and selenium are antioxidants; in dection 4.3.3. adding more context about complement proteins; for Line 466/7 add significance to the statement about experimental results.

It would help to label the perifovea and parafovea on Fig 1 and to explain that they are parts of the macula.

Page 5. Use "photoreceptors" instead of "photoreceptive neurons", as least after first mention.

Page 2. Define ceroid.

Page 6. When describing rod dimensions, need to specify human.

Page 6. Line 283 Use "capuchin monkeys" instead of "cebus"

Page 7. Explain why 11-cis-retinal would be removed from OS discs.

Page 23. Discussion of Vit D should be a separate topic.

Line 1288 Typo: 1,1387

Sections about sun exposure studies are very long and detailed. Please condense.

It was difficult to follow the discussion of autofluorescence and the specific excitation/emission wavelengths. A table would help and some of the detail could be removed from the text.

Overall, it was clear that the author put extensive effort into this review and I would like to see it published. But I think it would have greater impact if somewhat condensed.

Author Response

First of all, I would like to thank Reviewer 1 for the constructive criticism.

This is a very comprehensive review of lipofuscin, covering in detail the components, oxidation properties, medical significance, autofluorescence, and relationship to sun exposure.

In general, the topic is very interesting and the writing is high quality. I like the critical evaluation of the literature, with comments about difficulties in interpretation or deficiencies in methodology.

My main criticism is that the review is overly lengthy. It includes vast amounts of detail about individual studies, rather than trying to summarize the existing literature. At times, an entire page is devoted to describing one research article (e.g., page 17/18). It is a shame because there are very interesting summary comments, but they get lost in the noise. The reader would probably learn more from a document half as long. Why not add additional tables to summarize some of the study details and use the text to focus more on the author's analysis of the work.

I have considerably shortened the chapter 4.6.10. There is no word limit and the reader can choose the sections of their interest and skip other sections. The researchers willing to develop instruments for monitoring oxidation state of lipofuscin and the effects of antioxidant treatments will be interested in details about different retinal fluorophores, their fluorescence characteristics and location in the retina. As for the description of the findings of Pan et al. (2021) starting at the bottom of page 17 in the previous version (bottom of page 22 in the revised version), it is the only paper providing evidence of potential retinal cytotoxicity of lipofuscin that is not induced by light. I intentionally described in detail what the findings were so it is clear they provide just a piece of circumstantial evidence for lipofuscin triggering what authors call “atypical necroptosis.” It cannot be excluded that the lethal effects on photoreceptor and RPE cells are caused by retinaldehydes whereas the phosphorylated MLKL promotes shedding lipofuscin-containing parts of RPE cells as a way of cellular protection from excessive concentrations of lipofuscin that may otherwise trigger harmful effects. Using inhibitors of RPE65 could help with the discrimination of the effects of lipofuscin from the effects of retinaldehydes. I have added that suggestion to the last paragraph of that section.

I have added a table with the main retinal fluorophores, their predominant location, and fluorescence characteristics.

At times, more background is needed to help the reader. Examples: in section 4.3, explaining that Vitamin A and selenium are antioxidants; in dection 4.3.3. adding more context about complement proteins; for Line 466/7 add significance to the statement about experimental results.

As suggested, an explanation of antioxidant roles of selenium and vitamin E has been added; additional information about C3 and MAC, as well as about the competition for lysosomes between phagocytosed POS and endocytosed MAC as a possible scenario responsible for decreased accumulation of lipofuscin has been added; the significance has been added (lines 585-587)

It would help to label the perifovea and parafovea on Fig 1 and to explain that they are parts of the macula.

An additional ring depicting the border between the para- and perifovea has been added and the figure caption has been expanded to make it clear that they are both part of the macula.

Page 5. Use "photoreceptors" instead of "photoreceptive neurons", as least after first mention.

Amended as suggested

Page 2. Define ceroid.

I have changed the wording so it is clear that ceroid refers to lipofuscin-like deposits that rapidly accumulate starting from an early age.

Page 6. When describing rod dimensions, need to specify human.

Amended as suggested (lines 252-3)

Page 6. Line 283 Use "capuchin monkeys" instead of "cebus"

Amended as suggested

Page 7. Explain why 11-cis-retinal would be removed from OS discs.

The explanation has been added (page 9, lines 377-381)

Page 23. Discussion of Vit D should be a separate topic.

The whole section on chronic sunlight exposure as a risk factor for AMD is substantially condensed so vitamin D is just mentioned as a confounding factor in studies of sunlight exposure as a potential risk factor for AMD, which in most studies has not been taken into account.

Line 1288 Typo: 1,1387

This part was removed from the manuscript.

Sections about sun exposure studies are very long and detailed. Please condense.

Condensed as suggested.

It was difficult to follow the discussion of autofluorescence and the specific excitation/emission wavelengths. A table would help and some of the detail could be removed from the text.

A table with the main retinal fluorophores, their predominant location, and fluorescence characteristics has been added. A table does not allow to include all details to make a convincing case that spectral changes of lipofuscin fluorescence excited by two-photon excitation using 920 nm wavelength can be discriminated from fluorescence of other fluorophores so they can serve as a way to monitor oxidative damage to the RPE and the efficacy of treatments with antioxidants.

Overall, it was clear that the author put extensive effort into this review and I would like to see it published. But I think it would have greater impact if somewhat condensed.

Reviewer 2 Report (New Reviewer)

Comments and Suggestions for Authors

The authors extensively presented a updated information regarding the role of lipofuscin in contributing to RPE disfunction and pathogenesis of AMD. They explored the potential application of lipofuscin fluorescence as a biomarker for oxidative damage in RPE and discussed the imaging characteristics of lipofuscin fluorescence in clinical examinations. The review holds significance for eye and vision researchers.

Review Feedback:

1. The review is lengthy, with some redundant content. For instance, Line 184, “Lipofuscin fluorescence”, could be merged with “Fluorescence of RPE lipofuscin” in Line 1439.

2. If content reduction proves challenging, consider dividing the review into two papers—one focusing on basic research and the other on clinical studies.

3.  Line 411, "Structure and composition of RPE lipofuscin," consider creating a diagram illustrating the lipofuscin structure. Additionally, a table listing all lipofuscin compositions could enhance clarity.

4. In Line 527, "Age-related changes in the distribution of lipofuscin in the human retina," please replace "retina" with "RPE" since lipofuscin is specifically observed in RPE, no other retinal cells.

5. Line 548, "Association of lipofuscin accumulation with retinal degenerations," considering to expand this section by incorporating additional references showing the association between lipofuscin accumulation and specific retinal degenerations like AMD.

6. Include information about A2E's effects on RPE death, including various forms of cell death, such as pyroptosis, necroptosis, apoptosis and ferroptosis if  there is any report

Author Response

First of all, I would like to thank Reviewer 2 for the constructive criticism.

Review Feedback:

  1. The review is lengthy, with some redundant content. For instance, Line 184, “Lipofuscin fluorescence”, could be merged with “Fluorescence of RPE lipofuscin” in Line 1439.

The review structure is to discuss lipofuscin and its properties in general including fluorescence which is a characteristic feature of all lipofuscins, and then it focuses more on RPE lipofuscin. At the end of the manuscript I go back to lipofuscins from other cells because the common feature of most lipofuscins are oxidized lipids.

  1. If content reduction proves challenging, consider dividing the review into two papers—one focusing on basic research and the other on clinical studies.

The section on chronic sunlight exposure and AMD has been substantially condensed.

  1. Line 411, "Structure and composition of RPE lipofuscin," consider creating a diagram illustrating the lipofuscin structure. Additionally, a table listing all lipofuscin compositions could enhance clarity.

A diagram was added depicting the formation of lipofuscin from the tip of photoreceptor outer segment, including the formation of A2E and structures of other bisretinoids (page 9; Figure 3). Several pie chart(s) were added showing ratios of chloroform-soluble and insoluble components pf lipofuscin (page 12; Figure 4) and lipid composition (page 13; Figure 5).

  1. In Line 527, "Age-related changes in the distribution of lipofuscin in the human retina," please replace "retina" with "RPE" since lipofuscin is specifically observed in RPE, no other retinal cells.

Corrected so it reads “Age-related changes in the topographical distribution of RPE lipofuscin in the human retina” (line 647-8).

  1. Line 548, "Association of lipofuscin accumulation with retinal degenerations," considering to expand this section by incorporating additional references showing the association between lipofuscin accumulation and specific retinal degenerations like AMD.

The initial reports of increased lipofuscin accumulation in AMD were based on increased fluorescence which later was found to result from stacking of RPE cells. There is no evidence of increased lipofuscin accumulation in AMD but there is an increased number of melanolipofuscin. The appropriate references are already cited.

  1. Include information about A2E's effects on RPE death, including various forms of cell death, such as pyroptosis, necroptosis, apoptosis and ferroptosis if  there is any report

A2E effects on cell death were reported in several papers but in all of them A2E was administered to cultured cells from its solution in an organic solvent.  As described in section 4.4 (top of page 14), A2E is strongly anchored within the lipofuscin granule so the effects it can exert when administered in a solution do not reflect the physiology of the retina where A2E is incorporated into lipofuscin granule. Therefore, the effects of A2E injected into cell culture media as a solution in an organic solvent are not discussed in this review.

Reviewer 3 Report (New Reviewer)

Comments and Suggestions for Authors

This comprehensive article examines into the intricate realms of lipofuscin, exploring its biochemistry and cell biology, shedding light on its role in retinal fluorescence, and discussing its potential as a biomarker for oxidative stress. Encompassing both fundamental research and its translational applications in the clinical setting, the review exhibits an impressive depth and breadth of coverage. Undoubtedly, this work stands as an invaluable resource, poised to significantly contribute to the knowledge base for researchers in the future.

Dr. Rozanowska's meticulous attention to detail is evident, particularly in addressing controversial topics within the field. The review adeptly presents a well-balanced overview, incorporating various perspectives and theories. The article's overall quality is commendable, characterized by well-crafted prose that enhances its informativeness.

Expectations are high for this review to become a cornerstone in the field, attracting numerous citations. I wholeheartedly recommend its publication after minor revisions. While the manuscript is strong, there are a few suggestions and corrections that would further enhance its clarity and precision. These are detailed below for the author's consideration and incorporation into the revision process.

Page 3, line 107: Does this refer to reactive oxygen species?

Page 3, line 145: Replace "mitochondrion" with "mitochondria."

Page 4, Line 151: Use "centers" instead of "centres."

Page 4, Line 184 to 196: In brief, please explain the necessary correction to achieve the appropriate spectral characteristics for a broader readership.

Page 6, line 259: A volume of 324 µm3 for these particles is quite impressive. This is approximately one-third of the volume of columnar RPE cells in the central retina, which are 14 µm tall and 10 µm wide. Is this interpretation correct?

Paragraph 4.3: The general readership would benefit from a brief explanation of vitamin E's role as an antioxidant in these processes. Additionally, it's important to note that vitamin A has significant stores, while this is not the case for vitamin E. Consequently, depletion of vitamin A and E may exhibit different onset times. The reviewer recommends a critical discussion of the data for the general readership rather than simply listing study outcomes.

Page 8, line 347: A schematic representation of A2E production and its relationship with enzymes involved in ocular vitamin A metabolism would be helpful for readers. For instance, the text does not mention the function of rdh12, which is a retinal dehydrogenase. Not all readers may be familiar with this metabolism.

Page 8, line 377: Please provide a brief explanation of the relationship between complement regulatory proteins and phagocytosis.

Paragraph 4.3.4: Start by introducing what macula pigments are since the general readership may not be familiar with their roles and concentrations. In general, consider adding a paragraph at the beginning of your review article to briefly introduce key components discussed later in the text and explain certain terms to the general readership. This could strengthen the paper troughout.

Paragraph 4.4: Enhance the description of the lipofuscin composition with a figure that displays the chemical structures of the major components.

Page 24, 1228: Use "American Indians" or "Indigenous Americans" instead of "Indians."

Paragraph 4.6.11: While these are all valid and critical points, it's important to conclude this paragraph. Does this mean that estimating the contribution of sunlight to the etiology of AMD is challenging or impossible?

Paragraph 4.6.12: The deceleration of dark adaptation in elderly individuals can be attributed to various factors, with retinaldehyde transport being a potential factor, albeit not likely to be a significant contributor. Interestingly, retarding the visual cycle may have a protective effect in preventing light-induced damage, as demonstrated by numerous animal studies.

Page 46, line 2241: The article lacks clarity on the meaning of cRORA. To enhance understanding, it is recommended to provide a comprehensive list of non-standard abbreviations.

Page 53, line 2541: While the reviewer recognizes the reference to a SNP, a brief explanation of rs1061170 is needed for clarity.

Page 53, line 2568 ff: Throughout the chapter, A2E and its oxidation products are consistently portrayed as 'adversaries.' However, the author shifts focus back to all-trans-retinal in this section. Notably, the cited studies predominantly concentrate on reducing A2E, and it is important to mention that deuterated DHA may not inhibit lipid peroxidation but could potentially slow it down.

Page 54, 2619: Once again, the text underscores the potent photosensitizing nature of all-trans-retinal. This is certainly true, particularly in laboratory conditions. However, it is intriguing that a complete bleach does not lead to cell degeneration, considering the approximate 5 mM concentration of rhodopsin in mouse retina. Additionally, the limited exposure of the human retina to light below 400 nm should be acknowledged. Thus, the role of all-trans-retinal as photosensitizer is limited and many researcher in the field agree with this concept.

Conclusion section: The conclusion section appears more as a glimpse into future research directions and the constraints of current studies. To enhance clarity, it is advisable to explicitly distinguish between the outlook and conclusion portions. Furthermore, the conclusion should succinctly highlight the major findings and draw significant conclusions from the research. The reviewer suggests listing a few key conclusions derived from the article for emphasis.

Author Response

First of all, I would like to thank Reviewer 3 for the constructive criticism.

This comprehensive article examines into the intricate realms of lipofuscin, exploring its biochemistry and cell biology, shedding light on its role in retinal fluorescence, and discussing its potential as a biomarker for oxidative stress. Encompassing both fundamental research and its translational applications in the clinical setting, the review exhibits an impressive depth and breadth of coverage. Undoubtedly, this work stands as an invaluable resource, poised to significantly contribute to the knowledge base for researchers in the future.

Dr. Rozanowska's meticulous attention to detail is evident, particularly in addressing controversial topics within the field. The review adeptly presents a well-balanced overview, incorporating various perspectives and theories. The article's overall quality is commendable, characterized by well-crafted prose that enhances its informativeness.

Expectations are high for this review to become a cornerstone in the field, attracting numerous citations. I wholeheartedly recommend its publication after minor revisions. While the manuscript is strong, there are a few suggestions and corrections that would further enhance its clarity and precision. These are detailed below for the author's consideration and incorporation into the revision process.

Page 3, line 107: Does this refer to reactive oxygen species?

The word “oxygen” was inserted between “reactive species” (line 108).

Page 3, line 145: Replace "mitochondrion" with "mitochondria."

Replaced by “a mitochondrion” (line 145).

 Page 4, Line 151: Use "centers" instead of "centres."

“Centres” is the British spelling of American “centers”

 Page 4, Line 184 to 196: In brief, please explain the necessary correction to achieve the appropriate spectral characteristics for a broader readership.

Explained as suggested (lines190-195).

 Page 6, line 259: A volume of 324 µm3 for these particles is quite impressive. This is approximately one-third of the volume of columnar RPE cells in the central retina, which are 14 µm tall and 10 µm wide. Is this interpretation correct?

The estimate refers to peripheral cells overlying 43 photoreceptors. The volume of phagocytosed tips of outer segments by a columnar cell from the fovea overlying only 23 photoreceptors would be 173 µm3 which corresponds to about 16% of cell body volume. Thank you for the idea to talk about it as a percentage of cell volume. I have added it to the end of section 4.2 (lines 266-269).

 Paragraph 4.3: The general readership would benefit from a brief explanation of vitamin E's role as an antioxidant in these processes. Additionally, it's important to note that vitamin A has significant stores, while this is not the case for vitamin E. Consequently, depletion of vitamin A and E may exhibit different onset times. The reviewer recommends a critical discussion of the data for the general readership rather than simply listing study outcomes.

The antioxidant role of vitamin E is now briefly explained. Elements of discussion have been added to that section. Indeed, after 26 weeks of dietary depletion, vitamin E was no longer detectable in the retina, whereas levels of vitamin A were depleted by 75-90%.

 Page 8, line 347: A schematic representation of A2E production and its relationship with enzymes involved in ocular vitamin A metabolism would be helpful for readers. For instance, the text does not mention the function of rdh12, which is a retinal dehydrogenase. Not all readers may be familiar with this metabolism.

Schematic diagrams depicting a part of the outer segment disc (Fig. 2) and A2E formation (Fig. 3) have been added. The localization and function of RDH12 have been added in lines 386-388, page 9.

 Page 8, line 377: Please provide a brief explanation of the relationship between complement regulatory proteins and phagocytosis.

As suggested, the section was expanded to suggest a mechanism by which preventing MAC formation can prevent the formation of lipofuscin (lines 450-460 in page 10-11)

 Paragraph 4.3.4: Start by introducing what macula pigments are since the general readership may not be familiar with their roles and concentrations. In general, consider adding a paragraph at the beginning of your review article to briefly introduce key components discussed later in the text and explain certain terms to the general readership. This could strengthen the paper troughout.

The caption of Fig. 1 has been expanded and Figure 2 has been added.

 Paragraph 4.4: Enhance the description of the lipofuscin composition with a figure that displays the chemical structures of the major components.

Figures 3-5 have been added.

 Page 24, 1228: Use "American Indians" or "Indigenous Americans" instead of "Indians."

Amended as suggested (line 1952).

 Paragraph 4.6.11: While these are all valid and critical points, it's important to conclude this paragraph. Does this mean that estimating the contribution of sunlight to the etiology of AMD is challenging or impossible?

As suggested, the concluding statement has been added (lines 1371-1377).

 Paragraph 4.6.12: The deceleration of dark adaptation in elderly individuals can be attributed to various factors, with retinaldehyde transport being a potential factor, albeit not likely to be a significant contributor. Interestingly, retarding the visual cycle may have a protective effect in preventing light-induced damage, as demonstrated by numerous animal studies.

Indeed, there are many factors that can affect dark adaptation in the elderly. One of the potential factors could be the inadequate supply of 11-cis-retinaldehyde to visual pigments.  This hypothesis of decreased availability of 11-cis-retinaldehyde was investigated by Hanneken et al. (2017) who found no difference in the ability of normal and AMD eyes to recover all rhodopsin during dark-adaptation post mortem suggesting that availability of 11-cis-retinal is not a limiting factor. It also suggests that the decreased permeability of thickened Bruch’s membrane to RBP-bound all-trans-retinol diffusing from the fenestrated choriocapillaris does not cause a decrease in the total amount of retinoids involved in the visual cycle below the amount of opsin. However, they allowed several hours for the dark adaptation to occur so delayed synthesis and/or trafficking of 11-cis-retinaldehydes in AMD retina cannot be excluded.

The mismanaged outer segments can be expected to impair the transport of not only retinoids but also ATP and NADPH as well as proteins involved in the phototransduction cascade or its deactivation that translocate to/from the inner/outer segment during adaptation to bright light, but are needed, depending on the protein, either inside or outside the outer segment to achieve the amazing sensitivity characteristic for the dark-adapted retina.

Inhibition of 11-cis-retinaldehyde synthesis has a protective effect demonstrated in numerous studies because it diminishes the accumulation of both retinaldehydes during light exposure thereby preventing their (photo)toxic effects. I have added that information to Fig. 2 caption.

The delayed dark adaptation due to the impaired trafficking of retinoids, ATP, and/or NADPH is likely to increase the susceptibility to light-induced retinal injury mediated by retinaldehydes.

Hanneken, A.; Neikirk, T.; Johnson, J.; Kono, M. Biochemical Measurements of Free Opsin in Macular Degeneration Eyes: Examining the 11-CIS Retinal Deficiency Hypothesis of Delayed Dark Adaptation (An American Ophthalmological Society Thesis). Trans Am Ophthalmol Soc 2017, 115, T1.

Page 46, line 2241: The article lacks clarity on the meaning of cRORA. To enhance understanding, it is recommended to provide a comprehensive list of non-standard abbreviations.

The abbreviation cRORA is defined the first time it is mentioned (line 2211) and is in the abbreviation list (at the end of the manuscript, just before the reference list)

 Page 53, line 2541: While the reviewer recognizes the reference to a SNP, a brief explanation of rs1061170 is needed for clarity.

Explained as suggested (line 2531).

 Page 53, line 2568 ff: Throughout the chapter, A2E and its oxidation products are consistently portrayed as 'adversaries.' However, the author shifts focus back to all-trans-retinal in this section. Notably, the cited studies predominantly concentrate on reducing A2E, and it is important to mention that deuterated DHA may not inhibit lipid peroxidation but could potentially slow it down.

In all three instances where A2E and its oxidation products are mentioned, I have now reiterated that the results of experiments where A2E is injected into a cell culture medium from a solution in organic solvent need to be treated with caution because it is not known if the same effects could be caused by A2E embedded into lipofuscin granule (lines 2472-2475, 2491-2492, 2511-2626). A2E and other bisretinoids are formed as a result of delayed clearance of retinaldehydes so it is not easy to discriminate whether the deleterious effects are due to bisretinoids or their precursors.

If the Reviewer considers the inhibition of lipid peroxidation as the initiation step, then indeed, depending on the initiator, there may be no difference between DHA and deuterated DHA. It is likely that due to its high oxidation potential, the hydroxyl radical can initiate lipid peroxidation by abstracting deuterium from deuterated DHA as easily as abstracting hydrogen from normal DHA. It is not clear whether the hydroperoxyl radical, that has lower oxidation potential, can abstract deuterium as easily from deuterated DHA as hydrogen from normal DHA. It is thought that the huge inhibition of lipid peroxidation by deuterated DHA occurs at the propagation step: the deuterium on the methylene group is less susceptible to abstraction by a lipid-derived peroxyl radical than a hydrogen on such a methylene group. As a result, it gives time for antioxidants, such as vitamin E, to scavenge the lipid-derived peroxyl radical or for the peroxyl radical to bump into another peroxyl radical resulting in the termination of the chain reaction. I have added that information on page 57, line 2574 and 2578-2583).

 Page 54, 2619: Once again, the text underscores the potent photosensitizing nature of all-trans-retinal. This is certainly true, particularly in laboratory conditions. However, it is intriguing that a complete bleach does not lead to cell degeneration, considering the approximate 5 mM concentration of rhodopsin in mouse retina. Additionally, the limited exposure of the human retina to light below 400 nm should be acknowledged. Thus, the role of all-trans-retinal as photosensitizer is limited and many researcher in the field agree with this concept.

We have presented results at ARVO 2018 showing that exposure of dark-adapted wild-type mice to light that results in absorption of similar numbers of photons as in the abca4(-/-)rdh8(-/-) double knockout mice leads to the total loss of OCT band corresponding to the outer nuclear layer whereas the same light exposure given after pre-exposure to non-damaging light allowing for clearance of more than 90% of retinaldehydes produces no detectable damage (as evaluated by OCT and SLO).

Fig. 1 includes transmission of different spectral bands to the retina and the caption includes a reference to a paper showing that the 320 nm transmission window persists even in old adults. A figure was added (Fig. 6) showing absorption spectra of all-trans-retinaldehyde and lipophilic component of lipofuscin as well as action spectra of photooxidation mediated by lipofuscin granules, as well as and their lipophilic and chloroform-insoluble components. Due to spectral changes in the transmission properties of the lens and lipofuscin accumulation with age, it is likely that the relative contribution of retinaldehydes to light-induced retinal toxicity decreases while the lipofuscin contribution increases.

Accumulation of bisretinoids in the RPE is the best evidence for the accumulation of free retinaldehydes in the retina (reviewed in [107]). Recent experimental evidence adds further support for it. In photoreceptor outer segments, retinaldehydes are in proximity to abundant phoshatidylethanolamine (PE) which outnumbers retinaldehyde at least in a 100:1 ratio [127]. The condensation of retinaldehyde with PE forming NRPE is fast as opposed to the slow reaction leading to the formation of bisretinoids ([125]; Xu et al., 2023). When free retinaldehyde is present, its exposure to UV or blue light leads to the efficient generation of singlet oxygen with quantum yields exceeding several-fold the quantum yields of singlet oxygen generation by the lipophilic components of lipofuscin [107].

Rozanowska, M.B.; Golczak, M., Maeda, A.; Palczewski, K. Role of Short-Term Light Adaptation Enabling Clearance of Retinaldehydes in Modulating Retinal Susceptibility to Light-Induced Injury. Investigative Ophthalmology & Visual Science 2018, Vol.59, 4978. ARVO Abstract.

Xu, T.; Molday, L.L.; Molday, R.S. Retinal-phospholipid Schiff-base conjugates and their interaction with ABCA4, the ABC transporter associated with Stargardt disease. J Biol Chem 2023, 299, 104614, doi:10.1016/j.jbc.2023.104614.

Conclusion section: The conclusion section appears more as a glimpse into future research directions and the constraints of current studies. To enhance clarity, it is advisable to explicitly distinguish between the outlook and conclusion portions. Furthermore, the conclusion should succinctly highlight the major findings and draw significant conclusions from the research. The reviewer suggests listing a few key conclusions derived from the article for emphasis.

As suggested, the Conclusion section was split into Conclusions and Future Research Directions.

Round 2

Reviewer 2 Report (New Reviewer)

Comments and Suggestions for Authors

no further concern. 

This manuscript is a resubmission of an earlier submission. The following is a list of the peer review reports and author responses from that submission.

Round 1

Reviewer 1 Report

Comments and Suggestions for Authors

This manuscript represents a prodigious achievement by a single author (>400 references) advocating for better ophthalmic imaging technology to incorporate old and new evidence that fundus autofluorescence emission spectra changes with aging and age-related macular degeneration (AMD), a major disease of the photoreceptor support system impacting vision. 

In brief, the retinal pigment epithelium (RPE) is a layer of nurse cells that maintain the photoreceptors internal to it and the choriocapillaris endothelium external to it. Due to the daily ingestion of membranes in photoreceptor outer segments (10% of the tips), the RPE accumulates a distinctive lipofuscin over the lifespan, starting in prenatal life, and this is visible in vivo as fundus autofluorescence. Signal sources for autofluorescence in addition to lipofuscin is melanolipofuscin and melanosomes, all organelles of lysosomal origin. The author reviews concepts of autofluorescence signal generation, lipofuscin and related organelles, spectral analysis, experimental systems for studying inclusions in cells, clinical autofluorescence imaging studies, trials in progress for AMD, among other topics.  

The part of the current manuscript with most potential is a synthesis of recent data from hyperspectral and lifetime autofluorescence imaging suggesting that emission spectra from RPE lipofuscin may drift to shorter wavelengths with age. The author is encouraged to focus and refine the review on this novel topic. Ideally the reader would come away with key questions and standardized and quantitative approaches for future research. The author is also encouraged to consult and refer readers to a recent authoritative review of clinical autofluorescence as applied to multiple retinal diseases [1].

Other sections of the manuscript are problematic. To support the utility of spectrally specific autofluorescence imaging, the author invokes models of pathogenesis that have not borne out well over time. Original ideas about RPE lipofuscin came from low spatial resolution assays of human eyes that found abundant bisretinoids in whole eye cups, model systems lacking a foveate retina (mice) or RPE polarity (cell lines), and misconceptions about the distribution of light on the retina. The author covers at length several long-standing concepts that are poorly supported by analyses of human eyes, raising questions of relevance to human health.

1.     A2E is the first discovered bisretinoid and most investigated in model systems. Starting in 2009 [2] it has been repeatedly shown by multiple labs using chromatography and mass spectrometry to be highest in peripheral retina, i.e., millimeters away from the part of retina affected by AMD (summarized by reference 113). Whole eye cup extractions used by prior authors were dominated by signal in peripheral retina. In mice the distribution of A2E is relatively homogenous. So, how A2E in the retinal periphery relates to a macular disease should be explained. Further, existing data indicate on oxidation products of A2E track the distribution of the native compound, i.e., they are also abundant in peripheral retina [3,4]. This argues against A2E being selectively oxidized and arguing for fundamental differences between central and peripheral retina.

2.     Ambient light is evenly distributed across the human retina to 50° eccentricity [5,6]. The schematic of light focusing on the fovea in central retina (Figure 1) and others like it in multiple journals is incorrect. Long-term studies in patient populations have not supported a role of light exposure in AMD progression [7-10]. Cell culture studies using strong light, described extensively in the current manuscript, are of questionable relevance to outer retinal physiology. An early paper on histologic autofluorescence quantification (reference 159) cited the light distribution studies and downplayed the role of light.  So, this topic should be re-thought. 

3.     The conclusion “Oxidized docosahexaenoic acid… a likely component of lipofuscin…” (and therefore spectral shifts are indicating oxidative damage to this vulnerable fatty acid) is not consistent with long-standing data on lipofuscin lipid composition. According to Bazan et al [11], lipofuscin is enriched in palmitic acid (16:0), arachidonic acid (20:4) and oleic acid (18:1) in contrast to photoreceptor outer segments, which are enriched in docosahexaenoic acid (22:6). Finally, not all outer segment material ends up in lipofuscin, as outer segment membrane lipids undergo to undergo fatty acid beta-oxidation to fuel the RPE (“metabolic ecosystem”) to spare glucose for the photoreceptors [12]. These new concepts of outer retinal physiology impacting the source material for lipofuscin should be considered. 

I offer a few specific comments, referring to pages in the reviewer pdf. Please note that references after 26 are off by one, because of an extra carriage return in reference 26. In general, the writing style could benefit from editing for conciseness and organization. 

p. 2, Among them, there is a group of neurodegenerative genetic disorders called neuronal ceroid lipofuscinosis (NCL), which is characterized by the formation of intracellular deposits in neurons and other cell types. 

NCL is genetically inherited disease. M.L. Katz wrote a wonderful review that distinguished between RPE lipofuscin and inclusions due to storage disorders such as NCL. He emphasized that the lipofuscin in each cell type containing is unique to that cell type [13], and he specifically distinguished between brain and RPE lipofuscin. Text on p. 3 mixes together lipofuscin from different organ systems. 

p. 4, In cases where care was taken to consider if the spectral correction was needed, it has been shown that the spectral range of fluorescence emission varies depending on the tis-sue from which the lipofuscin is derived. 

This is a good point and could be used by the author to promulgate standardized analysis of spectra.

p. 4, autoxidation of docosahexaenoic acid (DHA) results in the formation of products with an emission maximum at about 610 nm when excited with 488 nm light and broad-band fluorescence with a maximum in blue when excited with UV light (74).

See comments above re DHA.

Figure 1: Retinal illuminance is homogenous to 50° eccentricity so the beam of light on the fovea needs to be fixed (see above). The macula lutea is 3 mm diameter, the central area is 6 mm diameter [14]. The figure needs a scale bar.

p. 6, These structural differences in POS of rods and cones may have important consequences for their phagocytosis and lysosomal degradation: in the case of rods, RPE phagocytoses the oldest discs, whereas, in the case of cones, the phagocytosed material is a mixture of old and recently added lipids and proteins. 

Reference for this information? Does not sound right to me. I did not see the classics of photoreceptor cell biology cited. 

Table 1. Studies of the relationship of fundus fluorescence with age – this is potentially an interesting table and could be greatly improved for impact. Please divide ex vivo and in vivo fluorescence detection studies. Please also indicate the detection technology and the retinal area sampled. Assess each study on a standard set of parameters so that it is easier to compare. Important contributions by Kleefeldt et al and Probster et al to lifespan autofluorescence should be included [15,16]. What is the inner and outer ring in the last entry? 

p. 10 ff – most of the described cell culture studies in this section used non-phenotypic RPE, by today’s standards. When RPE in culture is polarized and joined by continuous tight junctions in a geometrically precise array, it is resistant to stress [17,18]. Few of the cited studies used cells meeting these standards. The lighting regimens were harsh and non-physiologic. Blue (and violet) wavelengths are the part of the action spectrum for opsins (rhodopsin and non-canonical opsins). Most used fluorescence microscopy to assess inclusions, few if any used transmission electron microscopy in addition to confirm the physical form of the autofluorescent material. It is not helpful to describe all these studies without appropriate evaluations, disclaimers, and comparisons to RPE in vivo/ in situ/ primary/ explants. Much of this section could be deleted in my opinion.

p. 11, It remains to be shown whether these A2E oxidation products can stimulate these deleterious effects in vivo or whether they are safely trapped in the lipofuscin granule. It has been reported that oxidized A2E can react with other oxidized A2E molecules or A2E itself forming high-molecular weight products, which are more hydrophobic than A2E, and therefore more likely to remain in the granule (204,205). 

This is a good point that suggests the author’s ambivalence about these model system studies. 

p. 12, While it is clear that lipofuscin exhibits photosensitizing properties and can affect cell function and viability in vitro upon exposure to light, the evidence for the contribution of lipofuscin to light-induced retinal injury in vivo is rather limited.

Does the author consider these studies valid? They are discussed at length. Not clear to me.

p. 30, Marmorstein et al. compared fluorescence properties …

Use of the terms autofluorescence vs fluorescence is inconsistent.

p. 30, However, in contrast to normal eyes, where the intensity maxima were similar for Bruch’s membrane and RPE, in AMD eyes the intensity maximum was about 1.8-fold greater for Bruch’s membrane than for RPE. The emission spectrum for sub-RPE deposits was similar in shape but the emission maximum was about twice smaller than for Bruch’s membrane. The emission maximum of the RPE fluorescence induced by excitation with 364 nm was at 540 nm, which is 15 nm shifted towards shorter wavelengths in comparison with the normal retina.

This text is difficult to read for a sense of quantitative relationships. The directions of the comparisons vary from one sentence to another. Please rephrase, e.g., group 1 > group 2 > group 3. Or something like that. 

p. 17, Because bisretinoids are the major emitter of lipofuscin fluorescence (105,106,145), it is plausible that some of the observed changes in fluorescence during lipofuscin photo-degradation can be ascribed to light-induced oxidation of bisretinoids, which are prone to oxidative degradation. 

Please comment how bisretinoids abundant in far peripheral retina, where the light levels are lower than in central retina, could be subject to photo-oxidation (see references above). I don’t see how this is possible. They can be photo-oxidized in vitro. 

Figure 3. Top panel: fluorescence of lipofuscin-laden ARPE-19 cell monolayer kept in the dark )left panel) or exposed daily for 14 days to 45 minutes of 9.76 mW/cm2 visible light providing a total dose of 369 J/cm2 (right panel). 

This figure show non-phenotypic cells. Were assays performed to verify RPE-specific phenotype? (e.g., gene expression, transepithelial resistance, calcium signaling, other [19]). Are junctional complexes intact? Was the fluorescent material investigated with high-resolution microscopy techniques? 

p. 20, Due to the high concentration of polyunsaturated fatty acids, especially in photoreceptor outer segments, lipofuscin, drusen, and Bruch’s membrane it can be expected that products of lipid oxidation and their adducts with proteins can contribute to retinal fluorescence (68-70,74,140,248). 

Author should be aware that studies of Bruch’s membrane lipids, verified as specific to Bruch’s by way of lipid histochemistry, have shown a high concentration of linoleate and low concentration of DHA [20,21]. So whatever autofluorescence appears in Bruch’s in studies like Marmorstein et al cannot be assumed to derive from DHA. 

p. 21, Sub-retinal and sub-RPE deposits, which accumulate with age and some of them are a hallmark of AMD, such as reticular pseudodrusen, basal laminar deposits, basal linear deposits and drusen, can fluorescence (275-277). 

can emit fluorescence signal … 

p. 23, Moreover, this filtering effect is likely to change with ageing because i) there is an overall loss of melanin; ii) there is an accumulation of lipofuscin and complex granules containing both melanin and lipofuscin; and iii) melanosomes redistribute from their initial location in the apical portion of the RPE and lipofuscin redistributes from its initial location at the basal site to become distributed more evenly throughout the cell. 

The evidence for loss of melanin with age comes from studies with an unknown degree of melanosome loss, to my understanding. Most RPE melanosomes are in the delicate apical processes [22,23], which may have been lost in the processing of tissues for these studies. This limitation should be addressed. 

p. 29, and is most pronounced in people of White ethnicity (284,335,337,338) (Table 1). 

The sample of other races is small, unfortunately. 

p. 33, damaged photoreceptors and absent RPE: there is no characteristic hypo-reflective layer corresponding to the outer nuclear layer and no characteristic hyper-reflective layers corresponding to the outer limiting membrane and the ellipsoid area of the inner segment. 

OCT has bands not layers – please include (or refer readers to) appropriate references. 

p. 33, It has been concluded that the residual debris and drusen-like deposits are the origins of the green-emitting fluorophores …

What is a “drusen-like deposit” - do you mean basal laminar deposit? Or drusen. These are two separate layers. Separation of BLamD from drusen by electron microscopy (by the Sarks) was an important milestone in AMD pathology. Author may find these recent papers on AMD deposits (and references within) of use. [24,25] All these layers are visible clinically so please mention carefully. 

p. 36, Feldman et al. reported fluorescence spectra of suspensions of RPE cells from two normal eyes and two AMD eyes {371}. 

Were these macular RPE cells or whole eye cup extractions?

p. 38, Secondly, in comparison with age-matched normal human retinas, AMD-affected retinas exhibit a greater level of protein modifications by oxidized lipids, such as CEP which has been detected in photoreceptor outer segments, RPE, Bruch’s membrane and drusen (395,396).

Please see comment above about low levels of DHA and high levels of linoleate in Bruch’s membrane. CEP adducts were diffusely distributed and not confined to Bruch’s or drusen, as were other lipids ostensibly of lipoprotein origin. 

1.         Schmitz-Valckenberg S, Pfau M, Fleckenstein M, Staurenghi G, Sparrow JR, Bindewald-Wittich A, Spaide RF, Wolf S, Sadda S, Holz FG. Fundus autofluorescence imaging. Prog Retin Eye Res. 2020;81:100893. PMID 32758681 

2.         Bhosale P, Serban B, Bernstein PS. Retinal carotenoids can attenuate formation of A2E in the retinal pigment epithelium. Archives of Biochemistry and Biophysics. 2009;483(2):175-181. PMID 18926795 

3.         Kotnala A, Senthilkumari S, Gong W, Stewart TG, Curcio CA, Halder N, Kumar A, Velpandian T. Retinal pigment epithelium in human donor eyes contains higher levels of bisretinoids including A2E in periphery than macula. Invest Ophthalmol Vis Sci. 2022;63(6):6. PMID 35671050 

4.         Ablonczy Z, Higbee D, Grey AC, Koutalos Y, Schey KL, Crouch RK. Similar molecules spatially correlate with lipofuscin and N-retinylidene-N-retinylethanolamine in the mouse but not in the human retinal pigment epithelium. Arch Biochem Biophys. 2013;539(2):196-202. PMID 23969078 

5.         Pflibsen KP, Pomerantzeff O, Ross RN. Retinal illuminance using a wide-angle model of the eye. Journal of the Optical Society of America. A, Optics and image science. 1988;5(1):146-150. PMID 3351651 

6.         Kooijman AC. Light distribution on the retina of a wide-angle theoretical eye. Journal of the Optical Society of America. 1983;73(11):1544-1550. PMID 6644400 

7.         Delcourt C, Cougnard-Gregoire A, Boniol M, Carriere I, Dore JF, Delyfer MN, Rougier MB, Le Goff M, Dartigues JF, Barberger-Gateau P, Korobelnik JF. Lifetime exposure to ambient ultraviolet radiation and the risk for cataract extraction and age-related macular degeneration: the Alienor Study. Invest Ophthalmol Vis Sci. 2014;55(11):7619-7627. PMID 25335979 

8.         Zhou H, Zhang H, Yu A, Xie J. Association between sunlight exposure and risk of age-related macular degeneration: a meta-analysis. BMC Ophthalmol. 2018;18(1):331. PMID 30572865 

9.         Lee JS, Li PR, Hou CH, Lin KK, Kuo CF, See LC. Effect of blue light-filtering intraocular lenses on age-related macular degeneration: A nationwide cohort study with 10-year follow-up. Am J Ophthalmol. 2021;234:138-146. PMID 34411525 

10.       Mainster MA, Findl O, Dick HB, Desmettre T, Ledesma-Gil G, Curcio CA, Turner PL. The blue-light-hazard vs. the blue-light-hype. Am J Ophthalmol. Aug 2022;240:51-57. PMID 35227699 

11.       Bazan HEP, Bazan NG, Feeney-Burns L, Berman ER. Lipids in human lipofuscin-enriched subcellular fractions of two age populations. Comparison with rod outer segments and neural retina. Invest. Ophthalmol. Vis. Sci. 1990;31:1433-1443. PMID 2387677 

12.       Reyes-Reveles J, Dhingra A, Alexander D, Bragin A, Philp NJ, Boesze-Battaglia K. Phagocytosis dependent ketogenesis in retinal pigment epithelium. J Biol Chem. Mar 16 2017;292(19):8038-8047. PMID 28302729 

13.       Katz ML, Robison WG, Jr. What is lipofuscin? Defining characteristics and differentiation from other autofluorescent lysosomal storage bodies. Arch Gerontol Geriatr. 2002;34(3):169-184. PMID 14764321 

14.       Polyak SL. The Retina. Chicago: University of Chicago; 1941.

15.       Pröbster C, Tarau I-S, Berlin A, Kleefeldt N, Hillenkamp J, Nentwich MM, Sloan KR, Ach T. Quantitative fundus autofluorescence in the developing and maturing healthy eye. Translational Vision Science & Technology. 2021;10(2):15. PMID 34003900 

16.       Kleefeldt N, Bermond K, Tarau I-S, Hillenkamp J, Berlin A, Sloan KR, Ach T. Quantitative fundus autofluorescence: advanced analysis tools. Translational Vision Science & Technology. 2020;9(8):2. PMID 32855849 

17.       Thurman JM, Renner B, Kunchithapautham K, Ferreira VP, Pangburn MK, Ablonczy Z, Tomlinson S, Holers VM, Rohrer B. Oxidative stress renders retinal pigment epithelial cells susceptible to complement-mediated injury. The Journal of biological chemistry. 2009;284(25):16939-16947. PMID 19386604 

18.       Zhang Q, Presswalla F, Calton M, Charniga C, Stern J, Temple S, Vollrath D, Zacks DN, Ali RR, Thompson DA, Miller JML. Highly differentiated human fetal RPE cultures are resistant to the accumulation and toxicity of lipofuscin-like material. Invest Ophthalmol Vis Sci. 2019;60(10):3468-3479. PMID 31408109 

19.       Miyagishima KJ, Wan Q, Corneo B, Sharma R, Lotfi MR, Boles NC, Hua F, Maminishkis A, Zhang C, Blenkinsop T, Khristov V, Jha BS, Memon OS, D'Souza S, Temple S, Miller SS, Bharti K. In pursuit of authenticity: induced pluripotent stem cell-derived retinal pigment epithelium for clinical applications. Stem Cells Transl Med. 2016;5(11):1562-1574. PMID 27400791 

20.       Wang L, Li C-M, Rudolf M, Belyaeva OV, Chung BH, Messinger JD, Kedishvili NY, Curcio CA. Lipoprotein particles of intra-ocular origin in human Bruch membrane: an unusual lipid profile. Invest Ophthalmol Vis Sci. 2009;50:870-877. PMID 18806290 

21.       Bretillon L, Thuret G, Gregoire S, Acar N, Joffre C, Bron AM, Gain P, Creuzot-Garcher CP. Lipid and fatty acid profile of the retina, retinal pigment epithelium/choroid, and the lacrimal gland, and associations with adipose tissue fatty acids in human subjects. Exp Eye Res. 2008;87(6):521-528. PMID 18801361 

22.       Anderson DH, Fisher SK, Erickson PA, Tabor GA. Rod and cone disc shedding in the rhesus monkey retina: a quantitative study. Exp. Eye Res. May 1980;30(5):559-574. PMID 7409012 

23.       Pollreisz A, Neschi M, Sloan KR, Pircher M, Mittermueller TJ, Dacey DM, Schmidt-Erfurth U, Curcio CA. An atlas of human retinal pigment epithelium organelles significant for clinical imaging. Invest Ophthalmol Vis Sci. 2020;61(8):13. PMID 32648890 

24.       Sura AA, Chen L, Messinger JD, Swain TA, McGwin Jr G, Freund KB, Curcio CA. Measuring the contributions of basal laminar deposit and Bruch’s membrane in age-related macular degeneration. Invest Ophthalmol Vis Sci. 2020;61(13):19. PMID 33186466 

25.       Chen L, Messinger JD, Kar D, Duncan JL, Curcio CA. Biometrics, impact, and significance of basal linear deposit and subretinal drusenoid deposit in age-related macular degeneration. Invest. Ophthalmol. Vis. Sci. 2021;62(1):33. PMID 33512402 

Comments on the Quality of English Language

The manuscript is very long and could be edited for conciseness. I wondered about the value of many of the studies being cited, as mentioned. 

Author Response

First of all, I would like to thank Reviewer 1 for their constructive criticism and for raising several important issues that led to improvements of the manuscript and clarifying some misconceptions.

This manuscript represents a prodigious achievement by a single author (>400 references) advocating for better ophthalmic imaging technology to incorporate old and new evidence that fundus autofluorescence emission spectra changes with aging and age-related macular degeneration (AMD), a major disease of the photoreceptor support system impacting vision. 

In brief, the retinal pigment epithelium (RPE) is a layer of nurse cells that maintain the photoreceptors internal to it and the choriocapillaris endothelium external to it. Due to the daily ingestion of membranes in photoreceptor outer segments (10% of the tips), the RPE accumulates a distinctive lipofuscin over the lifespan, starting in prenatal life, and this is visible in vivo as fundus autofluorescence. Signal sources for autofluorescence in addition to lipofuscin is melanolipofuscin and melanosomes, all organelles of lysosomal origin. The author reviews concepts of autofluorescence signal generation, lipofuscin and related organelles, spectral analysis, experimental systems for studying inclusions in cells, clinical autofluorescence imaging studies, trials in progress for AMD, among other topics.  

The part of the current manuscript with most potential is a synthesis of recent data from hyperspectral and lifetime autofluorescence imaging suggesting that emission spectra from RPE lipofuscin may drift to shorter wavelengths with age. The author is encouraged to focus and refine the review on this novel topic. Ideally the reader would come away with key questions and standardized and quantitative approaches for future research. The author is also encouraged to consult and refer readers to a recent authoritative review of clinical autofluorescence as applied to multiple retinal diseases [1].

 My intention for the review was to provide an update not just on lipofuscin fluorescence but also other aspects discussed in the manuscript. Some research areas have not progressed much since other comprehensive reviews were published so these sections of the manuscript may appear shorter in comparison with others.

I have also expanded the conclusions section with some open questions and proposing new avenues for future research.

The reference suggested by Reviewer 1 was already cited in section 6.2. I have added another citation of this reference at the beginning of this section.

Other sections of the manuscript are problematic. To support the utility of spectrally specific autofluorescence imaging, the author invokes models of pathogenesis that have not borne out well over time. Original ideas about RPE lipofuscin came from low spatial resolution assays of human eyes that found abundant bisretinoids in whole eye cups, model systems lacking a foveate retina (mice) or RPE polarity (cell lines), and misconceptions about the distribution of light on the retina. The author covers at length several long-standing concepts that are poorly supported by analyses of human eyes, raising questions of relevance to human health.

 I appreciate all the points raised by the reviewer and addressed them in the revised manuscript as detailed below.

  1. A2E is the first discovered bisretinoid and most investigated in model systems. Starting in 2009 [2] it has been repeatedly shown by multiple labs using chromatography and mass spectrometry to be highest in peripheral retina, i.e., millimeters away from the part of retina affected by AMD (summarized by reference 113). Whole eye cup extractions used by prior authors were dominated by signal in peripheral retina. In mice the distribution of A2E is relatively homogenous. So, how A2E in the retinal periphery relates to a macular disease should be explained. Further, existing data indicate on oxidation products of A2E track the distribution of the native compound, i.e., they are also abundant in peripheral retina [3,4]. This argues against A2E being selectively oxidized and arguing for fundamental differences between central and peripheral retina.

I am not aware of any evidence that A2E in the retinal periphery relates to a macular disease so I cannot fulfil the Reviewer’s request to explain how it does so.

Thank you for pointing me to reference 2, the paper by Bhosale et al. (2009) (Bhosale, Serban, and Bernstein 2009). The data on A2E content from peripheral and macular areas of 2- and 9-year-old monkeys presented in that paper suggest that at an early age A2E accumulates faster in the macula than in the mid-periphery, but at the age of 9 years, the differences are less pronounced. These data support the hypotheses that A2E in lipofuscin undergoes oxidative degradation with age and that the macula is under increased oxidative stress in comparison with the periphery. I have added this and other relevant information from the paper in section 7.2 and cited it in sections 4.4 and 7.2.

In most experiments on mice, A2E was extracted from the entire eyecup and no attempts were undertaken to evaluate its distribution across the retina. The distribution of A2E across the retina of 6-month-old Sv127 mouse obtained by MALDI imaging by Ablonczy et al. appears to be non-uniform with a focal increase in one retinal area (Figure 3 in (Ablonczy, Higbee, Grey, et al. 2013)).

The existing data indicate only that the very early oxidation products track the distribution of A2E. The studies [3,4] cited by the Reviewer quantified only A2E and A2E oxidation products formed by the addition of one or two oxygen atoms forming monofuranoid oxide of A2E and presumably 7,8,7’,8’-bis-epoxide (Kotnala et al. 2022; Ablonczy, Higbee, Anderson, et al. 2013; Ablonczy, Higbee, Grey, et al. 2013). Ben Shabat et al. (2002) and Gaillard et al. (Ben-Shabat et al. 2002; Gaillard et al. 2004; Avalle et al. 2004) reported various other A2E photooxidation resulting from the sequential addition of more oxygen atoms to A2E and formation of various epoxides, including nonaepoxide. Gaillard group reported A2E photooxidation and autooxidation leading to the formation of various scission products, including reactive carbonyls, bis-furanoid oxide of A2E, mono-furanoid oxide of A2E with the second oxygen addition to the cyclohexynyl ring, as well as high-molecular-weight aggregates (Dillon et al. 2004; Wang et al. 2006; Murdaugh et al. 2011; Murdaugh et al. 2010). Because of increased exposure to light (Sliney 2005), the environment of the central retina is more oxidizing than in the peripheral retina, so it can be expected that the central retina will be depleted from A2E and its early oxidation products faster than the peripheral retina. This is now discussed in section 7.2.

I agree that there are fundamental differences between the central and peripheral retina including the distribution of neurons and their morphology, and the distribution of incident light. Of particular relevance to the review is the length of the rod outer segment and the time needed for its renewal, which varies across different retinal areas. This information has been added as a new section 4.3.4 of the manuscript. The section on light distribution has been added also as a new section 4.6.11 How much sunlight reaches the retina?

  1. Ambient light is evenly distributed across the human retina to 50° eccentricity [5,6]. The schematic of light focusing on the fovea in central retina (Figure 1) and others like it in multiple journals is incorrect. Long-term studies in patient populations have not supported a role of light exposure in AMD progression [7-10]. Cell culture studies using strong light, described extensively in the current manuscript, are of questionable relevance to outer retinal physiology. An early paper on histologic autofluorescence quantification (reference 159) cited the light distribution studies and downplayed the role of light.  So, this topic should be re-thought. 

Thank you to the Reviewer for raising the important point of light distribution across the retina that is highly relevant to lipofuscin formation and its potential phototoxicity and photodegradation. The refractive power of the human eye is a well-established fact, and it is measured routinely in every typical eye examination. Pflibsen at al. and Kooijman, the authors of papers cited by the Reviewer, calculated/measured retinal illuminance from a light source providing uniform luminance across the entire hemisphere surrounding the eye. This type of illumination (known as Ganzfeld illumination) was designed to provide equal illumination of the spots on the retina while testing the topographical sensitivity to flashes of light appearing across the visual field.

Under physiological conditions, the radiance (and consequently luminance) across the visual field varies considerably. Overall, the central retina is exposed to much greater levels of light than other parts (Sliney 2005) but no attempt has been undertaken to quantitatively evaluate the topography of retinal irradiance encountered during different daily activities at different times of the day at different seasons, atmospheric conditions, and latitudes.

When outdoors, as long as the Sun is in the visual field, it is the source of the greatest irradiance on the retina. It has been estimated that, in an emmetropic eye, direct gazing at the midday Sun on a cloudless sky with a constricted pupil of 2 mm in diameter produces an image of the Sun on the retina of 0.16 mm in diameter, and irradiance in that small area is about 11 W/cm2 (Weinstein and Rylander 1978; Allen and Richey 1966). Obviously, that irradiance depends on the latitude, season, time of day, and atmospheric conditions. Other estimates of retinal irradiance in a human eye viewing the mid-day sun vary between 1.6 W/cm2 for 400-500 nm range (Ham et al. 1978) to 18-122 W/cm2 for the full spectral range (including UV and infrared) (Allen and Richey 1966; Friedman and Kuwabara 1968). The calculations of Pflibsen at al. suggest that if the Sun is at the 50° eccentricity to the visual axis, its projection on the retina exposes the retina to irradiance which is only 10% to 20% smaller (depending on age) than that due to gazing at the Sun directly, just the image of the Sun is focused at a different retinal area than the centre of the fovea (Pflibsen, Pomerantzeff, and Ross 1988). I have added a section 4.6.11 How much sunlight reaches the retina?

The epidemiological study, meta-analysis of epidemiological studies, and clinical trial cited by the Reviewer [7-9] (Delcourt et al. 2014; Zhou et al. 2018; Lee et al. 2022), indeed do not support the role of sunlight exposure in AMD progression. Altogether, there are more studies showing a statistically significant positive association of AMD with sunlight exposure than studies finding no association or negative association. None of these studies evaluated exposure to light of the retina. I have added to the manuscript another section 4.6.10 about the studies investigating an association between sunlight exposure and AMD.

Cell culture studies described in section 4.6.1  used irradiance levels that are 10 to 100-fold greater than the average irradiance of the human retina in daylight (with no direct projection of the Sun on the retina) (Sliney 2005; Rozanowska, Rozanowski, and Boulton 2009) but much smaller in comparison with the irradiance in the area of the retina, 0.16 mm in diameter, where the image of the Sun is focused and can provide local irradiance of 1.6 W/cm2 for 400-500 nm range (Ham et al. 1978; Friedman and Kuwabara 1968; Allen and Richey 1966). Therefore, light levels used in experiments on cultured cells described in the manuscript are of physiological relevance. At the end of section 4.6.1 I have added considerations of the physiological relevance of light exposures used in experiments on lipofuscin-laden cultured cells.

The reference cited by the reviewer (Dorey et al. 1989) downplayed the role of light based on work of Pflibsen at al. and Kooijman mentioned above and assumed that there is no increased illumination of the retina at the posterior pole but, as stated above, these authors evaluated the illuminance of the retina exposed to uniform luminance across the entire visual field, which is not applicable to physiological conditions. Downplaying the role of light based on these data was incorrect.

  1. The conclusion “Oxidized docosahexaenoic acid… a likely component of lipofuscin…” (and therefore spectral shifts are indicating oxidative damage to this vulnerable fatty acid) is not consistent with long-standing data on lipofuscin lipid composition. According to Bazan et al [11], lipofuscin is enriched in palmitic acid (16:0), arachidonic acid (20:4) and oleic acid (18:1) in contrast to photoreceptor outer segments, which are enriched in docosahexaenoic acid (22:6). Finally, not all outer segment material ends up in lipofuscin, as outer segment membrane lipids undergo to undergo fatty acid beta-oxidation to fuel the RPE (“metabolic ecosystem”) to spare glucose for the photoreceptors [12]. These new concepts of outer retinal physiology impacting the source material for lipofuscin should be considered. 

I have addressed the different fatty acid composition in POS and RPE in section 4.4 by providing reasons why these differences can occur with emphasis on DHA, including its recycling into the outer segment and use for beta-oxidation, as well as non-enzymatic oxidation.

I offer a few specific comments, referring to pages in the reviewer pdf. Please note that references after 26 are off by one, because of an extra carriage return in reference 26. In general, the writing style could benefit from editing for conciseness and organization. 

Thank you for identifying the source of the error in reference numbering. The additional number appearing in paragraph number was not present in the submitted version of the manuscript so some re-formatting must have taken place before it was sent for review. I have corrected this and read the manuscript thoroughly to see if any sections could be more concise, and added additional headings to improve its organisation.

  1. 2, Among them, there is a group of neurodegenerative genetic disorders called neuronal ceroid lipofuscinosis (NCL), which is characterized by the formation of intracellular deposits in neurons and other cell types. 

NCL is genetically inherited disease. M.L. Katz wrote a wonderful review that distinguished between RPE lipofuscin and inclusions due to storage disorders such as NCL. He emphasized that the lipofuscin in each cell type containing is unique to that cell type [13], and he specifically distinguished between brain and RPE lipofuscin. Text on p. 3 mixes together lipofuscin from different organ systems. 

 I have included two additional sentences in that paragraph and cited the reference there as well as in the beginning of section 4.3. Katz and his co-workers also observed that lipofuscin in the same type of cell can have different properties depending on the material it is formed from: In vitamin A and E deficiency RPE lipofuscin forms but its fluorophores exhibit different solubility in organic solvents than for lipofuscin formed under normal levels of vitamin A.

  1. 4, In cases where care was taken to consider if the spectral correction was needed, it has been shown that the spectral range of fluorescence emission varies depending on the tissue from which the lipofuscin is derived. 

This is a good point and could be used by the author to promulgate standardized analysis of spectra.

 I have added a sentence in the Conclusion section recommending using corrected emission spectra in the future work for comparisons of fluorescence characteristics of lipofuscin from different tissues.

  1. 4, autoxidation of docosahexaenoic acid (DHA) results in the formation of products with an emission maximum at about 610 nm when excited with 488 nm light and broad-band fluorescence with a maximum in blue when excited with UV light (74).

See comments above re DHA.

 Oxidation contributes to lipofuscin formation and, once formed, lipofuscin can stay in the RPE for decades. Therefore, it would be highly unlikely for DHA, which is highly susceptible to peroxidation, to remain in lipofuscin in a non-oxidized state for long. It can be expected that freshly formed lipofuscin contains less oxidized DHA than lipofuscin aged for months, years, and decades. Lipofuscin isolations include a mixture of lipofuscin of different ages. DHA oxidation is addressed in sections 4.4.

Figure 1: Retinal illuminance is homogenous to 50° eccentricity so the beam of light on the fovea needs to be fixed (see above). The macula lutea is 3 mm diameter, the central area is 6 mm diameter [14]. The figure needs a scale bar.

As mentioned above, the homogenous retinal illuminance up to 50° eccentricity can occur only when the luminance is homogenous across the visual field. To emphasize the refractive power of the eye, I have added an image of the Sun. The 3 mm diameter macula lutea refers to the area that is visible post-mortem as a yellow spot due to the presence of lutein and zeaxanthin. This can vary between individuals depending on the dietary intake of these xanthophylls and their bioavailability. In the manuscript I use the more common definition of macula which is 5.5 mm in diameter (Yanoff and Duker 2013).

I have added information to the figure caption that the left part of the figure is not to scale. The right part of the figure is drawn to scale as long as the length of the horizontal axis of the optic disc corresponds to the average length for a woman, which is 1.7 mm.

  1. 6, These structural differences in POS of rods and cones may have important consequences for their phagocytosis and lysosomal degradation: in the case of rods, RPE phagocytoses the oldest discs, whereas, in the case of cones, the phagocytosed material is a mixture of old and recently added lipids and proteins. 

Reference for this information? Does not sound right to me. I did not see the classics of photoreceptor cell biology cited. 

I have added appropriate references (Kevany and Palczewski 2010; Mustafi, Engel, and Palczewski 2009).

Table 1. Studies of the relationship of fundus fluorescence with age – this is potentially an interesting table and could be greatly improved for impact. Please divide ex vivo and in vivo fluorescence detection studies. Please also indicate the detection technology and the retinal area sampled. Assess each study on a standard set of parameters so that it is easier to compare. Important contributions by Kleefeldt et al and Probster et al to lifespan autofluorescence should be included [15,16]. What is the inner and outer ring in the last entry? 

The order in the table is based on the excitation wavelength. The fluorescence characteristics in vivo and ex vivo are similar as shown by Delori et al., (1995) (and is stated in the 2nd paragraph of sections 6.2). I have shaded in grey the studies done on tissue ex vivo and added information on the retinal areas sampled. The main point of the table is to provide a brief summary of findings from different studies of the relationship between the age and fluorescence which varies depending on the excitation and emission wavelengths. More details are included in the text and in the cited papers. Thank you for pointing me to the papers I missed. They are now included in the table and text. The dimensions of the rings in the ETDRS grid are now added in the table.

  1. 10 ff – most of the described cell culture studies in this section used non-phenotypic RPE, by today’s standards. When RPE in culture is polarized and joined by continuous tight junctions in a geometrically precise array, it is resistant to stress [17,18]. Few of the cited studies used cells meeting these standards. The lighting regimens were harsh and non-physiologic. Blue (and violet) wavelengths are the part of the action spectrum for opsins (rhodopsin and non-canonical opsins). Most used fluorescence microscopy to assess inclusions, few if any used transmission electron microscopy in addition to confirm the physical form of the autofluorescent material. It is not helpful to describe all these studies without appropriate evaluations, disclaimers, and comparisons to RPE in vivo/ in situ/ primary/ explants. Much of this section could be deleted in my opinion.

In most studies included in the manuscript, the cells were indeed studied in non-polarized cultures without staining for tight junctions. There is a possibility they are less or more resistant to stress than RPE cells in vivo. However, the cited studies used appropriate controls and they do contribute to understanding lipofuscin formation and its effects.

I addressed the light exposures above and hopefully convinced the Reviewer that they were physiologically relevant. Under bright light, most visual pigments are in the bleached state.

The study by Thurman et al. (Thurman et al. 2009) cited by the Reviewer [17] investigated the effect of VEGF on the permeability of tight junctions so it was essential to have polarised cultures grown in inserts so the transepithelial resistance could be measured. For other experiments described in that paper, ARPE-19 cells were grown on plates or glass coverslips, and no checks were done for the continuity of tight junctions or for the expression of RPE-specific genes. Also, no comparisons were done to see if the cells cultured on inserts are more resistant to hydrogen peroxide than cells cultured on plates/coverslips. The other paper cited by the Reviewer [17] by Zhang et al. is one of the studies included in the manuscript (Zhang et al. 2019). It showed that differentiated human foetal RPE cultures accumulate less lipofuscin-like pigments than polarized cultures of ARPE-19 cells when fed non-oxidized POS. It is unclear which of these cultures is more similar to the adult human RPE in vivo.

I agree that cells in culture can have different susceptibility to stress than under physiological conditions.

Fluorescence microscopy was indeed the typical method for assessing lipofuscin and was not always accompanied by TEM or another method of assessing lipofuscin granules. This information is always added where relevant.

  1. 11, It remains to be shown whether these A2E oxidation products can stimulate these deleterious effects in vivo or whether they are safely trapped in the lipofuscin granule. It has been reported that oxidized A2E can react with other oxidized A2E molecules or A2E itself forming high-molecular weight products, which are more hydrophobic than A2E, and therefore more likely to remain in the granule (204,205). 

This is a good point that suggests the author’s ambivalence about these model system studies. 

  1. 12, While it is clear that lipofuscin exhibits photosensitizing properties and can affect cell function and viability in vitro upon exposure to light, the evidence for the contribution of lipofuscin to light-induced retinal injury in vivo is rather limited.

Does the author consider these studies valid? They are discussed at length. Not clear to me.

I clarified the statement by adding three sentences in that paragraph:  

Experiments in vitro add valuable information on the mechanisms that could be involved in the effects of lipofuscin. … The main difficulty with the interpretation of the results of in vivo studies is in distinguishing whether light-induced injury to the retina is caused by lipofuscin or by retinaldehydes. Nevertheless, the studies in vivo with the potential involvement of lipofuscin are discussed at length to show the circumstantial evidence that is available.

  1. 30, Marmorstein et al. compared fluorescence properties …

Use of the terms autofluorescence vs fluorescence is inconsistent.

The usage of the word “autofluorescence” was explained at the beginning of section 6.2. I have double-checked that the word “fluorescence” is used unless it is a part of the name of the imaging technique called fundus autofluorescence.

  1. 30, However, in contrast to normal eyes, where the intensity maxima were similar for Bruch’s membrane and RPE, in AMD eyes the intensity maximum was about 1.8-fold greater for Bruch’s membrane than for RPE. The emission spectrum for sub-RPE deposits was similar in shape but the emission maximum was about twice smaller than for Bruch’s membrane. The emission maximum of the RPE fluorescence induced by excitation with 364 nm was at 540 nm, which is 15 nm shifted towards shorter wavelengths in comparison with the normal retina.

This text is difficult to read for a sense of quantitative relationships. The directions of the comparisons vary from one sentence to another. Please rephrase, e.g., group 1 > group 2 > group 3. Or something like that. 

 I have amended the 2nd sentence of the above text: The emission spectrum for Bruch’s membrane was similar in shape but the emission maximum was about twice greater than for sub-RPE deposits.

 17, Because bisretinoids are the major emitter of lipofuscin fluorescence (105,106,145), it is plausible that some of the observed changes in fluorescence during lipofuscin photo-degradation can be ascribed to light-induced oxidation of bisretinoids, which are prone to oxidative degradation. 

Please comment how bisretinoids abundant in far peripheral retina, where the light levels are lower than in central retina, could be subject to photo-oxidation (see references above). I don’t see how this is possible. They can be photo-oxidized in vitro. 

I added references to the same papers as those discussed in the preceding paragraph to make it clear that the sentence refers to the photodegradation of lipofuscin in vitro.

Light reaches all parts of the retina, just the central retina is exposed to higher intensities of light than the peripheral retina. Because of that, it can be expected that both - the formation of lipofuscin and photodegradation of bisretinoids occur faster in the central retina than in the periphery. This is consistent with the experimental data of Feeney-Burns on age dependence of lipofuscin content and Bhosale et al. (2009) on age-dependent A2E content, which are now described in sections 4.5.1 and 7.2, respectively.

Figure 3. Top panel: fluorescence of lipofuscin-laden ARPE-19 cell monolayer kept in the dark )left panel) or exposed daily for 14 days to 45 minutes of 9.76 mW/cm2 visible light providing a total dose of 369 J/cm2 (right panel). 

This figure show non-phenotypic cells. Were assays performed to verify RPE-specific phenotype? (e.g., gene expression, transepithelial resistance, calcium signaling, other [19]). Are junctional complexes intact? Was the fluorescent material investigated with high-resolution microscopy techniques? 

The figure shows fully confluent ARPE-19 cells maintained for 2 weeks after reaching full confluency (as judged by phase contrast microscopy) before the start of feeding with LF at each media change done 3 times per week, 13 times in total followed by daily exposures to light/dark for 14 days (in total 9 weeks after seeding). No assays were performed to verify RPE-specific phenotype. The live cells were imaged with low-resolution microscopy, followed by solubilization in Triton X-100 and measurement of fluorescence spectra in a spectrofluorometer. These details are included in the methods section of the paper cited in the figure caption. Imaging of live cells with high-resolution microscopy, ideally in polarized culture, would enable monitoring in real time the formation and removal of vesicles containing lipofuscin. I have added in the Conclusions section the need to investigate the exocytosis of lipofuscin in polarised RPE cell cultures and in vivo.

  1. 20, Due to the high concentration of polyunsaturated fatty acids, especially in photoreceptor outer segments, lipofuscin, drusen, and Bruch’s membrane it can be expected that products of lipid oxidation and their adducts with proteins can contribute to retinal fluorescence (68-70,74,140,248). 

Author should be aware that studies of Bruch’s membrane lipids, verified as specific to Bruch’s by way of lipid histochemistry, have shown a high concentration of linoleate and low concentration of DHA [20,21]. So whatever autofluorescence appears in Bruch’s in studies like Marmorstein et al cannot be assumed to derive from DHA. 

Polyunsaturated fatty acids include all fatty acids with two or more unsaturated double bonds. As described in section 3, the contribution of products of lipid oxidation to lipofuscin fluorescence was considered already decades ago but the fluorescence exhibited maxima in blue or green. None of these studies looked at fluorescence of oxidized DHA.

DHA does not absorb UVA or visible light so it cannot be responsible for the fluorescence of Bruch’s membrane excited by such light. As explained above, in a mixture of DHA and LA, LA can be spared from oxidation by the greater susceptibility to oxidation of DHA (Else and Kraffe 2015). Oxidation products of DHA do fluoresce upon excitation with UV and blue light (Rozanowska and Rozanowski 2022). CEP has been identified in photoreceptor outer segments, lipofuscin, RPE-drusen/Bruch’s membrane/choroid complex (Crabb et al. 2002; Gu et al. 2003) so it is likely other products of DHA oxidation are present there as well. I added information on DHA oxidation products in the Conclusions.

  1. 21, Sub-retinal and sub-RPE deposits, which accumulate with age and some of them are a hallmark of AMD, such as reticular pseudodrusen, basal laminar deposits, basal linear deposits and drusen, can fluorescence (275-277). 

can emit fluorescence signal … 

Changed to “can emit fluorescence.”

  1. 23, Moreover, this filtering effect is likely to change with ageing because i) there is an overall loss of melanin; ii) there is an accumulation of lipofuscin and complex granules containing both melanin and lipofuscin; and iii) melanosomes redistribute from their initial location in the apical portion of the RPE and lipofuscin redistributes from its initial location at the basal site to become distributed more evenly throughout the cell. 

The evidence for loss of melanin with age comes from studies with an unknown degree of melanosome loss, to my understanding. Most RPE melanosomes are in the delicate apical processes [22,23], which may have been lost in the processing of tissues for these studies. This limitation should be addressed. 

 I have added appropriate references including a review where age-related changes in RPE melanin content were discussed. Two studies quantified melanin on sections from fixed eyes and evaluated the intracellular distribution of melanin granules (Weiter et al. 1986; Feeney-Burns, Hilderbrand, and Eldridge 1984). The other two quantified melanin in isolated RPE so indeed some melanosomes could be lost (Sarna et al. 2003; Schmidt and Peisch 1986) but this would underestimate the melanin content mainly in RPE from young eyes and, consequently, underestimate the age-elated loss of melanin.

  1. 29, and is most pronounced in people of White ethnicity (284,335,337,338) (Table 1). 

The sample of other races is small, unfortunately. 

  1. 33, damaged photoreceptors and absent RPE: there is no characteristic hypo-reflective layer corresponding to the outer nuclear layer and no characteristic hyper-reflective layers corresponding to the outer limiting membrane and the ellipsoid area of the inner segment. 

OCT has bands not layers – please include (or refer readers to) appropriate references. 

Thank you for pointing this out. I have replaced the “layer(s)” with “band(s)” in both cases where it was applicable.

  1. 33, It has been concluded that the residual debris and drusen-like deposits are the origins of the green-emitting fluorophores …

What is a “drusen-like deposit” - do you mean basal laminar deposit? Or drusen. These are two separate layers. Separation of BLamD from drusen by electron microscopy (by the Sarks) was an important milestone in AMD pathology. Author may find these recent papers on AMD deposits (and references within) of use. [24,25] All these layers are visible clinically so please mention carefully. 

The green-fluorescent areas correspond to dome-shaped hyperreflective areas in OCT images and may not be drusen but rather debris in the remaining retina so I replaced “the residual debris and drusen-like deposits” with “the residual debris overlying Bruch’s membrane.”

  1. 36, Feldman et al. reported fluorescence spectra of suspensions of RPE cells from two normal eyes and two AMD eyes [371]. 

Were these macular RPE cells or whole eye cup extractions?

These were macular RPE cells so the word “macular” is added where applicable.

  1. 38, Secondly, in comparison with age-matched normal human retinas, AMD-affected retinas exhibit a greater level of protein modifications by oxidized lipids, such as CEP which has been detected in photoreceptor outer segments, RPE, Bruch’s membrane and drusen (395,396).

Please see comment above about low levels of DHA and high levels of linoleate in Bruch’s membrane. CEP adducts were diffusely distributed and not confined to Bruch’s or drusen, as were other lipids ostensibly of lipoprotein origin. 

As explained above, in the presence of DHA, linoleate can be protected from lipid peroxidation. I have added “diffusely distributed” to the cited sentence and added another sentence: “Moreover, it has been shown that Western blots of proteins extracted from RPE-Bruch’s membrane-choroid complexes isolated from 11 AMD and 11 normal eyes show positive staining of numerous proteins for anti-CEP antibody, and this staining is more pronounced in AMD tissues than in normal tissues (Crabb et al. 2002).”

  1. Schmitz-Valckenberg S, Pfau M, Fleckenstein M, Staurenghi G, Sparrow JR, Bindewald-Wittich A, Spaide RF, Wolf S, Sadda S, Holz FG. Fundus autofluorescence imaging. Prog Retin Eye Res. 2020;81:100893. PMID 32758681 
  2. Bhosale P, Serban B, Bernstein PS. Retinal carotenoids can attenuate formation of A2E in the retinal pigment epithelium. Archives of Biochemistry and Biophysics. 2009;483(2):175-181. PMID 18926795 
  3. Kotnala A, Senthilkumari S, Gong W, Stewart TG, Curcio CA, Halder N, Kumar A, Velpandian T. Retinal pigment epithelium in human donor eyes contains higher levels of bisretinoids including A2E in periphery than macula. Invest Ophthalmol Vis Sci. 2022;63(6):6. PMID 35671050 
  4. Ablonczy Z, Higbee D, Grey AC, Koutalos Y, Schey KL, Crouch RK. Similar molecules spatially correlate with lipofuscin and N-retinylidene-N-retinylethanolamine in the mouse but not in the human retinal pigment epithelium. Arch Biochem Biophys. 2013;539(2):196-202. PMID 23969078 
  5. Pflibsen KP, Pomerantzeff O, Ross RN. Retinal illuminance using a wide-angle model of the eye. Journal of the Optical Society of America. A, Optics and image science. 1988;5(1):146-150. PMID 3351651 
  6. Kooijman AC. Light distribution on the retina of a wide-angle theoretical eye. Journal of the Optical Society of America. 1983;73(11):1544-1550. PMID 6644400 
  7. Delcourt C, Cougnard-Gregoire A, Boniol M, Carriere I, Dore JF, Delyfer MN, Rougier MB, Le Goff M, Dartigues JF, Barberger-Gateau P, Korobelnik JF. Lifetime exposure to ambient ultraviolet radiation and the risk for cataract extraction and age-related macular degeneration: the Alienor Study. Invest Ophthalmol Vis Sci. 2014;55(11):7619-7627. PMID 25335979 
  8. Zhou H, Zhang H, Yu A, Xie J. Association between sunlight exposure and risk of age-related macular degeneration: a meta-analysis. BMC Ophthalmol. 2018;18(1):331. PMID 30572865 
  9. Lee JS, Li PR, Hou CH, Lin KK, Kuo CF, See LC. Effect of blue light-filtering intraocular lenses on age-related macular degeneration: A nationwide cohort study with 10-year follow-up. Am J Ophthalmol. 2021;234:138-146. PMID 34411525 
  10. Mainster MA, Findl O, Dick HB, Desmettre T, Ledesma-Gil G, Curcio CA, Turner PL. The blue-light-hazard vs. the blue-light-hype. Am J Ophthalmol. Aug 2022;240:51-57. PMID 35227699 
  11. Bazan HEP, Bazan NG, Feeney-Burns L, Berman ER. Lipids in human lipofuscin-enriched subcellular fractions of two age populations. Comparison with rod outer segments and neural retina. Invest. Ophthalmol. Vis. Sci. 1990;31:1433-1443. PMID 2387677 
  12. Reyes-Reveles J, Dhingra A, Alexander D, Bragin A, Philp NJ, Boesze-Battaglia K. Phagocytosis dependent ketogenesis in retinal pigment epithelium. J Biol Chem. Mar 16 2017;292(19):8038-8047. PMID 28302729 
  13. Katz ML, Robison WG, Jr. What is lipofuscin? Defining characteristics and differentiation from other autofluorescent lysosomal storage bodies. Arch Gerontol Geriatr. 2002;34(3):169-184. PMID 14764321 
  14. Polyak SL. The Retina. Chicago: University of Chicago; 1941.
  15. Pröbster C, Tarau I-S, Berlin A, Kleefeldt N, Hillenkamp J, Nentwich MM, Sloan KR, Ach T. Quantitative fundus autofluorescence in the developing and maturing healthy eye. Translational Vision Science & Technology. 2021;10(2):15. PMID 34003900 
  16. Kleefeldt N, Bermond K, Tarau I-S, Hillenkamp J, Berlin A, Sloan KR, Ach T. Quantitative fundus autofluorescence: advanced analysis tools. Translational Vision Science & Technology. 2020;9(8):2. PMID 32855849 
  17. Thurman JM, Renner B, Kunchithapautham K, Ferreira VP, Pangburn MK, Ablonczy Z, Tomlinson S, Holers VM, Rohrer B. Oxidative stress renders retinal pigment epithelial cells susceptible to complement-mediated injury. The Journal of biological chemistry. 2009;284(25):16939-16947. PMID 19386604 
  18. Zhang Q, Presswalla F, Calton M, Charniga C, Stern J, Temple S, Vollrath D, Zacks DN, Ali RR, Thompson DA, Miller JML. Highly differentiated human fetal RPE cultures are resistant to the accumulation and toxicity of lipofuscin-like material. Invest Ophthalmol Vis Sci. 2019;60(10):3468-3479. PMID 31408109 
  19. Miyagishima KJ, Wan Q, Corneo B, Sharma R, Lotfi MR, Boles NC, Hua F, Maminishkis A, Zhang C, Blenkinsop T, Khristov V, Jha BS, Memon OS, D'Souza S, Temple S, Miller SS, Bharti K. In pursuit of authenticity: induced pluripotent stem cell-derived retinal pigment epithelium for clinical applications. Stem Cells Transl Med. 2016;5(11):1562-1574. PMID 27400791 
  20. Wang L, Li C-M, Rudolf M, Belyaeva OV, Chung BH, Messinger JD, Kedishvili NY, Curcio CA. Lipoprotein particles of intra-ocular origin in human Bruch membrane: an unusual lipid profile. Invest Ophthalmol Vis Sci. 2009;50:870-877. PMID 18806290 
  21. Bretillon L, Thuret G, Gregoire S, Acar N, Joffre C, Bron AM, Gain P, Creuzot-Garcher CP. Lipid and fatty acid profile of the retina, retinal pigment epithelium/choroid, and the lacrimal gland, and associations with adipose tissue fatty acids in human subjects. Exp Eye Res. 2008;87(6):521-528. PMID 18801361 
  22. Anderson DH, Fisher SK, Erickson PA, Tabor GA. Rod and cone disc shedding in the rhesus monkey retina: a quantitative study. Exp. Eye Res. May 1980;30(5):559-574. PMID 7409012 
  23. Pollreisz A, Neschi M, Sloan KR, Pircher M, Mittermueller TJ, Dacey DM, Schmidt-Erfurth U, Curcio CA. An atlas of human retinal pigment epithelium organelles significant for clinical imaging. Invest Ophthalmol Vis Sci. 2020;61(8):13. PMID 32648890 
  24. Sura AA, Chen L, Messinger JD, Swain TA, McGwin Jr G, Freund KB, Curcio CA. Measuring the contributions of basal laminar deposit and Bruch’s membrane in age-related macular degeneration. Invest Ophthalmol Vis Sci. 2020;61(13):19. PMID 33186466 
  25. Chen L, Messinger JD, Kar D, Duncan JL, Curcio CA. Biometrics, impact, and significance of basal linear deposit and subretinal drusenoid deposit in age-related macular degeneration. Invest. Ophthalmol. Vis. Sci. 2021;62(1):33. PMID 33512402 

References cited in response to Reviewer 1:

Ablonczy, Z., D. Higbee, D. M. Anderson, M. Dahrouj, A. C. Grey, D. Gutierrez, Y. Koutalos, K. L. Schey, A. Hanneken, and R. K. Crouch. 2013. "Lack of correlation between the spatial distribution of A2E and lipofuscin fluorescence in the human retinal pigment epithelium."  Invest Ophthalmol Vis Sci 54 (8):5535-42. doi: 10.1167/iovs.13-12250.

Ablonczy, Z., D. Higbee, A. C. Grey, Y. Koutalos, K. L. Schey, and R. K. Crouch. 2013. "Similar molecules spatially correlate with lipofuscin and N-retinylidene-N-retinylethanolamine in the mouse but not in the human retinal pigment epithelium."  Arch Biochem Biophys 539 (2):196-202. doi: 10.1016/j.abb.2013.08.005.

Allen, R. G., Jr., and E. O. Richey. 1966. "Eclipse burns in humans and laboratory threshold measurements in rabbits. SAM-TR-66-45."  Tech Rep SAM-TR:1-5.

Avalle, L. B., Z. Wang, J. P. Dillon, and E. R. Gaillard. 2004. "Observation of A2E oxidation products in human retinal lipofuscin."  Exp Eye Res 78 (4):895-8. doi: 10.1016/j.exer.2003.10.023.

Ben-Shabat, S., Y. Itagaki, S. Jockusch, J. R. Sparrow, N. J. Turro, and K. Nakanishi. 2002. "Formation of a nonaoxirane from A2E, a lipofuscin fluorophore related to macular degeneration, and evidence of singlet oxygen involvement."  Angew Chem Int Ed Engl 41 (5):814-7. doi: 10.1002/1521-3773(20020301)41:5<814::aid-anie814>3.0.co;2-2.

Bhosale, P., B. Serban, and P. S. Bernstein. 2009. "Retinal carotenoids can attenuate formation of A2E in the retinal pigment epithelium."  Arch Biochem Biophys 483 (2):175-81. doi: 10.1016/j.abb.2008.09.012.

Crabb, J. W., M. Miyagi, X. Gu, K. Shadrach, K. A. West, H. Sakaguchi, M. Kamei, A. Hasan, L. Yan, M. E. Rayborn, R. G. Salomon, and J. G. Hollyfield. 2002. "Drusen proteome analysis: an approach to the etiology of age-related macular degeneration."  Proc Natl Acad Sci U S A 99 (23):14682-7. doi: 10.1073/pnas.222551899.

Delcourt, C., A. Cougnard-Gregoire, M. Boniol, I. Carriere, J. F. Dore, M. N. Delyfer, M. B. Rougier, M. Le Goff, J. F. Dartigues, P. Barberger-Gateau, and J. F. Korobelnik. 2014. "Lifetime exposure to ambient ultraviolet radiation and the risk for cataract extraction and age-related macular degeneration: the Alienor Study."  Invest Ophthalmol Vis Sci 55 (11):7619-27. doi: 10.1167/iovs.14-14471.

Dillon, J., Z. Wang, L. B. Avalle, and E. R. Gaillard. 2004. "The photochemical oxidation of A2E results in the formation of a 5,8,5',8'-bis-furanoid oxide."  Exp Eye Res 79 (4):537-42. doi: 10.1016/j.exer.2004.06.024.

Dorey, C. K., G. Wu, D. Ebenstein, A. Garsd, and J. J. Weiter. 1989. "Cell loss in the aging retina. Relationship to lipofuscin accumulation and macular degeneration."  Invest Ophthalmol Vis Sci 30 (8):1691-9.

Else, P. L., and E. Kraffe. 2015. "Docosahexaenoic and arachidonic acid peroxidation: It's a within molecule cascade."  Biochim Biophys Acta 1848 (2):417-21. doi: 10.1016/j.bbamem.2014.10.039.

Feeney-Burns, L., E. S. Hilderbrand, and S. Eldridge. 1984. "Aging human RPE: morphometric analysis of macular, equatorial, and peripheral cells."  Invest Ophthalmol Vis Sci 25 (2):195-200.

Friedman, E., and T. Kuwabara. 1968. "The retinal pigment epithelium. IV. The damaging effects of radiant energy."  Arch Ophthalmol 80 (2):265-79. doi: 10.1001/archopht.1968.00980050267022.

Gaillard, E. R., L. B. Avalle, L. M. Keller, Z. Wang, K. J. Reszka, and J. P. Dillon. 2004. "A mechanistic study of the photooxidation of A2E, a component of human retinal lipofuscin."  Exp Eye Res 79 (3):313-9. doi: 10.1016/j.exer.2004.05.005.

Gu, X., S. G. Meer, M. Miyagi, M. E. Rayborn, J. G. Hollyfield, J. W. Crabb, and R. G. Salomon. 2003. "Carboxyethylpyrrole protein adducts and autoantibodies, biomarkers for age-related macular degeneration."  J Biol Chem 278 (43):42027-35. doi: 10.1074/jbc.M305460200.

Ham, W. T., Jr., J. J. Ruffolo, Jr., H. A. Mueller, A. M. Clarke, and M. E. Moon. 1978. "Histologic analysis of photochemical lesions produced in rhesus retina by short-wave-length light."  Invest Ophthalmol Vis Sci 17 (10):1029-35.

Kevany, B. M., and K. Palczewski. 2010. "Phagocytosis of retinal rod and cone photoreceptors."  Physiology (Bethesda) 25 (1):8-15. doi: 10.1152/physiol.00038.2009.

Kotnala, A., S. Senthilkumari, G. Wu, T. G. Stewart, C. A. Curcio, N. Halder, S. B. Singh, A. Kumar, and T. Velpandian. 2022. "Retinal Pigment Epithelium in Human Donor Eyes Contains Higher Levels of Bisretinoids Including A2E in Periphery than Macula."  Invest Ophthalmol Vis Sci 63 (6):6. doi: 10.1167/iovs.63.6.6.

Lee, J. S., P. R. Li, C. H. Hou, K. K. Lin, C. F. Kuo, and L. C. See. 2022. "Effect of Blue Light-Filtering Intraocular Lenses on Age-Related Macular Degeneration: A Nationwide Cohort Study With 10-Year Follow-up."  Am J Ophthalmol 234:138-146. doi: 10.1016/j.ajo.2021.08.002.

Murdaugh, L. S., L. B. Avalle, S. Mandal, A. E. Dill, J. Dillon, J. D. Simon, and E. R. Gaillard. 2010. "Compositional studies of human RPE lipofuscin."  J Mass Spectrom 45 (10):1139-47. doi: 10.1002/jms.1795.

Murdaugh, L. S., S. Mandal, A. E. Dill, J. Dillon, J. D. Simon, and E. R. Gaillard. 2011. "Compositional studies of human RPE lipofuscin: mechanisms of molecular modifications."  J Mass Spectrom 46 (1):90-5. doi: 10.1002/jms.1865.

Mustafi, D., A. H. Engel, and K. Palczewski. 2009. "Structure of cone photoreceptors."  Prog Retin Eye Res 28 (4):289-302. doi: 10.1016/j.preteyeres.2009.05.003.

Pflibsen, K. P., O. Pomerantzeff, and R. N. Ross. 1988. "Retinal illuminance using a wide-angle model of the eye."  J Opt Soc Am A 5 (1):146-50. doi: 10.1364/josaa.5.000146.

Rozanowska, M. B., and B. Rozanowski. 2022. "Photodegradation of Lipofuscin in Suspension and in ARPE-19 Cells and the Similarity of Fluorescence of the Photodegradation Product with Oxidized Docosahexaenoate."  Int J Mol Sci 23 (2). doi: 10.3390/ijms23020922.

Rozanowska, M., B. Rozanowski, and M.  Boulton. 2009. "Photobiology of the retina: Light damage to the retina. (accessed on 23 March 2023)." In Photobiological Sciences Online: http://www.photobiology.info, edited by K.C. Smith. Herndon, VA, USA: American Society for Photobiology.

Sarna, T., J. M. Burke, W. Korytowski, M. Rozanowska, C. M. Skumatz, A. Zareba, and M. Zareba. 2003. "Loss of melanin from human RPE with aging: possible role of melanin photooxidation."  Exp Eye Res 76 (1):89-98. doi: 10.1016/s0014-4835(02)00247-6.

Schmidt, S. Y., and R. D. Peisch. 1986. "Melanin concentration in normal human retinal pigment epithelium. Regional variation and age-related reduction."  Invest Ophthalmol Vis Sci 27 (7):1063-7.

Sliney, D. H. 2005. "Exposure geometry and spectral environment determine photobiological effects on the human eye."  Photochem Photobiol 81 (3):483-9. doi: 10.1562/2005-02-14-RA-439.

Thurman, J. M., B. Renner, K. Kunchithapautham, V. P. Ferreira, M. K. Pangburn, Z. Ablonczy, S. Tomlinson, V. M. Holers, and B. Rohrer. 2009. "Oxidative stress renders retinal pigment epithelial cells susceptible to complement-mediated injury."  J Biol Chem 284 (25):16939-16947. doi: 10.1074/jbc.M808166200.

Wang, Z., L. M. Keller, J. Dillon, and E. R. Gaillard. 2006. "Oxidation of A2E results in the formation of highly reactive aldehydes and ketones."  Photochem Photobiol 82 (5):1251-7. doi: 10.1562/2006-04-01-RA-864.

Weinstein, G. W., and H. G. Rylander. 1978. "Photocoagulation of the fovea."  Trans Am Ophthalmol Soc 76:278-95.

Weiter, J. J., F. C. Delori, G. L. Wing, and K. A. Fitch. 1986. "Retinal pigment epithelial lipofuscin and melanin and choroidal melanin in human eyes."  Invest Ophthalmol Vis Sci 27 (2):145-52.

Yanoff, M., and J.S. Duker. 2013. Ophthalmology: Expert Consult: Online and Print: Elsevier Health Sciences.

Zhang, Q., F. Presswalla, M. Calton, C. Charniga, J. Stern, S. Temple, D. Vollrath, D. N. Zacks, R. R. Ali, D. A. Thompson, and J. M. L. Miller. 2019. "Highly Differentiated Human Fetal RPE Cultures Are Resistant to the Accumulation and Toxicity of Lipofuscin-Like Material."  Invest Ophthalmol Vis Sci 60 (10):3468-3479. doi: 10.1167/iovs.19-26690.

Zhou, H., H. Zhang, A. Yu, and J. Xie. 2018. "Association between sunlight exposure and risk of age-related macular degeneration: a meta-analysis."  BMC Ophthalmol 18 (1):331. doi: 10.1186/s12886-018-1004-y.

Reviewer 2 Report

Comments and Suggestions for Authors

The review by Rozanowska “Lipofuscin, Its Origin, Properties, and Contribution to Retinal Fluorescence as a Potential Biomarker of Oxidative Damage to the Retina” is an extraordinarily  comprehensive, well written and useful review of lipofuscin in ocular tissues. The author describes, analyzes and insightfully critiques over 400 original research reports on the topic. This will be a widely cited resource both for newcomers to the field and for experts who want an insightful and critical review of what already is known and where the field likely will be headed. 

My only suggestion for writing is to add more subheadings within each section. The review is so comprehensive that having subheadings will make it easier to navigate and search. 

A serious problem that needs to be fixed is that in very many cases citations throughout the text do not refer to the appropriate reference. The authors need to check all the citations in this manuscript (not just the ones listed below – there are many more incorrect citations). 

Specific comments:

1.     discussion at bottom of p. 5 about characteristics of rods - I think the author is referring to human photoreceptors but this should be emphasized. 

2.     Please thoroughly check the reference citations. For example - this does not seem to be the correct citation: "increased accumulation of RPE lipofuscin in response to rearing in light/dark cycle as opposed to rearing in dark or in response to short-term exposures to light causing damage to photoreceptors (reviewed in [88]). " Also, the reference to 113 in the next sentence seems incorrect.

3.     end of section 4.2 - is this the correct citation? "regulatory protein, complement receptor 1-like protein y (CRRY) in the RPE [102]. "

4.     section 4.5 "contraditing these findings (reviewed by Curcio[159]). Curcio is not an author on that reference.

5.     another incorrect reference..."or lutein and zeaxanthin inhibits lipofuscin accumulation (reviewed in [171]). "

Author Response

First of all, I thank the Reviewer for their constructive criticism which resulted in an improved manuscript

The review by Rozanowska “Lipofuscin, Its Origin, Properties, and Contribution to Retinal Fluorescence as a Potential Biomarker of Oxidative Damage to the Retina” is an extraordinarily  comprehensive, well written and useful review of lipofuscin in ocular tissues. The author describes, analyzes and insightfully critiques over 400 original research reports on the topic. This will be a widely cited resource both for newcomers to the field and for experts who want an insightful and critical review of what already is known and where the field likely will be headed. 

Thank you to the Reviewer for their positive comments on the manuscript.

My only suggestion for writing is to add more subheadings within each section. The review is so comprehensive that having subheadings will make it easier to navigate and search. 

As suggested, I have added more subheadings.

A serious problem that needs to be fixed is that in very many cases citations throughout the text do not refer to the appropriate reference. The authors need to check all the citations in this manuscript (not just the ones listed below – there are many more incorrect citations). 

All references in the reference list after reference 26 were incorrect due to an apparent corruption after uploading the manuscript where reference 26 was split into 2 paragraphs and both parts were given numbers. All references have been checked.

Specific comments:

  1. 1.discussion at bottom of p. 5 about characteristics of rods - I think the author is referring to human photoreceptors but this should be emphasized. 

Amended as suggested 

  1. Please thoroughly check the reference citations. For example - this does not seem to be the correct citation: "increased accumulation of RPE lipofuscin in response to rearing in light/dark cycle as opposed to rearing in dark or in response to short-term exposures to light causing damage to photoreceptors (reviewed in [88]). " Also, the reference to 113 in the next sentence seems incorrect.

Checked and corrected.

  1. end of section 4.2 - is this the correct citation? "regulatory protein, complement receptor 1-like protein y (CRRY) in the RPE [102]. "

 corrected

  1. section 4.5 "contraditing these findings (reviewed by Curcio[159]). Curcio is not an author on that reference.

 corrected

  1. another incorrect reference..."or lutein and zeaxanthin inhibits lipofuscin accumulation (reviewed in [171]). "

 corrected

Round 2

Reviewer 1 Report

Comments and Suggestions for Authors

The author is partly responsive but is recalcitrant on the intertwined topics of A2E distribution in the retina and light distribution on the retina. Page numbers refer to the cover letter. 

p. 2, I am not aware of any evidence that A2E in the retinal periphery relates to a macular disease so I cannot fulfil the Reviewer’s request to explain how it does so.

This response is unhelpful – multiple papers since 2009 (human, monkey, HPLC, MS) have shown that A2E is much higher in peripheral retina than central retina, where AMD occurs. That’s a 10-13 mm distance between central area and equatorial retina, using the Polyak (neuroanatomical) definitions. If the author cannot accept those findings (and tell the rest of us why we shouldn’t either) then the overall premise of the paper is questionable.

p. 2, Thank you for pointing me to reference 2, the paper by Bhosale et al. (2009) (Bhosale, Serban, and Bernstein 2009).

The authors found another Bhosale 2009 reference (with monkey eyes), and not the one I cited (51 human eyes). And I provided 3 others (of a total of ~8) all showing the same thing, including one study with >200 human eyes [1], that A2E is abundant in peripheral retina.  The world of A2E changed in the last 15 years. 

Because of increased exposure to light (Sliney 2005),

This paper is an inaccurate depiction of light on the retina. Others preceding were much stronger, see below.

p. 3, citation to Sliney 2005

Kooijman 1986 (Figure 3), Pflibsen 1988 (Figure 3), and the classic optic text Atchison and Smith 2000 (Figure 16.2) references show physical and ray-tracing eye model evidence that retinal illuminance is reasonably uniform out to 30-40°. (Atchison, D. A., Smith, G., 2000. Optics of the human eye (Oxford, Butterworth-Heinemann). The author seems to be assuming that uniform retinal illuminance across the retina is homogeneous - none of these authors say that. Conversely, the idea that looking directly at a bright light (like sun-gazing) is more alike human visual experience than light spread across the retina, is specious, in this reviewer’s opinion. Optics and retinal topography evolved together to maximize survival in specific habitats. Sun-gazing and phone-gazing are not part of that.

p. 4, once formed lipofuscin can stay in the RPE for decades.

This widely held view assumes that lipofuscin never turns over and in fact this has never been tested. A recent study suggests a lot of dynamism [2].

p. 4, In the manuscript I use the more common definition of macula which is 5.5 mm in diameter (Yanoff and Duker 2013).

More common but incorrect. This is the central area of Polyak (continuous ganglion cell layer), who considers the macula lutea to be ~ 3 mm in diameter.

p. 5, For other experiments described in that paper, ARPE-19 cells were grown on plates or glass coverslips, and no checks were done for the continuity of tight junctions or for the expression of RPE-specific genes.

Expectations were low and not references to in vivo RPE. RPE phenotype became more interesting when it was decided that cultured iPS cells could be made ready for transplant. [3]

p. 8, Two studies quantified melanin on sections from fixed eyes and evaluated the intracellular distribution of melanin granules (Weiter et al. 1986; Feeney-Burns, Hilderbrand, and

Eldridge 1984).

No photo-documentation was provided by either of those authors to verify the state of the apical processes. Feeney-Burns et al said the retina was long post-mortem and detached. 

1.         Kotnala, A., et al., Retinal pigment epithelium in human donor eyes contains higher levels of bisretinoids including A2E in periphery than macula. Invest Ophthalmol Vis Sci, 2022. 63(6): p. 6.

2.         Chen, L., et al., Histology and clinical imaging lifecycle of black pigment in fibrosis secondary to neovascular age-related macular degeneration Exp Eye Res, 2022. 214: p. 108882.

3.         Miyagishima, K.J., et al., In pursuit of authenticity: induced pluripotent stem cell-derived retinal pigment epithelium for clinical applications. Stem Cells Transl Med, 2016. 5(11): p. 1562-1574.

Comments on the Quality of English Language

Still a difficult read. 

Author Response

Thank you to the Reviewer for reading my response to their comments.

The author is partly responsive but is recalcitrant on the intertwined topics of A2E distribution in the retina and light distribution on the retina. Page numbers refer to the cover letter. 

  1. 2, I am not aware of any evidence that A2E in the retinal periphery relates to a macular disease so I cannot fulfil the Reviewer’s request to explain how it does so.

This response is unhelpful – multiple papers since 2009 (human, monkey, HPLC, MS) have shown that A2E is much higher in peripheral retina than central retina, where AMD occurs. That’s a 10-13 mm distance between central area and equatorial retina, using the Polyak (neuroanatomical) definitions. If the author cannot accept those findings (and tell the rest of us why we shouldn’t either) then the overall premise of the paper is questionable.

I agree that there is consistent evidence that A2E is present in the adult human eye in higher concentrations in the periphery than in the macula/central retina. I cited the relevant papers supporting it already in the first version of the manuscript (pages 10 and 50-51 in the current version).

I am not aware of any studies investigating how A2E in the periphery relates to a macular disease.

Whether A2E in the macula contributes to the development of macular disease is a different question. The preliminary data from a clinical trial of the deuterated form of vitamin A, which prevents A2E formation and slows the progression of geographic atrophy in Stargardt’s macular dystrophy, are the most convincing piece of evidence for the role of A2E in the progression of macular disease (covered in pages 18-19, 53 and 56).

I can only speculate about the mechanisms by which A2E contributes to the macular disease. From Bhosale et al. (2009) data from 2-year-old monkeys on the accumulation of A2E in the macula and surrounding area (8 mm in diameter), it appears that A2E accumulation happens at a very early age and occurs faster there than in the mid-periphery (covered in pages 50-51 in the current version). A2E was not evaluated in the human retinas at such a young age. Faster accumulation of A2E in the macula than periphery means that A2E can inhibit lysosomal proton pumps more in the macula than periphery thereby facilitating faster accumulation of lipofuscin. The apparent age-related slowing down of A2E accumulation in the macula may be due to its faster degradation in the macula than in the periphery resulting from the increased exposure to light. Greater fluxes of light in the central retina are expected to cause more photoexcitations of lipofuscin photosensitizers and their deleterious consequences in the macula than in the periphery.

  1. 2, Thank you for pointing me to reference 2, the paper by Bhosale et al. (2009) (Bhosale, Serban, and Bernstein 2009).

The authors found another Bhosale 2009 reference (with monkey eyes), and not the one I cited (51 human eyes). And I provided 3 others (of a total of ~8) all showing the same thing, including one study with >200 human eyes [1], that A2E is abundant in peripheral retina.  The world of A2E changed in the last 15 years. 

Bhosale et al. 2009 have reported in the same paper on A2E in human and monkey retinas (as well as other animals). As stated above, papers on A2E distribution in the human retina were cited already in the 1st version of the manuscript.

Because of increased exposure to light (Sliney 2005),

This paper is an inaccurate depiction of light on the retina. Others preceding were much stronger, see below.

  1. 3, citation to Sliney 2005

Kooijman 1986 (Figure 3), Pflibsen 1988 (Figure 3), and the classic optic text Atchison and Smith 2000 (Figure 16.2) references show physical and ray-tracing eye model evidence that retinal illuminance is reasonably uniform out to 30-40°. (Atchison, D. A., Smith, G., 2000. Optics of the human eye (Oxford, Butterworth-Heinemann). The author seems to be assuming that uniform retinal illuminance across the retina is homogeneous - none of these authors say that. Conversely, the idea that looking directly at a bright light (like sun-gazing) is more alike human visual experience than light spread across the retina, is specious, in this reviewer’s opinion. Optics and retinal topography evolved together to maximize survival in specific habitats. Sun-gazing and phone-gazing are not part of that.

I do not assume the retinal illuminance is homogenous – just the opposite. Due to the differences in the luminance across the visual field and the optics of the eye, the retinal illuminance is not homogeneous.

 Kooijman and Pflibsen evaluated retinal illuminance under conditions where the luminance across the visual field was uniform. Kooijman achieved this by placing an eye above a hole in a ping-pong ball whose internal surface was homogeneously illuminated. Pflibsen used computer modelling of retinal illumination based on the assumption of “a uniformly luminous hemisphere, a sufficient size to be effectively infinite in radius, with the eye at its center of curvature.” Their work was useful for the development of instruments testing sensitivity to light of different portions of the retina, which are routinely used in the ophthalmic examination of the eye. As stated already in my previous response, neither Kooijman nor Pflibsen evaluated retinal illuminance under physiological conditions where the luminance across the visual field varies depending on light sources and objects reflecting light present in the visual field.

 Dr Sliney is a renowned expert in the field of light exposure of the retina who served as a member, advisor, or chairman of numerous committees active in the establishment of safety standards for the protection of the retina against non-ionizing radiation, including ANSI and ICNIRP. The cited paper by Dr Sliney talks about illuminance of the retina in the natural environment.

  1. 4, once formed lipofuscin can stay in the RPE for decades.

This widely held view assumes that lipofuscin never turns over and in fact this has never been tested. A recent study suggests a lot of dynamism [2].

I have changed the word “can” to “may” (page 9) as indeed there are no studies following the fate of the same lipofuscin granules in the RPE over decades and the longest follow-up was just for 16 weeks (Mata et al. 2000). The cited paper [2] does not report on lipofuscin dynamism. In the manuscript, the removal of lipofuscin from the RPE has been covered in multiple places (pages 17, 31, 44, 51, 53, 56-57).

  1. 4, In the manuscript I use the more common definition of macula which is 5.5 mm in diameter (Yanoff and Duker 2013).

More common but incorrect. This is the central area of Polyak (continuous ganglion cell layer), who considers the macula lutea to be ~ 3 mm in diameter.

The definition of the macula used in the manuscript is clearly defined in Figure 1. I do follow the more common definition of the macula because most of the studies discussed in the manuscript use it as well as ophthalmology textbooks. It is based on neuroanatomical findings: the edge of the macula (as well as and the outer edge of the perifovea) is where the density of retinal ganglion cells drops to a single layer. This definition is used by most ophthalmology textbooks, including one by the American Academy of Ophthalmology:

https://www.aao.org/education/bcscsnippetdetail.aspx?id=0d7f907c-fb23-44b6-8482-b1d4e1aca277#:~:text=Surrounding%20the%20fovea%20is%20the,(Table%201%2D1).

  1. 5, For other experiments described in that paper, ARPE-19 cells were grown on plates or glass coverslips, and no checks were done for the continuity of tight junctions or for the expression of RPE-specific genes.

Expectations were low and not references to in vivo RPE. RPE phenotype became more interesting when it was decided that cultured iPS cells could be made ready for transplant. [3]

The point of that experiment was to see whether lipofuscin can be photobleached in cultured cells without affecting cell viability. The removal of lipofuscin was a surprising finding. Definitely, it is worth investigating in polarized cultures of RPE to see whether the removal occurs on the apical or basal side. This was added in the 1st revision of the manuscript (page 56 of the current version).

  1. 8, Two studies quantified melanin on sections from fixed eyes and evaluated the intracellular distribution of melanin granules (Weiter et al. 1986; Feeney-Burns, Hilderbrand, and Eldridge 1984).

No photo-documentation was provided by either of those authors to verify the state of the apical processes. Feeney-Burns et al said the retina was long post-mortem and detached. 

Even if melanosomes in the apical processes of the RPE were lost before melanin quantification, that loss would be greater in young eyes than in old eyes, therefore, underestimating the age-related loss of melanin. This means that the statement in the manuscript about the age-related loss of melanin in the RPE is correct.

  1. Kotnala, A., et al., Retinal pigment epithelium in human donor eyes contains higher levels of bisretinoids including A2E in periphery than macula.Invest Ophthalmol Vis Sci, 2022. 63(6): p. 6.
  2. Chen, L., et al., Histology and clinical imaging lifecycle of black pigment in fibrosis secondary to neovascular age-related macular degeneration Exp Eye Res, 2022. 214: p. 108882.
  3. Miyagishima, K.J., et al., In pursuit of authenticity: induced pluripotent stem cell-derived retinal pigment epithelium for clinical applications.Stem Cells Transl Med, 2016. 5(11): p. 1562-1574.

Round 3

Reviewer 1 Report

Comments and Suggestions for Authors

Kooijman 1986, Pflibsen 1988, and classic optic texts show physical and ray-tracing eye model evidence that retinal illuminance is reasonably uniform out to 30-40°. These were summarized and re-affirmed in a PhD thesis in Physics 2013 designed expressly to answer the question of light on the retina for the purpose of understanding what if any photoprotection was conferred by macular xanthophyll pigment. [1] The Sliney reference was written by a regulator.  

1.         Gibert JC. Distribution of light in the human retina under natural viewing conditions: Physics, Florida International University; 2013.

Comments on the Quality of English Language

Readable.

Author response to report 3:

Kooijman 1986, Pflibsen 1988, and classic optic texts show physical and ray-tracing eye model evidence that retinal illuminance is reasonably uniform out to 30-40°. These were summarized and re-affirmed in a PhD thesis in Physics 2013 designed expressly to answer the question of light on the retina for the purpose of understanding what if any photoprotection was conferred by macular xanthophyll pigment. [1] The Sliney reference was written by a regulator.  

  1. Gibert JC. Distribution of light in the human retina under natural viewing conditions: Physics, Florida International University; 2013.

The Reviewer 1 appears to be greatly influenced by the paper by Dorey et al. (1989) and possibly also by a more recent paper by Gambril et al. (2019), who cited incorrectly the work of Kooijman (1986) and Pflibsen et al. (1988) as evidence of uniform illumination “well beyond the 21° macula” of the retina under natural viewing conditions. As explained in my previous response, the experimental setup and the assumptions made for calculations need to be considered when looking at the results presented in Kooijman and Pflibsen’s papers. It is worth having a thorough read of these papers and understanding of what they show before claiming that all journal articles with diagrams depicting the refractive power of the human eye are wrong.

The PhD thesis by Gibert is another example (in addition to the above-mentioned papers as well as Polyak, 1941 and Bhosale et al., 2009) where the Reviewer 1 cites the source to support their views, where in fact the cited source does not. In this case, the PhD thesis contradicts the Reviewer’s view on uniform light distribution on the retina.

Gibert’s PhD thesis clearly illustrates how the illuminance across the retina can vary depending on what we look at. The calculations are based on the results obtained by Kooijman and Pflibsen’s et al. Graphs in Figures 3.1 and 3.2 on page 32 of the thesis show the cumulative retinal illuminance over a period of about 10 minutes in a situation when a participant was fixating on either the brightest or darkest areas, respectively, of the sequence of photographic images presented on a computer screen. As expected, when fixating on the brightest area, the greatest illuminance is at the fovea, whereas, when fixating on the darkest area, there is a global minimum of retinal illuminance at the fovea. The graphs also show various local minima and maxima corresponding to the various luminance levels in the area over which the calculations were made spreading 15° in the superior and inferior directions, and 20° in the nasal and temporal directions. The calculations of retinal illuminance were also done for participants walking freely in the building or around the FIU campus. Some results presented in the thesis were published (Bone et al., 2012; 2013) and I cited this work in the current version of the manuscript on page 56.

Dorey, C.K.; Wu, G.; Ebenstein, D.; Garsd, A.; Weiter, J.J. Cell loss in the aging retina. Relationship to lipofuscin accumulation and macular degeneration. Invest Ophthalmol Vis Sci 1989, 30, 1691-1699.

Gambril, J.A.; Sloan, K.R.; Swain, T.A.; Huisingh, C.; Zarubina, A.V.; Messinger, J.D.; Ach, T.; Curcio, C.A. Quantifying Retinal Pigment Epithelium Dysmorphia and Loss of Histologic Autofluorescence in Age-Related Macular Degeneration. Invest Ophthalmol Vis Sci 2019, 60, 2481-2493, doi:10.1167/iovs.19-26949.